# Gold Nanomaterials-Based Electrochemical Sensors and Biosensors for Phenolic Antioxidants Detection: Recent Advances

**DOI:** 10.3390/nano12060959

**Published:** 2022-03-14

**Authors:** Rita Petrucci, Martina Bortolami, Paola Di Matteo, Antonella Curulli

**Affiliations:** 1Department of Basic and Applied Sciences of Engineering, Sapienza University of Rome, 00161 Rome, Italy; rita.petrucci@uniroma1.it (R.P.); martina.bortolami@uniroma1.it (M.B.); p.dimatteo@uniroma1.it (P.D.M.); 2Consiglio Nazionale delle Ricerche, Istituto per lo Studio dei Materiali Nanostrutturati, Unità Operativa di Support, Sapienza, 00161 Rome, Italy

**Keywords:** phenolic antioxidants, gold nanomaterials, electrochemical (bio)sensors, flavonoids, phenolic acids, stilbenes, synthetic antioxidants

## Abstract

Antioxidants play a central role in the development and production of food, cosmetics, and pharmaceuticals, to reduce oxidative processes in the human body. Among them, phenolic antioxidants are considered even more efficient than other antioxidants. They are divided into natural and synthetic. The natural antioxidants are generally found in plants and their synthetic counterparts are generally added as preventing agents of lipid oxidation during the processing and storage of fats, oils, and lipid-containing foods: All of them can exhibit different effects on human health, which are not always beneficial. Because of their relevant bioactivity and importance in several sectors, such as agro-food, pharmaceutical, and cosmetic, it is crucial to have fast and reliable analysis Rmethods available. In this review, different examples of gold nanomaterial-based electrochemical (bio)sensors used for the rapid and selective detection of phenolic compounds are analyzed and discussed, evidencing the important role of gold nanomaterials, and including systems with or without specific recognition elements, such as biomolecules, enzymes, etc. Moreover, a selection of gold nanomaterials involved in the designing of this kind of (bio)sensor is reported and critically analyzed. Finally, advantages, limitations, and potentialities for practical applications of gold nanomaterial-based electrochemical (bio)sensors for detecting phenolic antioxidants are discussed.

Table of Content
1. Introduction22. Au nanomaterials42.1 Au nanoparticles52.2 Au nanocages62.3 Nanoporous gold62.4 Au-based nanomaterials72.4.1 Bimetallic nanoparticles72.4.2 Au-based nanocomposites83. Gold nanomaterials applications to electrochemical sensors for phenolic antioxidants detection: some examples113.1. Phenolic acids113.1.1 Gallic acid113.1.2. Hydroxycinnamic acids133.1.3. Some consideration on phenolic acids (bio)sensors base on Au nanomaterials243.2. Stilbenes 273.3. Flavonoids293.3.1. Luteolin323.3.2. Myricetin343.3.3. Quercetin363.3.4. Rutin423.3.5. Catechin473.3.6. Some considerations on flavonoids (bio)sensors based on Au nanomaterials493.4. Synthetic phenolic antioxidants523.4.1. Butylated hydroxyanisole533.4.2. Tert-butylhydroquinone553.4.3. Propyl gallate and octyl gallate573.4.4. Some considerations on synthetic phenolic antioxidants (bio)sensors based on Au nanomaterials584. Conclusions60References63

## 1. Introduction

Antioxidants have received growing interest due to their role as protecting agents in food, cosmetics, and pharmaceutical products against oxidative degradation and oxidative stress-mediated pathological processes in the human body [1,2].

Among the different classes of antioxidants, phenolic compounds are one of the wider and more well-known groups. It is evident that this class of molecules includes all those with at least one phenolic unit in the structure, and it can be further divided in natural and synthetic phenolic antioxidants.

Among the natural antioxidants, there are compounds with only one phenolic ring such as phenolic acids and phenolic alcohols, and others with more than one phenolic ring commonly referred to as polyphenols. Different classifications have been reported in the literature, but according to the most accepted, they are grouped in phenolic acids, phenolic alcohols, lignans, stilbenes, and flavonoids [2,3]. Flavonoids can be further classified into six subclasses, namely flavonols, flavones, isoflavones, flavanones, anthocyanidins, and flavanols (catechins and proanthocyanidins) [3].

Natural phenolic antioxidants are secondary metabolites produced by plants, essential for growth and reproduction and protection from pathogen bacteria attacks [3,4,5].

Several recent reviews highlighted the phenolic antioxidants action against oxidative stress-mediated pathological processes [1,3,4,5,6,7] and have also evidenced that a diet rich in natural phenolic antioxidants is strictly connected to protective action against the development of cancer, cardiovascular diseases, diabetes, osteoporosis, and neurodegenerative diseases [1,3,4,5,6,7].

For these reasons, polyphenols have attracted the attention and interest of food producers and consumers. The European Commission provided the basic legal rules regarding nutrition claims in Europe through the European Commission Regulation No. 1924/2006 [8]. On this basis, many applications have been presented, considering the beneficial effects of polyphenols on human health according to the literature findings [9,10]. At present, the health claims related to polyphenols, approved by the European Food Safety Authority (EFSA), are limited to olive oil polyphenols [11,12]. All other applications, regarding several foods such as fruits, chocolate, honey, etc., have been rejected by EFSA due to not being substantiated by scientific evidence [9]. Consequently, it is necessary and mandatory to develop smart and reliable systems for natural polyphenols detection.

Synthetic phenolic antioxidants (SPAs) are purposefully prepared and used as additives to avoid or delay lipid oxidation due to fat degradation in oils and lipid-containing foods [1]. In this review, we have considered the most popular SPAs, such as butylated hydroxyanisole (BHA), tert-butylhydroquinone (TBHQ), propyl gallate (PG), and octyl gallate (OG). Compared to the corresponding natural antioxidants, i.e., tocopherols, present in plants and animals, the synthetic varieties show a stable structure, higher thermal stability, and stronger antioxidant capacity. Moreover, they are available in large quantities in order to meet the food industry demand, just to make an example.

Since they are added during food processing, all the corresponding steps have to be rigorously verified and regulated [13,14], including the additive quantity involved, to avoid undesirable side effects [13]. Within the EU, according to the EFSA evaluations, the added quantity of each additive is rigorously determined for each food typology, so that the daily consumption does not exceed the Admissible Daily Intake (ADI), which represents the total tolerated amount of a particular additive that can be ingested daily over a lifetime without appreciable unwanted effects on health [14].

For example, the ADI for TBHQ is 0.7 mg/kg body weight [15], for BHA is 1 mg/kg body weight [16], and for PG is 1.4 mg/kg body weight [17], while EFSA did not determine the ADI for OG because no particularly new or relevant data on kinetics and metabolism have been provided [18].

It is important to note, as the SPAs are classified as preservatives due to their action against lipid/fat oxidation, they are labelled with a number preceded by the letter E, which represents Europe. Considering the whole class of preservatives, the number ranges from 200 to 399. In particular, the SPAs present in this review are labelled E 319 for TBHQ, E 320 for BHA, E 310 for PG, and E 311 for OG.

Antioxidants’ action against oxidative stress-mediated pathological processes in humans is strictly connected to the scavenging activity towards different types of free radicals, which are considered responsible for pathological diseases such as cancer and diabetes, among others, and to the inhibition of the correlated oxidative mechanisms. Consequently, several detection methods for the antioxidant capacity (AOC) have been proposed and reported in the literature, based on different methodological and instrumental approaches, ranging from High-Performance Liquid Chromatography (HPLC) to Nuclear Magnetic Resonance (NMR) and Capillary Electrophoresis (CE) [5,19]. The corresponding analytical protocols and procedures are well-known, such as oxygen radical absorbance capacity (ORAC), Folin–Ciocalteu (FC), 2,2′-azinobis-3-ethylbenzothiazoline-6-sulphonic acid/Trolox equivalent (ABTS/TEAC) antioxidant capacity, the 2,2-diphenyl-1-picrylhydrazyl (DPPH) free radical scavenging method, ferric-reducing antioxidant power (FRAP), and cupric-reducing antioxidant capacity (CUPRAC) [1,6,20,21,22,23]. All these approaches are based on different principles, giving rise to results that are produced through different methods and therefore not comparable with each other. Finally, we would like to evidence that there is an objective lack of selectivity in relation to the determination of a single species. On the other hand, as already reported above, it is necessary and mandatory to develop smart and selective systems for the detection and quantification of specific antioxidants.

This review is focused on the application of different gold-based nanomaterials for electrochemical biosensing and sensing systems to detect particular and significant antioxidants, both natural and synthetic. In the literature, several recent reviews are focused on the application of nanomaterials in the electrochemical (bio)sensing area and beyond [4,6,19,24,25,26,27,28,29,30,31,32,33,34] for the detection of different analytes, including antioxidants. It should be stressed, however, that their focus is mainly on the application of carbon-based nanomaterials, such as carbon nanotubes and graphene, among others, with minor attention paid to the application of gold-based nanomaterials.

This review aims to provide an informative overview of which gold-based nanomaterials are applied to the design and assembly of electrochemical (bio)sensors, highlighting the examples related to the determination of phenolic antioxidants and finally indicating the strengths, limits, and future perspectives.

We organized this review in two parts. The first part briefly describes the nanomaterials used in the mentioned (bio)sensors, and the second part reports the most significant examples regarding the determination of antioxidants.

With regard to the (bio)sensors, special attention is paid to the role of the nanomaterial involved, the type of receptors, the selectivity of the (bio)sensors, the possibility of using them on real samples and in real environments, and finally, comparisons with official validation methods.

Brief comments and/or observations are reported at the end of each subsection, but a more detailed and in-depth discussion can be found in Section 4.

In this review, different electrochemical techniques are mentioned, such as chronoamperometry (CA), cyclic voltammetry (CV), linear sweep voltammetry (LSV), differential pulse voltammetry (DPV), square-wave voltammetry (SWV), and electrochemical impedance spectroscopy (EIS).

For more details about the theories underlying the different electrochemical approaches used for electrochemical (bio)sensors, several books and reviews are available in the literature [24,35,36,37,38].

## 2. Au Nanomaterials

Concerning the area of electrochemical (bio)sensors, increasing attention has been paid by researchers to gold-based nanomaterials for their biocompatibility, good thermal and electrical conductivity, chemical stability, and high volume/surface ratio [39]. Different types of gold nanomaterials have been reported in the literature such as nanoparticles, nanocages, nanorods, nanowires, bimetallic nanoparticles, nanothorns, nanoporous gold, and nanocomposites, to name the most significant. Figure 1 introduces the most significant applications of Au and Au-based nanomaterials to the electrochemical sensing area.

In addition, several synthesis methods have been developed, including physical, chemical, biological, and electrochemical ones [39]. In this review, the gold nanomaterials mentioned in the various examples of electrochemical (bio)sensors for the determination of antioxidants will be briefly introduced.

### 2.1. Au Nanoparticles

Numerous methods for the fabrication of gold nanoparticles (AuNPs), such as reduction, photochemical reduction, and seed growth, have been reported [40]. Conventional synthesis methods, i.e., Turkevich–Frens and Brust–Schiffrin methods, are, even today, well known, very widespread, and frequently used [40]. On the other hand, new methods such as the seed-mediated growth method and green synthesis have been developed.

Considering the seed-mediated method, small-sized AuNPs are firstly prepared via reducing gold salts with a strong and conventional reducing agent, and then used as seeds to obtain larger-sized AuNPs by adding a gold salt solution in the presence of a weaker reducing agent, such as ascorbic acid (AA). Finally, a structure-organizing and -dispersing agent is added to prevent further nucleation and/or aggregation and to support the AuNPs’ growth. However, it is difficult to optimize the growth conditions so as to prevent additional nucleation during the growing step; moreover, it is not very easy to simultaneously control all experimental parameters, such as the seed concentration, reducing agents, and rate of reducing agent addition [40].

Recently, aiming to avoid or limit the use of toxic solvents and chemicals and the consequent environmental impact, green AuNPs synthesis methods have been investigated using green materials such as bacteria and fungi and/or plant extracts and supported by techniques such as photochemistry or microwave. The corresponding synthetic procedure is generally divided into two parts: The intracellular biosynthesis and the extra-cellular biosynthesis. The results seem to be very promising, but the experimental protocols require some improvement to optimize these innovative and green synthetic approaches [40,41].

In Figure 2, all the synthesis methods reported in the literature are represented and summarized.

As a final comment, the preparation of AuNPs with a controlled morphology produced a wide range of different nanoparticle geometries, such as nanorods, nanotriangles, nanocubes, nanostars, and nanothorns, among others. These non-spherical nanoparticles are defined as anisotropic nanoparticles, i.e., they have shape-dependent chemical and physical properties, and seem to be very attractive, in addition to their spherical counterparts, for applications in sensing areas.

In order to deepen the topic, we suggest several particularly meaningful reviews and papers [40,41,42,43,44,45,46,47,48], including an interesting example of a hybrid organic–inorganic metal nanoparticles application to develop a localized surface plasmon resonance (LSPR)-based biosensor for antibiotics detection [47].

### 2.2. Au Nanocages

Au nanocages are hollow porous Au nanoparticles, with hollow interiors and porous walls, and their sizes range of 10 to 150 nm. The conventional synthetic procedure consists of a galvanic replacement involving HAuCl_4_ and Ag nanostructures such as nanocubes and/or nanospheres. Tuning the molar ratio of Ag to HAuCl_4_ and wall thickness allows one to control and tailor the properties of the resulting nanocages [49,50,51].

The principal types of Au nanocages, derived from different types of Ag templates, are reported in Figure 3.

In addition, Raveendran and co-workers [52] reported an innovative and more sustainable method for the synthesis of Au nanocages by microwave heating. This method presents several advantages with respect to the conventional one, such as a shorter reaction time, milder laboratory conditions, the possibility of large-scale production, and accurate monitoring of the temperature and power.

Finally, it is to be underlined that particular properties of Au nanocages, such as compact sizes and biocompatibility combined with the ease of surface modification, make them interesting materials for electrochemical (bio)sensing area [50].

### 2.3. Nanoporous Gold

In recent years, nanoporous metals have gained increasing relevance in the (bio)sensing area because these materials possess a high specific surface area, well-defined pore sizes, and functional sites, properties required for developing smart and innovative sensing devices [53,54,55]. In this context, nanoporous gold (NPG) seems to be one of the most interesting and appealing nanoporous materials, because it has several particular features, i.e., high surface area, electrochemical activity, biocompatibility, and sustainability, in addition to simple synthesis procedures [53,54,55]. The most widespread approaches for NPG synthesis involve the template method, de-alloying, and electrochemical methods [53,54,55].

The template method uses organic or inorganic porous material as a template, including, for example, anodic aluminum oxide (AAO). Gold is incorporated into the template and, finally, after removing the original template via annealing and/or dissolution, NPG is obtained. Generally, this approach is used for synthesizing highly ordered NPG and it is not convenient for large-scale preparation.

Considering the de-alloying method, at least binary or multicomponent alloys are involved. The less-noble metals present in the alloy are oxidized, so the gold atoms are aggregated, and a complex NPG structure is created. It is evident that the de-alloying method involves a simple and convenient operative procedure, resulting in an NPG uniform structure with controllable porosity through the corrosion parameters. In addition, this synthetic method is applied to NPG production on a large scale.

The electrochemical method can be considered an ideal method for NPG preparation. By means of a simple and convenient operative procedure involving template-free one-step electrodeposition followed by a galvanic metal displacement (GMD), it is possible to tailor and modulate the morphology and porosity of NPG by modifying the electrodeposition parameters, and to follow the reaction progress.

### 2.4. Au-Based Nanomaterials

For synergic improvement of the peculiar physical, electrical, and optical properties of gold nanomaterials, by combining them with other nanomaterials, several interesting Au-based nanomaterials have been designed and prepared for different applications in the sensing area. Au nanostructures can be easily modified and/or functionalized with other molecules or nanomaterials, such as metal and metal oxide-nanoparticles, polymers, and carbon nanomaterials.

#### 2.4.1. Bimetallic Nanoparticles

By combining two different metals into bimetallic nanoparticles, it is possible not only to improve the particular properties of the corresponding monometallic nanoparticles, but also to obtain a new nanomaterial with different and novel properties, deriving from the two metals’ synergistic blending [39].

In the literature, there are two main approaches to synthesize bimetallic nanoparticles: Bottom-up (via the two metal cations’ reduction) and top-down (by cutting pieces of nanoscale-level larger objects, for example, by laser ablation) [56,57]. Figure 4 illustrates the two synthetic pathways.

The bottom-up method usually requires metal cations as soluble salts, generally in an aqueous solution, where a reducing agent able to determine and affect the particles properties is added. The presence of the two metal cations can also influence the properties and nature of the resulting bimetallic nanoparticles. In other words, if the metal cations are both present in the same solution, the resulting nanoparticles are alloyed nanoparticles. On the other hand, a sequential addition induces so-called seeded growth, leading to core-shell particles. The nobler metal (Au) precursor is reduced first, and the resulting nanoparticles act as seeds. After the addition of the less-noble metal (Ag) precursor and a further amount of the reducing agent, the obtained bimetallic nanoparticles present a core composed of the nobler metal (Au) and a shell made of the less-noble counterpart. The opposite sequence, i.e., first the reduction of the less-noble metal, is more problematic, because the redox chemistry is difficult to control at the nanoscale level, even if using a stronger reducing agent together with the nobler metal.

Laser ablation is the most significant top-down method. In this case, for example, a bimetallic alloy is treated with a laser beam. Under optimized experimental conditions, it is possible to obtain well-dispersed bimetallic nanoparticles, and they can be further cut in pieces at nanoscale. Another option is a two-step synthesis, i.e., mixing of nanoparticles of the two different metals can be irradiated by a laser beam. By modulating and controlling the energy of the laser beam, it is possible to alternatively synthesize nanoalloys or bimetallic nanoparticles [57].

#### 2.4.2. Au-Based Nanocomposites

Nanocomposites can be defined as a synergistic blending of nanomaterials with polymers and/or other nanoscale materials, such as nanotubes, nanowires, nanorods, quantum dots, nanoclusters, or nanoparticles. The result is a new nanomaterial, which not only improves the properties of the starting materials but can also provide very peculiar and interesting characteristics. Recently, numerous gold-based nanocomposites have been designed and synthesized, such as metal oxide-gold nanocomposites, gold-carbon nanomaterials nanocomposites, and polymer-gold nanocomposites. All these nanocomposites have found applications in the (bio)sensing area [39,58].

As the first example, we would like to introduce Au-metal oxide nanocomposites. They have been applied in different research areas [59] and they have demonstrated the feasibility of the integration of two different nanostructured materials to obtain unique nanohybrid composites with appropriate functionalities.

As reported in the literature [59], five different classes of noble metal-metal oxide nanocomposites can be identified: (1) Noble metal-decorated metal oxide nanoparticles; (2) noble metal-decorated metal oxide nanoarrays; (3) noble metal–metal oxide core-shell nanostructures; (4) noble metal–metal oxide yolk-shell nanostructures; and (5) Janus noble metal–metal oxide nanostructures. In this review, the reported examples of electrochemical (bio)sensing devices include only metal oxide-gold nanoparticles because in the literature it is the only nanocomposite among those mentioned above to be used in sensors for the detection of phenolic antioxidants, as far as we know.

The synthetic approach is very important to define, tailor, and design properties and applications of nanocomposites. To synthesize metal oxide-gold nanoparticles, various methodological procedures have been developed, and we would like to mention the most important and widespread, i.e., chemical reduction, photoreduction, electrodeposition, and deposition-precipitation methods [59]. All of these procedures involve sequential reductions, starting from the precursor of the metal oxide nanoparticles and then moving to the gold nanoparticles precursor. The eventual deterioration of the nanocomposite electronic properties, due to the residual reducing agent used in the chemical reduction, can be avoided by means of photoreduction or electrodeposition, since photo- or electro-generated electrons can act as reducing agents.

Recently, many (bio)sensing devices based on gold-carbon nanocomposites have been developed, and in particular, we focused our attention on AuNPs-carbon nanotubes (CNTs) and AuNPs-graphene nanohybrids. They are the nanocomposites mainly involved in the design and assembly of electrochemical (bio)sensors for the determination of phenolic antioxidants. Thanks to peculiar properties, such as high conductivity, electron transfer improvements, and a large surface area, CNTs represent a class of nanomaterials that has found great possibilities of application in the (bio)sensing area. Moreover, the ease of modifying CNTs made them one of the most studied and used among nanostructured materials for the synthesis of nanocomposites with AuNPs. Two different modification methods are generally employed to prepare Au-CNTs nanocomposites. The first involves the direct attachment of Au nanostructures to CNTs, while the second includes the presence of a type of connection among Au nanostructures and CNTs, such as a Au-S bond (self-assembly) and/or non-covalent links including π-π stacking, hydrophobic interactions, and electrostatic forces [39,60].

Currently, graphene is the most-used carbon-based nanomaterial to synthesize gold-based nanocomposites to assemble sensors for the determination of phenolic antioxidants, as will be evident from the examples given in Section 3. In fact, graphene has more attractive properties than CNTs for the realization of sensing devices and for integration with Au nanostructures, such as the absence of metal impurities and higher surface area, resulting in better interfacial contact with other eventual components of nanohybrid and 2D structures, which are very useful in the complex architecture of a nanocomposite [61].

Furthermore, the presence of noble metal nanoparticles in a nanohybrid formulation, working as nano-spacers and improving the electrical conductivity among graphene layers, prevents or limits the graphene layers’ agglomeration.

In the literature [62], graphene (G) is used as graphene oxide (GO) and reduced graphene oxide (rGO). It should be remembered that reduced graphene oxide can be obtained both chemically and electrochemically (ErGO).

The synthetic procedures of AuNPs-graphene nanocomposites can be divided into two groups, as illustrated in Figure 5: AuNPs-embedded graphene nanocomposites and graphene-encapsulated AuNPs [61]. In this review, the reported examples include only the first option as the nanohybrid preparation method. The first group, i.e., AuNPs-embedded graphene nanocomposites, can be further subdivided into two classes of procedures: In situ and ex situ.

The in situ method involves metal seed formation in the presence of pristine or functionalized graphene nanostructures, followed by the successive growth of AuNPs onto the graphene surfaces. The other technique, i.e., ex situ, includes separate synthesis of nanoparticles in appropriate sizes and shapes, followed by subsequent attachment to the surface of the properly functionalized graphene, including either covalent or noncovalent interactions, such as van der Waals interactions, hydrogen bonding, π-π stacking, or electrostatic interactions.

It should be stressed that the in situ method may involve several experimental approaches, among which we highlight the simultaneous reduction and electrochemical reduction methods. In the case of simultaneous reduction, the nanocomposite synthetic pathway is the simultaneous reduction of Au-metal precursors and graphene nanostructures in the same solution. The fundamental principle is that the graphene surface functional groups can induce the link of Au ions through electrostatic interactions, and the reducing agent addition can speed up the process.

The electrochemical reduction method is a flexible, fast, and green approach for Au-graphene nanohybrids synthesis. The classical electrochemical deposition method results in the deposition of graphene layers on the electrodic surface, followed by the immersion of the modified electrode in an electrolytic solution containing the metal precursor, and finally the application of an appropriate electrochemical potential.

Recently, other carbon-based materials have been employed in the formulation of AuNPs-carbon nanocomposites, such as carbon black (CB) graphite paper and graphene quantum dots (GQDs). In particular, GQDs are defined as a 0D nanomaterial with properties coming from both graphene and carbon dots (CDs) [63,64].

Compared their precursor graphene, GQDs have more accessible edges, more surface-active sites, and a larger specific surface area. In addition, they are easy to prepare, and a result of these characteristics, their application in developing nanohybrids with noble nanomaterials, and in particular with AuNPs, is currently growing. In general, the corresponding nanohybrid synthetic methods mainly include chemical reduction, thee hydrothermal method, and the assembled method, each providing different levels of control of the size, morphology, and distribution of AuNPs and GQDs [64]. Under the chemical reduction protocol, AuNPs can grow on the dots’ surface because of the functional groups of GQDs bearing oxygen moieties, therefore accelerating the formation of nanoparticles by means of a reduction pathway. Otherwise, it is possible to synthetize the AuNPs separately, to better control and tailor their morphology and size, and then to modify the GQDs.

Finally, under the hydrothermal method and electrostatic assembling method, GQDs grow preferentially on AuNPs surface, forming a “dot on particles” structure [65].

Gold nanoparticles can be incorporated in conducting polymers, and the resulting nanocomposites have been successfully applied in different research fields, including the sensing area. The conducting polymers’ (CPs) high conductivity is explained by the presence of mobile electrons along the polymer structure; moreover, CPs are materials with high porosity and roughness, which are very attractive properties for electrochemical sensor assembly and increasing the electrochemical active surface area [66,67].

Au nanoparticles incorporated in CPs supply electrocatalytic sites to the resulting nanocomposite and, in addition, the conducting polymer thin film acts as a dispersing agent for metal nanoparticles [39,68,69]. Several CPs, such as polythiophene, polypyrrole (PPy), polyaniline (PANI), and poly(3,4-ethylenedioxythiophene) (PEDOT), have been prepared and widely exploited in combination with AuNPs, using different synthetic procedures.

The corresponding nanocomposites can be synthesized and deposited onto the electrodic surface mainly by chemical and electrochemical methods, or alternatively, by a combination of them.

Chemical methods comprise the sol-gel method, template synthesis, or synthesis of the components separately followed by the two components mixing [68,69]. The nanohybrids can be deposited on the electrode by spin coating, dip coating, drop casting, layer-by-layer assembly, or the Langmuir Blodgett technique.

Electrochemical methods seem more convenient than chemical ones, because they include a more sustainable single-step process. Finally, in situ electropolymerization, i.e., electrochemical polymerization of the conducting polymer in the presence of AuNPs, is a very efficient synthetic method, resulting in the most widely used approach [68,69].

Not only are CPs used to synthesize nanocomposites with gold nanoparticles but also polymers of natural origin, such as chitosan, gelatin, cyclodextrin, etc. They can act as a dispersing, stabilizing, and/or reducing agent, avoiding the agglomeration of the nanoparticles and creating a synergistic network with AuNPs, able to significantly affect the sensor performances and especially to provide a green and sustainable approach to the synthesis of nanocomposites [39,40].

As a final comment, we point out that gold-based nanocomposites might include more than two nanomaterials, for example a combination of AuNPs, graphene, conducting polymers, and so on. As we will see in Section 3, many nanocomposites present very complex structures and architectures, but this does not always correspond to a highly performing sensor.

## 3. Gold Nanomaterials Applications to Electrochemical Sensors for Phenolic Antioxi-Dants Detection: Some Examples

In this section, several examples of electrochemical sensors employing gold-based nanomaterials and applied to detect different phenolic antioxidants are illustrated and discussed.

### 3.1. Phenolic Acids

Phenolic acids represent one of the most important group of natural antioxidants, comprising two main subgroups: Hydroxycinnamic and hydroxybenzoic acids and their derivatives. The hydroxybenzoic acids are benzoic acid derivatives, while the hydroxycinnamic acids are cinnamic acid derivatives [2,4].

They are widespread in plant, fruits, seeds, roots, and food products such as grapes, beer, coffee, tea, officinal herbs, honey, fruit juices, and so on. Phenolic acids are regarded as powerful antioxidants, and their antioxidant capacity depends on the number and position of the -OH moieties present on the benzene ring.

A recent review stigmatizes the beneficial effects on human health ascribed to the intake of natural phenolic acids, and the effects on their antioxidant properties under different processing conditions [70]. Processing methods are carefully analyzed, and phenolic acids are reported to behave differently according to the experimental procedures; higher preservation of the phenolic acids content and, consequently, their antioxidant capacity, was observed under mild processing techniques such as pasteurization, with respect to other more severe processing conditions, such as frying, boiling, and/or steaming.

#### 3.1.1. Gallic Acid

Regarding hydroxybenzoic acids, gallic acid (GA, chemical structure in Figure 6), is one of the most important antioxidants belonging to this subgroup.

GA can be found in tea leaves, grapes, red fruits such as strawberries, berries, and so on, as well as in many other officinal herbs and in drinks and beverages, such as teas and red wine [71]. Due to its antioxidant capacity, it was regarded as effective against the development of different diseases such as cancer, cardiovascular, and neurodegenerative diseases [71].

As a first example, we would like to introduce a sensor based on gold microclusters (AuMCs) electrodeposited on sulfonate functionalized graphene (SFG). The nanocomposite was used to modify a glassy carbon electrode (GCE) for the detection of GA in black teas and urine samples in the presence of uric acid [72].

AuMCs can be understood as dense nanoparticles and, from a morphological investigation, they are spheroidal and uniformly deposited on the film of SFG with an average diameter of around 18 nm. Moreover, the modified electrode surface evidenced a high roughness, improving the electrochemically active electrodic surface.

Under optimized conditions, the GA amount was evaluated by DPV, showing a linear concentration range of 0.05 to 8.0 μM and a detection limit (LOD) of 10.7 nM. Moreover, repeatability, reproducibility, and stability were investigated, and results can be considered satisfactory. The sensor selectivity was analyzed at different concentrations of different potential interferences including K^+^, Na^+^, Ni^+^, Ca^2+^, Mn^2+^, Cu^2+^, ascorbic acid, theophylline, caffeine, fructose, sucrose, tryptophan, tyrosine, and cysteine. The sensor performances were not particularly affected by the presence of interferences, except for GA at a concentration level comparable to that of polyphenolic compounds such as rutin, catechin, caffeic acid, or quercetin. All these molecules present a catechol (1,2-dihydroxybenzene) moiety in their chemical structure.

Finally, the sensor was applied to real samples of black teas and urine, and the resulting recoveries, ranging from 96.0% to 102.4%, were comparable with those obtained by HPLC, ranging from 99.2% to 104.0%.

The next sensor involves the detection of GA, again in urine, but this time in the presence of uric acid and in green teas and in fruit juices [73].

A nanocomposite including zirconia nanoparticles (ZrO_2_NPs), choline chloride (ChCl), and gold nanoparticles (AuNPs) has been employed to modify a carbon paste electrode (CPE). Firstly, ZrO_2_NPs were synthesized via the sol-gel method and functionalized with ChCl, and then AuNPs were prepared via chemical reduction and supported on the functionalized ZrO_2_NPs. The ZrO_2_-ChCl-AuNPs/CPE was obtained by mixing the appropriate amount of ZrO_2_-ChCl-AuNPs with graphite and paraffin. The presence of ZrO_2_-ChCl and AuNPs at the same time in the nanocomposite enhanced the electrocatalytic and conductivity properties of the electrode against those of a CPE modified with only ZrO_2_-ChCl or AuNPs.

Under optimized experimental conditions, by means of DPV, a linear concentration range was obtained from 0.22 to 55 μM, with a detection limit of 0.025 μM. Repeatability (RSD 1.9) and reproducibility (RSD 3.95%) of the sensor were investigated, obtaining interesting results. The sensor resulted in being stable if stored for three weeks, and no considerable fluctuation of the electrochemical response was observed (6.8% for GA). Considering the sensor selectivity, none of the evaluated interfering compounds affected the sensor performances, but this time, the effect of polyphenolic compounds [72] was not considered. ZrO_2_-ChCl-AuNPs/CPE was applied for the determination of GA in green tea, fruit juices, and urine in the presence of uric acid, obtaining satisfactory results in terms of the corresponding recoveries, ranging from 97.9% to 103.8%.

The next example describes a biosensor based on the immobilization of tyrosinase (TYR) by crosslinking AuNPs coated with an eggshell membrane (ESM) and deposited on a glassy carbon electrode (GCE) [74]. The morphological investigation evidenced that AuNPs as well as the enzyme were adsorbed and bound on the ESM surface.

The AuNPs’ action enhanced the electron transfer from and to the electrode surface and consequently improved the biosensor performances with respect to the biosensor assembled without them. The biosensor has been applied in the detection by DPV of GA, caffeic acid (CA), and catechin hydrate (CH): The linear concentration range was from 6 μM to 70 μM for GA, from 5 μM to 65 μM for CA, and from 5 μM to 115 μM for CH. The detection limits were of 1.707 μM, 0.752 μM, and 0.714 μM for GA, CA, and CH, respectively.

The effect of common interferences such as glucose, fructose, ascorbic acid, ethanol, and acetic acid was analyzed, and none of the tested interfering compounds affected the biosensor performances. Reproducibility (RSD 4.78%) and reusability (RSD 2.2%) were evaluated, with interesting results. The sensor resulted in being stable if stored for one month in the refrigerator, and a decrease in the response of 7.0% was observed. Finally, the biosensor was applied to determine the total polyphenol content in black teas and red wine, with results comparable with those obtained by HPLC. In our opinion, the introduction of eggshell to support the enzyme immobilization made this sensor one of the first examples of an eco-friendly and cost-effective biosensor.

As a last example for the GA detection sensor, we illustrate an electrochemical sensor based on a screen-printed carbon electrode (SPCE) modified with AuNPs, synthesized by a green method using an extract of *Acanthophora algae* as the reducing agent [75]. This method can be considered cost effective, green, nontoxic, and fast, producing high-purity nanoparticles. The modified electrode was electrochemically characterized and compared with the bare one, evidencing the role of AuNPs in increasing the electrical conductivity.

The resulting electrochemical sensor was employed to evaluate the total phenolic content in the *Acanthophora algae* extracts. The total phenolic content was expressed in terms of the GA content in the dried extract (DE), i.e., 1.34 mg (GA)/g of DE.

No other data on the analytical performances, such as the linear concentration range, LOD, selectivity, reproducibility, and so on, were provided.

#### 3.1.2. Hydroxycinnamic Acids

The second subgroup of phenolic acids involves hydroxycinnamic acids and their derivatives. They are synthesized by the shikimic acid pathway, a metabolic route generally operating in plants, but also in fungi and algae [4,76,77,78]. Figure 7 summarizes the chemical structures of the hydroxycinnamic acids and derivatives considered in this review: Caffeic acid, ferulic acid, chlorogenic acid, and rosmarinic acid.

Caffeic acid (CA, 3,4-dihydroxycinnamic acid, chemical structure in Figure 7) is usually present in red wine, green tea, coffee, fruits, and vegetables, it displays antibacterial and anti-inflammatory activity, and it is one of the most common antioxidants.

Recently, a review reported the most representative examples of electrochemical sensors or biosensors based on carbon nanomaterials for detecting CA, illustrating the possible future developments for enhancing (bio)sensor performances [79].

A sensor based on a gold electrode modified with a nanocomposite including chitosan (CHI) and AuNPs [80,81] is introduced as the first case of a sensor applied for the detection of CA in complex matrix. In this approach, a green synthetic procedure [80] was investigated to create a collaborative network of AuNPs highly dispersed in the CHI matrix. The nanostructure and surface functionalities of AuNPs-CHI-modified electrodes play a key role; in fact, the presence of AuNPs can significantly affect the electron transfer from and to the electrode and the conductivity of the nanocomposite, while the surface functional groups can support the interaction with CA.

DPV was employed to determine CA, with a linear concentration range of 5.00 × 10^−8^ M to 2.00 × 10^−3^ M and a limit of detection of 2.50 × 10^−8^ M. Stability, repeatability, and reproducibility were tested, obtaining interesting results in terms of RSD%.

Potential interfering agents present in wines and beverages, such as other phenolic antioxidants (gallic and protocatecuic acid, chlorogenic and ferulic acid, rutin, quercetin, and catechin) and nonphenolic antioxidants (ascorbic acid) were tested. The sensor resulted in being selective considering the interference concentrations generally involved in wines and beverages such as fruit juices and soft drinks. CA was also determined in real matrices, i.e., commercial white and red wine, after simple, appropriate dilution with the supporting electrolyte, obtaining data in good agreement with the literature.

The next examples involve nanocomposites including PEDOT and AuNPs.

The first one is a sensor obtained by modifying GCE with a nanocomposite where AuNPs were embedded in the polymer matrix [82]. First, a PEDOT layer was prepared on the GCE using sinusoidal voltage (SV) superimposed on a constant potential. In the second step, AuNPs were electrodeposited in situ onto the PEDOT layer, always by means of the SV method.

AuNPs electrodeposited by SV method displayed a more regular size distribution and a more homogeneous dispersion onto the PEDOT layer, if compared with the AuNPs electrodeposited by the potentiostatic method or drop-casted on the polymer. Consequently, the AuNPs obtained by the SV method were used to assemble the corresponding sensor (PEDOT-AuNPs-SV/GCE) to determine CA. The linear concentration range obtained was from 10 μM to 1 mM, with an LOD of 4.24 μM; the reproducibility was also investigated with promising results in terms of RSD%. In addition, the sensor was applied in the determination of the total polyphenols content in juice samples, expressed as equivalents of CA, with values comparable to those obtained by chromatographic methods.

A more complex architecture of the nanocomposite, including PEDOT and AuNPs, is involved in the successive example [83]. Carbon spheres, and in particular resorcinol formaldehyde yolk-shell-structured carbon spheres (YRFC), can represent an alternative material for modifying electrodes because of their high surface area, large pore volume, and good chemical and thermal stabilities. A combination of YRFC with thiol-grafted PEDOT (PEDOT-MeSH, methanethiol) and AuNPs was proposed to assemble a sensor for the detection of CA and the well-known antibiotic levofloxacin (LVF). The formulation of the nanocomposite includes the preparation of a binary composite (PEDOT-MeSH/YRFC) and inclusion of AuNPs on PEDOT-MeSH/YRFC. PEDOT-MeSH was obtained by chemical oxidation of a thiol-grafted 3,4-ethylenedioxythiophene monomer (EDOT-MeSH) in the presence of YRFC [82]. From the morphological analysis, it is evident that YRFC can be considered a hard template where the polymer grew and where AuNPs are incorporated. Moreover, the surface of YRFC resulted in being uniformly coated by PEDOT-MeSH with well-dispersed AuNPs. The nanocomposite was used to modify GCE and determine CA and LVF by means of DPV. A linear detection range was obtained from 0.02 to 320 μM for both CA and LVF, with detection limits of 0.014 μM and 0.018 μM, respectively. Ascorbic acid, paracetamol, uric acid, and another antibiotic such as terramycin were considered as possible interfering compounds, but none of them affected the sensor performances. Reproducibility, reusability, and stability were tested, obtaining interesting data in terms of RSD%. Finally, the sensor was applied to real spiked samples of human serum and urine, obtaining interesting results in terms of recoveries, ranging from 97.4 to 105.0%.

Herein we introduce two biosensors, both including SPCEs and AuNPs.

In the first case, a biosensor for CA detection was obtained by modifying a SPCE with AuNPs, immobilizing the laccase (LAC) enzyme, and finally electrosynthesizing a PPY layer [84]. AuNPs were electrodeposited on the electrode surface, and then the enzyme was adsorbed overnight on the nanostructured material. The electropolymerization to prepare PPY was the final electrode modification step. The conducting polymer and the gold nanomaterial contribute to increasing the electrical conductivity and speeding up the electron transfer, although the enzyme slightly reduces this improvement because of its steric hindrance. On the other hand, the enzyme resulted in being fundamental for biosensor selectivity, being the key recognition element. Under optimized conditions, CA was determined by amperometry with a linear concentration range of 1 to 250 μM and a low detection limit of 0.83 μM.

The polyphenol content expressed as CA amount was detected in different propolis extracts and compared with results obtained through the official method of analysis, i.e., Folin–Ciocalteu (spectrophotometric) method. Results are not completely comparable, which could be due to the completely different methodological approach and different analysis time: 15 min (electrochemical) vs. 85 min (spectrophotometric).

In the second case, the biosensor was assembled by immobilizing the tyrosinase enzyme (TYR) on a commercially available AuNPs/SPCE by drop-casting and using glutaraldehyde as cross-linking agent [85]. The AuNPs/SPCE surface evidenced an almost uniform distribution of the nanoparticles. After TYR immobilization, a slight change in the electrode surface was observed, showing the enzymatic clusters crosslinked to it and firmly fixed on top of the sensing layer. The biosensor was employed to detect, via amperometry, different phenolic compounds, i.e., catechol, phenol, caffeic acid, and tyrosol, to evaluate the polyphenols content in commercial beers.

Concerning CA, a linear concentration range of 2.5 to 12.5 μM and a detection limit of 2.3 μM were obtained. By comparing the results from the different phenolic compounds analyzed, the polyphenols content was expressed in terms of tyrosol amount because of a wider linear concentration range and lower LOD.

Another sensor for the detection of CA was obtained by modifying a GCE with PtAuRu nanoparticles [86]. This trimetallic nanohybrid material was synthesized by a one-pot method, consisting of simultaneous chemical reduction of the corresponding metallic precursors. The resulting nanoparticles were homogeneous, and each nanoparticle contained all the metals used during the synthesis, i.e., Pt, Au, and Ru. It is not clear if they can be assumed as alloyed nanoparticles, as described in Section 2.4.1.

The electrochemical behavior of the modified electrode was compared with that of the bare one, and the increase in the electrochemical response can be correlated to the presence of the trimetallic nanomaterial and the consequent enhancement of the active electrodic surface. A further increase in the electrochemical response was observed when the sensor was illuminated with visible light during the CA determination. This effect, linked to visible light illumination, is probably due to the surface plasmon resonance (SPR) effect of Au in the nanoparticles. Under optimized conditions, including visible light illumination, the nanoparticles-modified GCE showed a linear concentration range of 0.7866 mM to 16.6 mM, with an LOD of 3.9 × 10^−7^ M. Repeatability, reusability, and stability were investigated, and interesting results were obtained. Unfortunately, this sensor was not applied to real samples.

Recently, semiconducting nanomaterials, and in particular graphitic carbon nitrides (g-C_3_N_4_), have attracted increasing attention in the (bio)sensing area because of their catalytic, optical, and electrical properties [87]. An overview of the g-C_3_N_4_-based nanomaterials used for sensors employed for environmental and healthcare applications has been reported in a review published in 2021 [87].

A nanocomposite involving gold nanoparticles and ultrathin g-C_3_N_4_ nanosheets was synthesized by means of the sonochemical technique and employed to modify an SPCE for CA determination [88]. g-C_3_N_4_ and AuNPs were prepared separately according to the corresponding literature procedures [40,87] and then the synthesis of the nanocomposite was performed by the ultrasonication of a solution containing both AuNPs and g-C_3_N_4_. The morphological analysis of the g-C_3_N_4_/AuNPs composite indicated a uniform distribution/decoration of AuNPs on the ultrathin g-C_3_N_4_ nanosheets.

The SPCE modification was obtained by drop-casting of the nanocomposite dispersion on the electrode surface. The modified SPCE was electrochemically characterized, evidencing an increase in the electrochemical response if compared to that of the bare electrode or to that of SPCE modified only with g-C_3_N_4_ nanosheets; this is probably due to the presence of AuNPs. Under optimized conditions, g-C_3_N_4_/AuNPs/SPCE showed linear concentrations for DPV from 0.5 to 155 nM with LOD of 0.1 nM and amperometry from 2.5 nM to 1025 nM with LOD of 0.5 nM. The sensor selectivity was tested against different possible interferences such as uric acid, gallic acid, ascorbic acid, catechol, hydroquinone, epinephrine, folic acid, L-dopa, and dopamine; they did not affect the CA response in the interference concentrations analyzed. Stability and reproducibility were investigated, and the obtained results can be considered acceptable. The sensor was then applied to detect CA in spiked real samples of fruits, leaves, and wines, with recoveries ranging from 98.0 to 100.3 %, evidencing the possible use of this sensor in food analysis and/or control laboratories.

A sensor based on an SPCE modified with a nanocomposite integrating tungsten disulfide (WS_2_) flakes, catechin-capped gold nanoparticles (AuNPs-CAT), and carbon black (CB) was developed for the simultaneous determination of hydroxycinnamic acids such as caffeic (CA), sinapic, and *p*-coumaric acids [89]. Selectivity, sensitivity, and reproducibility were explained with the well-known antifouling properties of WS_2_ flakes decorated with AuNP-CAT and incorporated in the CB conductive network. CB is an old and low-cost carbon-based nanomaterial evidencing a large surface area, excellent electrical conductivity, dispersibility in solvents, possible easy functionalization, and very accurate reviews are available in the literature for further details [90,91].

The synthesis of AuNPs-CAT involved a green synthetic approach where catechin acted as a reducing and dispersing agent. Moreover, the catechin shells around AuNPs allowed adsorption on the WS_2_ flakes. WS_2_ flakes’ modification with AuNPs-CAT was carried out using a sonochemical procedure. During this process, discoloration of the AuNPs-CAT solution was observed, indicating that the nanoparticles were adsorbed on WS_2_ flakes. The AuNPs-CAT/WS_2_ composite was then mixed in a CB dispersion, and the final nanocomposite was drop-casted onto the SPCE. We can evidence the different role of the nanocomposite components: CB and AuNPs-CAT together guarantee high conductivity and electroactivity, and WS_2_ guarantees a peculiar antifouling performance. After the analytical parameter optimization, linear concentration ranges were obtained for all the examined analytes. In particular, for CA, the linear concentration range was 0.3–200.0 μM with an LOD of 0.1 μM.

Reproducibility (RSD ≤ 3.0%) and repeatability (RSD ≤ 4.0%) were evaluated, with interesting results. The sensor resulted in beng stable for six months. The responses of the SPE-CB/WS2-AuNP-CT sensor decreased to 95% in the first two months and resulted in being stable for the following four months (up to 94%). The selectivity was also tested, selecting ions, sugars, and organic acids as interferences, and choosing the corresponding concentrations according to literature indications [89], and the sensor resulted in being selective. The sensor’s applicability to real samples was investigated, selecting five commercial products (rapeseed oil, *Kalanchoe crenata*, apple puree, homogenized apple, and apple juice), obtaining acceptable results in terms of recovery, ranging from 86 to 108%.

The combination of AuNPs and graphene or its derivatives is the basis of the nanocomposites used to assemble some significant examples of sensors for the determination of CA.

A possible combination is that in which the “ingredients” are nanoparticles and graphene nanosheets (GNs). Pang and coworkers [92] prepared a nanocomposite including AuNPs and GNs to modify a GCE for determining CA. A GNs dispersion was casted onto the electrode surface and then AuNPs were electrodeposited onto GNs/GCE. From the morphological analysis, AuNPs seemed uniformly distributed onto GNs, and GNs presented wrinkles like a paper sheet; the wrinkles avoid the agglomeration and aggregation of GNs. The nanocomposite evidenced higher electrical conductivity and higher electrochemical active surface area with respect to the bare and/or GNs modified GCE, promoting the electron transfer and corresponding electrocatalytic activity. The sensor was employed to detect CA by means of DPV. A linear concentration range of 5 × 10^−7^ to 5 × 10^−5^ M with a detection limit of 5 × 10^−8^ M was obtained.

Repeatability (RSD 3.4%) and reproducibility (RSD ≤ 4.7%) were evaluated, with interesting results. Concerning the stability, the sensor was stable if stored in dry conditions at room temperature for 30 days. The electrochemical response of the sensor decreased less than 10% of its initial value. Selectivity was examined considering phenolic acids, flavonoids, and other organic molecules such as ascorbic acid, urea, citric acid, and glucose as possible interferences. The results showed good selectivity.

AuNPs/GNs/GCE was applied to the determination of CA in spiked real drug samples, with recoveries ranging from 97.6% to 101.8%, and the CA contents agreed with those declared by manufacturers.

Antiochia [93] assembled two biosensors immobilizing the laccase enzyme from *Trametes versicolor (*TvL) onto commercial SPCEs, some of which were modified with multiwalled carbon nanotubes (MWCNTs) and others with G. AuNPs and the enzyme were deposited on both modified electrodes at the same time. Further, a polymeric film of poly(vinyl alcohol), *N*-methyl-4(4′-formylstyryl)pyridinium methosulfate acetal (PVA-SbQ) was employed to entrap TvL and AuNPs. In the corresponding experimental procedure, the photo-cross-linking of the styryl pyridinium groups of the PVA-SbQ creates a network for entrapping enzyme and AuNPs.

All the results coming from using AuNPs with different dimensions and the two different carbon nanomaterials were compared. Considering that the electrodes modified with G and MWCNTs were commercial, it was not possible to verify their structure, composition, and morphology. The best results in terms of analytical performances were obtained with MWCNTs and the smallest AuNPs. A possible explanation is that with a more nanostructured and nanosized electrode surface, a friendly environment for an enzyme’s favorable orientation is offered, improving the related electron transfer. In particular, the biosensor analytical performances for CA detection were obtained by chronoamperometry and the resulting linear range was 1–100 μM, with an LOD of 0.5 μM. Unfortunately, studies on stability, selectivity, reproducibility, and repeatability of the biosensor were missing, as well as the application to real samples.

Gao group proposed a Pd-AuNPs/PEDOT/reduced graphene oxide (Pd-AuNPs/PEDOT/rGO) nanocomposite, synthesized by a one-pot method, for modifying a GCE for CA determination by means of DPV [94].

During the nanohybrid synthesis, the metallic salt precursors were reduced by the monomer of the polymer, and the monomer was simultaneously chemically oxidized and polymerized. Next, in the same solution, GO was reduced by NaBH_4_. The molar ratio of the metallic nanoparticle precursor plays a fundamental role in the morphology and structure definition of the resulting nanocomposite. In particular, the molar ratio of the nanoparticle precursor equivalent to 1/2 was selected to guarantee uniform distribution of the spherical bimetallic nanoclusters on rGO, enhancing the diffusion of the analyte and the charge transfer between the composite and electrolyte. Moreover, the rGO and PEDOT action enhanced the electrical conductivity and the electron transfer rate of the composite.

The sensor was assembled by drop-casting a nanocomposite suspension on the GCE surface. The analytical parameters were optimized, and a linear concentration range of 0.001 to 55 μM resulted, with a detection limit of 0.37 nM. The selectivity of the sensor was investigated in the presence of common interfering agents, such as malic acid, tartaric acid, citric acid, ascorbic acid, catechol, glucose, carbamide, *p*-coumaric acid, gallic acid, vanillic acid, sinapic acid, and ferulic acid. Repeatability (RSD 3.52%) was evaluated, with interesting results. Concerning the stability, the sensor was stable if stored in air for 10 days; in fact, the current response decreased to about 3.97% of its initial value.

Pd-Au/PEDOT/rGO was also applied to determine CA in spiked diluted red wine samples with good recovery results, ranging from 98.0 to 104%.

Previously, Gao’s group [95] synthesized a AuNPs/PEDOT/rGO nanocomposite by the same one-pot method and used it to modify a GCE for CA determination via DPV. As for the bimetallic nanocomposite, the AuNPs precursor was reduced by the monomer of the polymer that was simultaneously chemically oxidized and polymerized. Next, in the same reaction mixture, graphene was reduced by NaBH_4_. The synergistic action of rGO, gold nanoparticles, and PEDOT improved the electrical conductivity and analyte diffusion and sped up the electron transfer rate. The sensor was assembled by drop-casting a nanocomposite suspension on the GCE surface. The analytical parameters were optimized, and a linear concentration range resulted from 0.01 to 46 μM, with a detection limit of 0.004 μM.

The selectivity of the sensor was investigated in the presence of the same interfering agents, such as malic acid, tartaric acid, citric acid, ascorbic acid, catechol, glucose, carbamide, *p*-coumaric acid, gallic acid, vanillic acid, sinapic acid, and ferulic acid, and the CA electrochemical response remained unchanged in the bulk solution with the interference concentration ranging from zero to 20-fold higher than that of CA. Reproducibility (RSD 2.58%) and repeatability (RSD 3.25%) were evaluated, with promising results. Concerning the stability, the sensor was stored in air for 7 days, and 98.25% of the initial electrochemical signal was obtained. After 14 days, 96.61% of the corresponding initial value was acquired, indicating good long-term stability of the modified electrode.

Au/PEDOT/rGO was also applied to determine CA in spiked diluted red wine samples, and the results agree with those present in the literature, but no recoveries data were provided.

If we want to analyze whether there has been a real improvement using bimetallic nanoclusters, this is evident from the analytical data of the sensor; in fact, the detection limit is 10-fold lower, and the linearity range is consequently wider.

Chen and co-workers proposed a sensor for CA detection based on nanocomposites including bimetallic Au/Pd nanoparticles and graphene flakes (Au/PdNPs/GF) [96].

The nanoparticles were simultaneously deposited on GF-modified GCE, by electrochemical deposition. From the material characterization, it is evident that the nanoparticles were uniformly distributed on graphene, but in our opinion, it is not sufficiently clear whether they can be considered alloyed nanoparticles. In addition, the synergistic action of bimetallic nanoparticles and graphene can be assumed to enhance the electrical conductivity and accelerate the electron transfer rate. The sensor assembly involved the drop-casting of a nanocomposite suspension on top of the electrode surface. After optimizing the experimental conditions, a linear concentration range of 0.03 to 938.97 μM with a detection limit of 0.006 μM was obtained by DPV. The sensor selectivity was investigated by testing 10-fold higher concentrations of dopamine, catechol, epinephrine, uric acid, hydroquinone, ascorbic acid, ferulic acid, and gallic acid; the determination of CA was not affected by the interferences examined. The sensor was considered stable. In fact, the storage stability of Au/PdNPs-GRF/GCE was monitored by storing it at room temperature in air for 15 days and the electrochemical signal decreased by only 1.7% of its initial value. Repeatability (RSD 1.2%) was considered acceptable in terms of RSD%.

Au/Pd/GF/GCE was applied to determine CA in commercial red wine samples, and no recoveries data, but rather only RSD%, were reported.

Harrath and co-workers reported green synthesis, under solar-light irradiation, of ternary nanocomposites including rGO nanosheets, AuNPs, and α-Fe_2_O_3_ nanoparticles, (Au@α-Fe_2_O_3_@rGO), using chlorophyll as the reductant and stabilizer for the metal nanoparticles. The resulting nanocomposite was applied to assemble an electrochemical sensor for CA detection [97]. The synthetic approach for the ternary nanohybrid involved two successive steps. The first one was the simultaneous GO hydrothermal reduction to rGO nanosheets and the hydrolysis of Fe^2+^ ions, so α-Fe_2_O_3_ nanoparticles resulted in being anchored to the surface of rGO nanosheets. The rGO oxygen moieties play a central role for the integration of AuNPs on the rGO nanosheets after the second synthetic step, i.e., the Au precursor photochemical reduction in the presence of chlorophyll. Finally, from the nanocomposite’s morphological and structural characterization, the α-Fe_2_O_3_ and AuNPs resulted in being uniformly distributed on the rGO sheets. The Au@α-Fe_2_O_3_@rGO nanocomposite was casted on GCE to assemble the CA sensor. The modified electrode was electrochemically characterized; the nanoparticles’ action synergistically improved the electrocatalytic performances of the sensor, while rGO acted as an electron transfer accelerator to and from the electrode surface. The CA detection was carried out by means of DPV, and a linear concentration range of 19 to 1869 μM with an LOD of 0.098 μM was obtained.

Reproducibility (RSD 2.70%) and repeatability (RSD 3.80%) were evaluated, with promising results. Concerning the stability, the sensor was stored at room temperature for 30 days and 93% of the initial electrochemical signal was obtained after this period, indicating the good long-term stability of the modified electrode. Considering the selectivity, different interferences were tested, such as ascorbic acid, dopamine, uric acid, glucose, Na^+^, NO_3_^-^, and Ca^2+^, and no particular effects on CA detection were evidenced. Finally, the sensor was applied to CA determination in spiked real coffee samples, but neither recovery data nor a comparison with an official validation analytical method were provided.

Herein, we would like to introduce an interesting and significant example of the application of AuNPs/GQDs nanocomposites for the detection of CA.

A gold nanoparticles–nitrogen-doped graphene quantum dots (AuNPs/NGQDs) nanohybrid material was prepared and employed to develop “on–off” nanosensors for photo-electrochemical (PEC) CA analysis under visible-light irradiation [98]. The PEC approach has recently attracted increasing attention as an analytical strategy, alternatively to more traditional methods such as chromatographic, fluorimetric, or electrochemical methods. Nevertheless, it should be stressed that the availability of effective photoactive materials is fundamental for smart sensors’ development.

For example, quantum dots and noble-metal nanoparticles, among others, are assumed to be interesting photoactive materials. Recently, nitrogen-doped graphene quantum dots (NGQDs), with unique quantum effects, being photostable and biocompatible, were applied for photocatalysis, as well as in the sensing area [99]. On the other hand, NGQDs’ slow electron conductivity limits their application in PEC analysis [99]. A strategy to overcome this limit is to combine them with two-dimensional and highly conductive nanomaterials such as AuNPs. In the literature, it is reported that the photocurrent response in NGQD/metallic NPs was higher than bare metallic NPs [98,99]. In fact, the charge transfer between the analyte and the nanocomposite can be accelerated, and consequently, the response signal can be enhanced. Moreover, the π–π stacking interaction between the NGQDs and the target can reduce the distance between the molecule and the electrode, therefore contributing to increasing the detection signal. Moreover, when compared to graphene sheets, NGQDs present a larger specific surface area and more accessible edges, making the absorption of the analyte more efficient. Therefore, the corresponding PEC sensing platform benefits of NGQDs’ quantum effects and AuNPs local surface plasmon resonance (LSPR) effects.

Concerning the determination of CA, under visible-light irradiation, NGQDs’ absorption of light energy produces NGQD* free radicals ready to oxidize CA. On the other hand, AuNPs are excited and produce a local LSPR effect, leading to the photo-generated electron holes. The photo-generated holes oxidize CA rapidly, transferring the electrons to the electrode [99]. Thus, the synergy of the NGQDs’ quantum effects and the AuNPs’ LSPR effect can perform PEC oxidation of CA. A schematic illustration of the sensing platform assembly and sensing mechanism is shown in Figure 8. It is obvious that the Au/NGQD composite can only work properly for PEC detection of CA under visible-light irradiation.

Upon modifying a GCE via drop-casting with a suspension containing the nanocomposite and then analyzing CA via amperometry, two linear concentration ranges of CA were yielded, the first from 0.11 to 30.25 μM and the second from 30.25 to 280.25 μM, while the detection limit was approximately 0.03 μM.

Stability and selectivity were examined using dopamine, uric acid, ascorbic acid, glucose, and rutin for the selectivity tests. The sensor resulted in being stable and selective except vs. dopamine, but the motivations explaining this behavior are not completely satisfactory. We have to point out that, unfortunately, reproducibility and repeatability of the sensor were not addressed or examined, nor was its possible application to real samples.

Another important hydroxycinnamic acid displaying beneficial effects on health such as antioxidant, antibacterial, anti-inflammatory, and antitumor activity [76], is chlorogenic acid (CGA). It is classified as an ester of caffeic acid and quinic acid, as shown in Figure 7, and it is present in several fruits, as well as in beverages such as coffee and tea.

Zhang et al. developed an interesting example of a CGA sensor synthesizing a structurally complex nanocomposite, including AuNPs, polyoxometalates (POMs), and 3D macroporous carbon (MPC) [100]. The same group prepared a similar nanocomposite using Pd nanoparticles instead of AuNPs to study the electrocatalytic behavior of hydrazine, H_2_O_2_ and nitrobenzene [101]. Considering the coupling of MPC with metallic nanoparticles, it should be underlined that synergistic enhancement of the electrocatalytic activity can be observed.

3D MPC is a carbon-based material, displaying a uniform and macroporous structure, high specific surface area, large pore volume, good electrical and thermal conductivity, and mechanical stability. It was applied in the sensing area in combination with carbon nanomaterials such as carbon nanotubes (CNTs), with interesting results [102].

POMs, in this case H_3_PW_12_O_40_, are polyatomic anionic ion clusters composed of d-block transition metal-oxides and act as reducing and capping agents for metal nanoparticles [100,101]. AuNPs synthesized using POMs as the reductant resulted in being distributed uniformly on the MPC surface. The resulting nanocomposite (Au-POMs-MPC) was dropped on GCE to assemble the CGA sensor.

The sensor’s analytical performance was evaluated by DPV. A linear concentration range of 2.28 nM to 3.24 μM was obtained with a detection limit of 2.15 nM. Reproducibility was evaluated as promising with an RSD value of 0.3%, while repeatability and selectivity data were not provided. The sensor was considered stable if stored at 4 °C for 15 days. The electrochemical response of the sensor decreased by approximately 4.2% of its initial value.

The sensor was applied to determine CGA in spiked pharmaceutical real samples, and results were compared with those obtained from the spectrophotometric analysis, evidencing acceptable results.

A sensor based on covalent organic frameworks (COFs) doped with AuNPs was developed for the determination of GCA in spiked real samples of coffee, apple juice, and honeysuckle [103]. COFs represent a class of multifunctional porous crystalline materials, involving 2D or 3D porous crystalline architectures composed of light elements (C, B, O, Si, and N) via strong covalent bonds (C-N, C=N, C=C-N, B-O) [104,105,106,107]. It is well known that AuNPs need a stabilizing and dispersing agent to avoid their aggregation and agglomeration. COFs can act as stabilizing and dispersing agents, and the resulting nanocomposite evidences improved electrical conductivity, thermal stability, and peculiar performances as electrodic materials.

1,3,5-tris(4-aminophenyl)benzene (TAPB) and 2,5-dimethoxyterephaldehyde (DMTP) were used as starting materials for COF synthesis according to the literature procedure [106]. AuNPs were then prepared “confined” in the COF’s structure by chemical reduction, as illustrated in Figure 9; from the morphological investigation, no Au aggregation and/or clusters were evidenced.

A nanocomposite suspension was dropped on GCE to assemble the sensor, and the analytical performances were investigated by DPV, as illustrated in Figure 10. A linear range of 1.0 × 10^−8^–4.0 × 10^−5^ M, an LOD of 9.5 × 10^−9^ M, as well as good repeatability in terms of RSD% were obtained. Reproducibility data were not provided.

Selectivity was also tested using different and common interfering ions and molecules. The results showed that 500-fold Fe^3+^, Na^+^, K^+^, Ca^2+^, Cu^2+^, Zn^2+^, Mg^2+^, PO_4_^3−^, NH_4_^+^, Cl^−^, and SO_4_^2−^, 100-fold uric acid, dopamine, guanine, L-dopa, glucose, hydroquinone, creatine, adenine, and catechol, 50-fold ascorbic acid, thymol, rutin, and quercetin, and 10-fold caffeic acid, gallic acid, and vanillic acid had no significant influence on the determination of CGA.

The sensor was applied to spiked real samples of coffee, apple juice, and honeysuckle, with recoveries ranging from 99.2 to 102.5%. These results were comparable with those obtained by HPLC.

Chauan et al. reported an electrochemical sensor for CGA detection modifying GCE via drop-casting of a suspension of AuNPs synthesized via chemical reduction [108]. The morphological analysis showed that AuNPs were uniformly distributed on the electrode’s surface. The sensor’s analytical performances were evaluated by SWV. A linear range of 0.4 to 3.6 μg mL^−1^ was obtained with a detection limit of 0.0140 μg mL^−1^.

Reproducibility, repeatability, and stability were studied with acceptable results in terms of RSD%. Interference analysis was carried out to analyze the sensor’s selectivity, using different ions and molecules, such as Na^+^, K^+^, Ca^2+^, Cu^2+^, Zn^2+^, Mg^2+^, glucose, sucrose, fructose, caffeic acid, and gallic acid. The interference effects were considered non-significant with respect to CGA detection. The modified electrode was applied to spiked real samples of black tea and green tea. Recovery was found in the range of 98–99%, but a comparison with other analytical methods was not provided.

Concerning CGA detection, Apetrei et al. compared the performances of different commercial SPCEs such as bare SPCEs, SPCEs modified with graphene, and finally, SPCEs modified with a graphene/AuNPs nanocomposite [109].

All the electrodes were electrochemically characterized, and the analytical performances were evaluated by CV. The best performances were obtained with SPCEs modified with the nanocomposite, with a linear range of 0.1–1.20 μM and an LOD of 0.062 μM. The operational stability was tested, and after 50 continuous measurements, the electrochemical signal was not significantly affected. Repeatability was analyzed, with interesting results in terms of RSD% (2.7%). Ascorbic, ferulic, and vanillic acid were tested as potential interfering compounds, but they did not significantly affect the CGA response.

The CGA sensor was applied to commercial nutraceutical products containing green coffee extracts. Results were comparable with those obtained by FT-IR spectroscopy, used as a validation method.

Ferulic acid (3-metoxy-4-hydroxycinnamic acid, FA, chemical structure in Figure 7) is another interesting hydroxycinnamic acid present in fruits, vegetables, rice, oats, and beverages such as wine and fruit juices, showing antioxidant, antibacterial, anti-inflammatory, antitumor, and even cutaneous photoprotective and antiaging properties [76]. It is considered an active ingredient in cosmetic formulations for cutaneous photoprotection and to prevent the skin aging, supporting the intracellular antioxidant defense mechanism [110,111,112,113].

Apetrei’s group assembled a biosensor for FA detection in cosmetic products, immobilizing TYR onto commercial SPCEs, modified with carbon nanofibers (CNFs) combined with AuNPs (CNF-AuNPs/SPCEs) [114]. TYR was immobilized on the surface of CNF-AuNPs/SPCEs by the drop-and-dry technique. The electrochemical behavior of the biosensor was compared with those of SPCEs modified with only CNFs and with CNFs and AuNPs without TYR, by means of CV.

CNFs present good mechanic and thermal conductivity, large surface area, a high surface-to-volume ratio, low ohmic resistance, and accelerate the electron transfer. On the other hand, AuNPs are biocompatible and increase the nanocomposite conductivity and electrochemical response. TYR, acting as a recognition key, provides better selectivity to the biosensor in a complex matrix with respect to CNF-AuNPs/SPCEs. Under optimized experimental conditions by means of CV, a linear concentration range of 0.1–1.6 μM with an LOD of 2.89 nM was obtained. The sensor resulted in being stable after 50 continuous measurements, resulting in being the corresponding electrochemical signal not significantly affected. Reproducibility (RSD ≤ 2%) and repeatability (RSD ≤ 3%) were addressed, and the results in terms of RSD% were acceptable. The biosensor was found to be selective towards the possible interfering compounds present in cosmetic formulations, such as propandiol, glycerine, and vitamin E, among others.

The biosensor was applied to detect FA in cosmetic products with different compositions and consistencies, such as serums, creams, and emulsions. The data obtained resulted in being comparable with those obtained by means of FT-IR spectroscopy, indicated as the validation method, but no recoveries data were provided.

Stradiotto et al. proposed an electrochemical sensor based on SPCE modified with GO, AuNPs, and a molecularly imprinted polymer (MIP) as the recognition element, for the detection of FA in orange peels [115]. Orange peels contain different phenolic acids, and FA is one of those present in larger amounts. Consequently, food, pharmaceutical, and cosmetics industries consider orange peel to be a cost-effective source of FA, since they employ FA as an important ingredient for several commercial products.

In the proposed electrochemical sensor, the combination of GO and AuNPs produced a nanocomposite with peculiar properties in terms of conductivity, biocompatibility, mechanical and thermal stability, and high surface-to-volume ratio, as already mentioned. The MIP’s role is related to the sensor selectivity, and it is considered a biomimetic material. During the MIP synthesis, a template molecule was enveloped in the polymer structure. At the end of the polymerization process, the template molecule was extracted, and cavities with dimensions, shapes, and orientations corresponding to those of the template were produced [116].

Considering the nanocomposite synthetic procedure, GO and AuNPs were electrodeposited on SPCE in two successive steps. Finally, MIP was prepared using phenol as the monomer and electropolymerization as the synthetic approach in the presence of FA the as template. FA was removed just after the polymerization ended. The nanocomposite was characterized by microscopic, spectroscopic, and electrochemical techniques. AuNPs resulted in being well distributed on the GO nanosheets, and the poly(phenol) polymeric film uniformly covered GO and AuNPs, even if some nanoparticles were still visible after the electropolymerization because the film was very thin.

The sensor parameters were investigated by DPV and two linear ranges of 10 nM–1 μM and 2–10 μM, respectively, were obtained, with an LOD of 3.1 nM. The two linear ranges are probably connected to the different accessibility and position of the imprinted cavities in the polymeric structure.

The inter-day and the intra-day repeatability was tested obtaining an RSD of 4.3% and 3.9%, respectively. These data were considered acceptable. The stability of the sensor was investigated by storing it in air at room temperature for 15 days. After this period, the current response of the electrode presented 92% of its initial value, and the sensor’s long-term stability was considered satisfactory. The sensor selectivity was tested on compounds generally present in orange peels such as gallic acid, caffeic acid, ascorbic acid, and vanillic acid. Results were explained by the different interactions of imprinted cavities with the different interfering molecules, therefore gallic acid, the smallest molecule with many functional groups, represented the major interference for FA detection. The sensor was applied to spiked real samples of orange peels, and the recoveries data were in the range of 99 to 103%.

Lastly, we would like to introduce another hydroxycinnamic acid derivative, i.e., rosmarinic acid (RA). It is an ester of caffeic acid (CA) and 3,4-dihydroxyphenil lactic acid (DHPLA), as shown in Figure 7. RA is found in many plants, herbs, spices, vegetables, and fruits and presents several beneficial properties such as antioxidant, anti-inflammatory, anti-mutagenic, and antitumoral, among others. A recent review reported plenty of examples of electrochemical (bio)sensors for detecting RA [117], but only one concerning AuNPs.

Scheeren’s group described an electrochemical biosensor for RA detection, based on the peroxidase (PER) enzyme, including gold nanoparticles dispersed in 1-butyl-3-methylimidazolium hexafluorophosphate ionic liquid (Au-BMIPF_6_) [118]. Au-BMIPF_6_ were incorporated and cross-linked in chitin (CTN), a polymer of natural origin, to create a supported ionic liquid-phase (SILP) catalyst with high catalytic activity and provide a platform for peroxidase immobilization (Au-BMI-PF_6_-CTN/PER). The enzyme was immobilized by absorption on the biopolymer composite. Au-BMI PF_6_-CTN/PER was incorporated in carbon paste (CP), and the Au-BMI-PF_6_-CTN/PER/CP electrode was assembled. This biosensor was used for RA determination in pharmaceutical samples by SWV. Under optimized conditions, the linear concentration range was 0.50–23.70 μM, with an LOD of 70.09 nM. The biosensor showed good repeatability (RSD 2.7%), reproducibility (RSD 4.9%), and long-term stability (15% decrease in response over 120 days). The method was applied to RA determination in pharmaceutical samples, with recovery values ranging from 98.3 to 106.2%. The effective PER immobilization in the modified CTN matrix, the high conductivity of the ionic liquid, the increase in electron transfer, and consequently the electrocatalytic activity promoted by AuNPs represent the key factors to explain the interesting performances of the Au-BMI PF_6_-CTN/PER/CP biosensor.

#### 3.1.3. Some Considerations on Phenolic Acids (Bio)sensors Based on Au Nanomaterials

As a conclusive comment regarding the reported examples of sensors for the detection of phenolic acids, we can observe that the LODs, independently of the analyte, are generally micromolar to nanomolar for several examples. Noteworthily, the complexity of the material does not always correspond to a better performance of the sensor.

Questionable points are represented by data relating to the selectivity, applicability to real samples, and subsequent validation of the proposed methods; in fact, these issues are not always adequately addressed.

The analytical performances of the reported electrochemical sensors for the determination of phenolic acids as well as the corresponding sensor formats are summarized in Table 1.

The linearity range and LOD data are indicated in the same unit of measurement (μM) to facilitate a comparison among them.

### 3.2. Stilbenes

Resveratrol belongs to the smallest group of phenolic antioxidants, i.e., stilbenes, and is the most well-known and studied for its role in preventing cancer and neurodegenerative and cardiovascular diseases [2,119,120,121]. Resveratrol exists in two isomeric forms as *cis*-resveratrol and *trans*-resveratrol. The *trans* isomer is more abundant and biologically active [121]; for this reason, we indicated the *trans* isomer simply as resveratrol in this review.

Resveratrol (3,5,4′-trihydroxy-trans-stilbene, RES, chemical structure in Figure 11) is composed of two phenolic rings linked by an ethylene bridge. This is a natural polyphenol present in vegetables and fruits, especially in grapes skin and seeds, and in discrete amounts in red wines (see French paradox [119]).

Concerning RES detection, we report three examples of electrochemical sensors including gold nanoparticles and MIPs in the sensing interface.

The first electrochemical sensor was developed using a nanocomposite based on a molecularly imprinted polymer (MIP) and gold and silver hollow nanoshells (Au@Ag) [122]. The nanocomposite (MIP/Au@Ag) was employed to modify an indium tin oxide (ITO) electrode (MIP/Au@Ag/ITO). Au@Ag bimetallic hollow nanoshells were electrodeposited on ITO, and then the MIP film was electropolymerized in situ on the nanoshells, with RES acting as the template molecule and *o*-phenylenediamine as the monomer. RES was removed just after the polymerization ended. The nanocomposite was characterized by microscopic, spectroscopic, and electrochemical techniques. Au@Ag hollow nanoshells increased the electron transfer rate from and to the ITO surface and provided a nanostructurated surface area suitable for MIP assembly. The MIP film covered the Au@Ag hollow nanoshells uniformly, providing specific binding cavities complementarily to RES molecules for both dimension and functionalization, thus acting as the fundamental recognition key element.

Under the optimal experimental conditions and by means of CV, a linear concentration range of 2.0 × 10^−11^–9.0 × 10^−9^ M with an LOD of 7.1 × 10^−12^ M was obtained. The selectivity was evaluated considering the following molecules as possible interfering compounds: Ellagic acid, polydatin, bisphenol A, dopamine, vitamin E, and proanthocyanidins; MIP’s crucial and fundamental role as the recognition element allowed us to define the sensor as selective. The good repeatability and stability can be explained with the mechanical and chemical stability of both Au@Ag bimetallic hollow nanoshells and the MIP film. The sensor was applied to detect RES in spiked real samples of fresh grape seed extracts. Recovery data, ranging from 96.7% to 106.3%, resulted in being comparable to those obtained by HPLC in terms of accuracy and precision.

A G/AuNPs and MIP nanohybrid were used to modify GCE to assemble an electrochemical RES sensor [123]. Firstly, G nanosheets and AuNPs were simultaneously electrodeposited on GCE. The MIP was then synthesized onto the modified electrode via chemical polymerization, with RES acting as the template molecule and acrylamide (AM) as the functional monomer. RES was removed just after polymerization.

G nanosheets and AuNPs increased the electron transfer rate from and to the GCE surface and provided a nanostructurated surface area suitable for MIP assembly. The MIP film covered the G/AuNPs nanosheets uniformly, acting as the fundamental recognition key element, as mentioned above. Under the optimal experimental conditions and by means of CV, a linear concentration range of 0.01–10 μM with an LOD of 0.0044 μM was obtained. No interferences could be observed with the addition of glucose, uric acid, ascorbic acid, dopamine, caffeine, catechin, quercetin, polydatin, bisphenol A, and ellagic acid at the same concentration of RES.

The sensor was stored in air at room temperature when not in use. The current responses to 1.0 μM RES decreased to 91.2% of its initial current after 30 days, indicating good long-term stability. The sensor also showed good repeatability (RSD 4.22%).

The sensor was applied to detect RES in spiked real samples of red wines and grape skins. Recovery data, ranging from 94.5% to 102.5%, were in agreement with those obtained by HPLC.

A molecular imprint-based electrochemical sensor for RES detection was prepared employing a nanocomposite with polyaniline (PANI), AuNPs, and MIP to modify a GCE [124]. AuNPs were electrodeposited on GCE in the first step and PANI was electrosynthesized on AuNPs. Finally, MIP was synthesized onto the modified electrode via chemical polymerization, with RES acting as a template molecule and AM as a functional monomer. RES was removed just after the polymerization. PANI/AuNPs nanocomposite can be assumed to be a conducting network facilitating the electron transfer, and as support for MIP assembly. Under the optimal experimental conditions, a linear concentration range of 1.0 μM–200 μM, with a detection limit of 87 nM, was obtained by DPV. Ferulic acid, catechin, caffeic acid, vanillin, and protocatechuic acid were investigated as possible interferences, and no effects on the RES detection were observed. Repeatability was investigated, with acceptable results in terms of RSD% (4.5%). Concerning the stability, the electrochemical signal remained constant after 24 h in the refrigerator, but additional information is not provided for long-term and/or operational stability. The sensor was applied to detect RES in spiked real samples of red wines. Recovery data, ranging from 97.7% to 108%, resulted in agreement with those obtained by HPLC.

As a final comment, the crucial role of MIPs for sensor selectivity in all the examples reported in this subsection has to be underlined. The analytical performances of the electrochemical sensors reported for the determination of RES as well as the corresponding sensor format are summarized in Table 2. The linearity range and LOD data are indicated in the same unit of measurement (μM) to facilitate comparisons among them.

### 3.3. Flavonoids

Flavonoids represent the largest group of phenolic antioxidants including, 6000 different compounds. They are present in fruits, plants, vegetables, and some beverages (tea, coffee, beer, wine, and fruit juices). They are further classified, depending on their chemical structure and functional moieties, into several subgroups such as flavonols, flavones, isoflavones, flavanones, anthocyanidins, and flavanols (catechins and proanthocyanidins) [3], as illustrated in Figure 12.

The basic chemical structure includes a benzene ring (A) condensed to a six-membered ring (C), linked to a second phenyl ring (B), while hydroxyl groups are present as substituents. Noteworthily, the antioxidant activity is strictly correlated to the number and position of the hydroxyl groups [2,3,125].

A recent review [126] reported several examples of electrochemical sensors and biosensors based on carbon nanomaterials, including carbon nanotubes, graphene, carbon and graphene quantum dots, mesoporous carbon, and carbon black, for the detection of flavonoids in food and pharmaceutical products, evidencing the challenges and future perspectives.

In the present review, we have focused our attention on luteolin as a flavone, on myricetin, quercetin, and rutin as flavonols, and on catechin as a flavanol, because in the literature, examples of (bio)sensors involving gold-based nanomaterials are present only for these flavonoids, as far as we know.

Finally, we introduce phloretin, a naturally occurring dihydrochalcone, present in the leaves, bark, and fruit of apple trees [127]. We remind the reader that chalcones are the biosynthetic precursors of flavonoids [128]. They are chemically classified as α, β-unsaturated ketones with two aromatics rings (A and B), linked by a three-carbon alkenone unit, as shown in Figure 13. Dihydrochalcones, such as phloretin (PH), present two benzene rings linked by a three-carbon alkanone unit.

Phloretin (chemical structure in Figure 14) is characterized by antibacterial, antiviral, antifungal, anti-inflammatory, antioxidant, and antitumor activities [127].

We would like to mention, at this point, an example that describes how an electrochemical cell-based biosensor can become a useful tool for the evaluation of the antioxidant capacity of a compound such as phloretin by means of electrochemical impedance spectroscopy (EIS).

A novel cell-based electrochemical biosensor was assembled to propose an oxidative damage model and evaluate the antioxidant capacity of PH on a 3D cell culture [129], as illustrated in Figure 15.

In particular, human lung adenocarcinoma epithelial cells (A549) were encapsulated in alginate gel and then immobilized onto a self-assembled L-cysteine/gold nanoparticle (AuNPs/L-Cys) nanocomposite. Finally, the nanocomposite was used to modify a glassy carbon electrode by drop-casting.

AuNPs are biocompatible and, as usual, accelerate the electron transfer rate, possessing different properties at the sensing interface such as its electrical conductivity, chemical stability, and electrocatalytic properties. In addition, L-cysteine provides several appropriate sites for immobilizing the A549-cells.

EIS data indicated an increase in the impedance value with the concentration of H_2_O_2_, selected as an oxidative stress inducer model. However, the EIS value decreased with the co-incubation of PH ranging from 10 to 100 μM, indicating that the EIS response is affected by the antioxidant concentration and by the incubation time. PH linear concentrations ranged from 20 μM to 100 μM with a detection limit of 1.96 μM. The biosensor has been applied to evaluate the PH antioxidant capacity in real cells with results comparable to those obtained with a conventional ROS assay, indicating that the biosensor resulted in a promising tool for antioxidant capacity determination.

#### 3.3.1. Luteolin

Luteolin (3,4,5,7-tetrahydroxy flavone, LUT, chemical structure in Figure 16) is a flavonoid present in various plants, fruits, and spices. It displays antioxidant, antibacterial, antiviral, anti-inflammatory, and antitumoral properties [130,131].

A nanocomposite including AuNPs and boron nitride nanosheets (BNNs) was used to modify GCE and assemble an electrochemical sensor for LUT detection [132]. The nanocomposite (BNNs/AuNPs) was synthesized by means of a simple two-step microwave approach. According to the morphological characterization, AuNPs were well-distributed without agglomeration and/or aggregation on BNNs, representing the active sites for LUT oxidation. BNNs can be assumed to be an ultrathin 2D nanomaterial showing a graphene-like layered structure and properties [133]. Two linear detection ranges from 5 to 1200 pM and 0.02 to 10 μM with a detection limit of 1.7 pM were obtained, using SWV to determine LUT. Operational stability was tested by twenty successive measurements using a luteolin concentration of 1 μM. After 20 measurements, the electrochemical response decreased to 81.7% of the initial value and was considered satisfactory. The proposed sensor also showed acceptable reproducibility with an RSD of 3.6%.

The selectivity of BNNs-AuNPs/GCE was tested towards the most common anion and cation and ascorbic acid, uric acid, lysine, glucose, and citric acid, and no detectable interference was evidenced. Finally, LUT was determined in peanut hull and Perilla spiked real samples with recoveries ranging from 98.3 to 103.38%. Results were almost comparable to those obtained by HPLC; the discrepancies were probably due to the extraction process and the different detection procedure that was optimized.

Chen’s group [134] developed an LUT sensor using a particular electrode, i.e., a gold nanocages (AuNCs)-modified carbon ionic liquid electrode (CILE). It was assembled involving 1-hexylpyridinium hexafluorophosphate (IL) as the binder, and then AuNCs were deposited on its surface, resulting in AuNCs-modified CILE (AuNCs/CILE).

From the nanocomposite morphological characterization, the CILE surface was uniform without separated carbon layers, and AuNCs resulted in being well-disseminated on it with well-defined hollow shapes. The AuNCs enhanced the electrodic conductivity and acted as accelerators of the electron transfer. The electrode was electrochemically characterized, and the redox behavior of LUT was studied. The sensor’s analytical performances were evaluated by DPV, and a linear concentration range of 1–1000 nm and an LOD of 0.4 nM were achieved. Reproducibility was analyzed, with acceptable results in terms of RSD% (2.4%). The long-term stability (3.48% decrease in response after 4 weeks at 4 °C in the refrigerator) was considered good. The sensor selectivity was examined considering the most common cations, lysine, threonine, glycine, alanine, and two flavonoids with chemical structures similar to that of LUT, i.e., baicalein and quercetin. All these ions and compounds could be present in biological and drug samples, but no detectable interference was evidenced.

Finally, AuNCs/CILE was applied to determine LUT in drug-spiked real samples, with good results in terms of recoveries, ranging from 95.0% to 96.7%.

A nanocomposite including MWCNTs, PEDOT, and AuNPs was prepared by means of a one-pot method and used to modify GCE to assemble an LUT sensor [135], where the monomer of the polymer was the reducing agent for the AuNPs precursor. In addition, as evidenced from the morphological analysis, PEDOT and MWCNTs were interconnected and networked to effectively stabilize AuNPs. As usual for this kind of nanocomposite, the carbon nanotubes, AuNPs, and conductive polymer altogether contribute to enhancing the electrode’s conductivity, therefore speeding up the electron transfer at the electrode surface. After the electrochemical characterization, the analytical performances were evaluated by SWV. Two linearity ranges of 0.001–0.1 mM and 0.1–15 mM were observed with a detection limit of 0.22 μM. The long-term stability was tested, and after 30 days at 5 °C in the refrigerator, the electrochemical signal decreased to 91.35% of its initial value. Reproducibility was analyzed evidencing acceptable results in terms of RSD% (1.94%). The inter-day and the intra-day repeatability was tested, obtaining an RSD of 4.3% and 1.4%, respectively. These data were considered good. The sensor selectivity was examined considering the most common salts and compounds present in drugs and biological samples, such as CaCl_2_, MgSO_4_, glucose, maltose, and curcumin, among others. In addition, flavonoids with similar chemical structures, such as quercetin, rutin, myricitrin, and diosmetin, were included, but no clear detectable interference was evidenced. Finally, the sensor was applied to detect LUT in human serum-spiked real samples, with good results in terms of recoveries ranging from 103.1 to 99.63%.

Biomass-derived porous carbon (BPC) sourced from banyan leaves was prepared by the calcination technique, obtaining three-dimensional interconnected frameworks. Furthermore, Au nanoflakes (AuNFs) were synthesized on BPC by the hydrothermal method to obtain AuNFs-BPC nanocomposite to modify GCE to determine LUT in pharmaceuticals products and human urine samples [136]. A scheme including the synthetic procedure and electrochemical sensor application is presented in Figure 17.

BPC shows good electrical conductivity and high chemical stability, as all the other carbon-based materials, but the low price, large specific surface area, tailored pore structure, modulating pore size, etc., make it particularly attractive for application in different fields including the sensing area [136]. BPC showed a three-dimensional porous structure and AuNFs uniformly covered the BCP surface, increasing the nanocomposite’s conductivity.

The optimized analytical parameters were evaluated by DPV: Two linearity ranges from 0.15 to 1.8 μM and 1.8 to 10.0 μM were calculated with a detection limit of 0.07 μM. Selectivity was investigated considering inorganic ions, amino acids, and glucose, and no clear interference was detected. Good results in terms of RSD% were obtained for the sensor’s reproducibility (3.54%). Concerning stability, interesting results were acquired. In fact, after one-week storage at 4 °C in the refrigerator, the electrochemical signal only decreased to 96.68% of its initial value, and after 100 cycles of continuous scanning by CV, the electrochemical signal only decreased to 98.66%.

Finally, the sensor was applied to determine LUT in drug- and human-urine-spiked samples, with acceptable recovery values, ranging from 97.00 to 105.00% for the drug and from 98.5 to 106.17% for urine.

The combination of AuNPs and G or its derivatives was the basis of the nanocomposites used to assemble some interesting examples of sensors for the determination of LUT.

Wu et al. prepared a mercapto-β-cyclodextrin/graphene nanosheets/AuNPs nanocomposite (SH-β-CD-GNs/AuNPs) to assemble an electrochemical sensor on a GCE for the detection of LUT in human serum [137]. SH-β-CD was selected for the nanohybrid synthesis using a one-pot method without the presence of additional surfactants.

As usual, the integration of GNs and AuNPs provided the nanohybrid with enhanced electrical conductivity and a higher active surface area. SH-β-CD can not only act as a dispersing agent and linker between AuNPs and GNs thanks to the SH functional group, but it can also accumulate more LUT on the electrode surface by host–guest interaction. Under optimized conditions, the analytical parameters were evaluated by DPV, and a linear concentration range of 10.0 pM to 10.0 μM and an LOD of 3.3 pM were achieved. In addition to the acceptable stability and reproducibility data, the sensor’s selectivity was analyzed. The presence of oxalic acid, cysteine, mannitol, urea, ascorbic acid, citric acid, glucose, naringenin, and rutin did not produce detectable interference. On the other hand, for isoquercetin, quercetin, and baicalein, an interference response less than 10% has been evidenced. SH-β-CD-GNs/AuNPs/GCE was applied to detect LUT in human-serum-spiked real samples, with good results in terms of recoveries, ranging from 97.1 to 103.2%.

A composite including AuNPs and three-dimensional graphene (3DG) was electrodeposited on CILE to assemble an electrochemical sensor for LUT [138].

CILE was prepared by using ionic liquid 1-hexylpyridinium hexafluorophosphate, as also reported for another LUT sensor (see Ref. [134]). The syntheses of 3DG and AuNPs were performed electrochemically in two successive steps. AuNPs resulted in being well and densely distributed on the 3DG structure. The sensor’s analytical performances were evaluated by DPV: A linear concentration range of 0.05–50.0 μM and an LOD of 7.59 nm were observed. The stability (10% decrease in response after 10 days at room temperature) and reproducibility (RSD 3.0%) were acceptable; the selectivity was analyzed, considering only amino acids and inorganic cations as potential interferences, and no clear interference was detected. The sensor was applied to detect LUT in spiked real samples of pharmaceutical products with recoveries ranging from 95.28 to 103.77%.

A nanocomposite involving AuNPs and GQDs was employed to modify GCE to assemble an electrochemical sensor for LUT detection [139]. Firstly, AuNPs and then GQDs were electrodeposited on the electrode in two successive steps. As a result, the highly active surface and conductivity of GQDs and the electrocatalytic properties of AuNPs were merged. Using DPV and under-optimized conditions, a linearity range of 1 × 10^−8^ to 1 × 10^−5^ M with a detection limit of 1.0 nM was obtained. According to the RSD% values, repeatability (RSD 2.6%) and reproducibility (RSD 4.3%) can be evaluated as acceptable. Concerning stability, after two weeks at 4 °C, the response decrease was only 6.9% of its initial value.

The selectivity data indicated that ascorbic acid, oxalic acid, cysteine, citric acid, glucose, and flavonoids with similar chemical structures, i.e., baicalein, naringenin, and isoquercitrin, had no clear interference.

GQDs/AuNPs/GCE was applied to detect LUT in peanut-hull-spiked real samples. Recoveries were in the range of 98.8–101.4%, and the RSD values were <2%.

#### 3.3.2. Myricetin

Myricetin (3,5,7,3′,4′,5′-hexahydroxyflavonol, MYR, chemical structure in Figure 18) is a polyhydroxyflavonol, present in natural plants, berries, other fruits, vegetables, honey, red wine, tea, and other food products [140].

It has been demonstrated [140] that MYR displays several pharmacological properties such as anti-inflammatory, antitumor, antibacterial, antiviral, and antioxidant. A recent review reported pharmacological studies on MYR published in the last five years to support the development of new drugs based on MYR for clinical use [140].

Noteworthily, very few papers on the determination of MYR using electrochemical methods are reported in the literature.

Among these, we point out three examples of electrochemical sensors for the determination of MYR including AuNPs.

Shams and co-workers prepared an electrochemical sensor for MYR detection using a nanocomposite based on MWCNTs, ethylenediamine (en), and AuNPs [141]. MWCNTs were deposited on GCE and en was electrochemically grafted to them. Finally, AuNPs, synthesized separately, were assembled via electrostatic interactions on the en layers electrografted to the carbon nanotubes network. The modified electrode was electrochemically characterized and used to determine MYR by cyclic voltammetry adsorptive stripping voltammetry (CVAdSV).

Under the optimized conditions, a linear concentration range of 5.0 × 10^−8^ M to 4.0 ×10^−5^ M and an LOD of 1.20 × 10^−8^ M were achieved. Intra-day repeatability (RSD 3.4%) and reproducibility (RSD 3.6%) resulted in being acceptable. Stability data were defined as satisfactory, but no result was provided. Selectivity was tested employing ascorbic acid, phenol, isoflavones, polysaccharides, amino acids, and inorganic salts, which are present in nutritious foods, supplementary drugs, fruit juices, and plant extracts. The sensor can be considered selective on the basis of the collected data. MWCNTs/en/AuNPs/GCE was applied to determine MYR in spiked real samples of black and green teas and in fruit juices, with interesting recoveries, ranging from 101–104%.

The next example [142] reported an electrochemical sensor for MYR employing a graphite electrode modified with AuNPs. The graphite electrode was fabricated through a green and low-cost process starting from exhausted batteries materials. AuNPs were electrodeposited on the electrode surface and the electrochemical parameters modulation allowed for tailoring of the particles’ dimension and shape. The modified electrode was electrochemically characterized, and MYR was determined by SWV. A linearity range of 2–10 μg mL^−1^ and an LOD of 0.4 μg mL^−1^ were observed. Stability, selectivity, repeatability, and reproducibility were missing, as well as applications to real samples.

A more recent electrochemical sensor for MYR detection was assembled based on a nanocomposite including POM clusters [100,101], mesoporous SnO_2_ nanospheres, and AuNPs [143]. The mesoporous SnO_2_ nanospheres were synthesized and non-covalently functionalized with poly dimethyl diallyl ammonium chloride (PDDA), a cationic polyelectrolyte, to increase their solubility. The ternary nanohybrid was used to modify an ITO electrode by means of the layer-by-layer (LbL) self-assembly technique. POM clusters (in this case, K_6_P_2_W_18_O_62_ briefly called P_2_W_18_) served to improve the electrochemical performance of MYR; the mesoporous SnO_2_ nanospheres together with AuNPs served to enhance the conductivity and electrocatalytic activity. An electroxidation mechanism of MYR at the (P_2_W_18_-SnO_2_-AuNPs)_3_/ITO electrode is illustrated in Figure 19.

The ITO-modified electrode was electrochemically characterized and used to determine MYR by amperometry, obtaining a linearity range of 1–110 μM and a detection limit of 67 nM.

No detectable amperometric responses were observed from glucose and inorganic salts. Reproducibility RSD 4.5%), operational stability (3.6% decrease in response after 20 successive scanning of CV), and storage stability (10% decrease in response after a week in air at room temperature) were considered interesting results. The applicability to real samples of fruit juices was investigated, and the recoveries, ranging from 97.65% to 103.07%, seem promising.

#### 3.3.3. Quercetin

Quercetin (3,3′,4′,5,7-pentahydroxyflavone, QR, chemical structure in Figure 20) is a flavonoid present in several fruits, plants, flowers, and vegetables, and its most relevant property is its well-known antioxidant activity.

In addition, QR displays anti-cancer activity by binding to cellular receptors and proteins. Furthermore, QR supports the effects of certain chemotherapeutic treatment, as recently reported in the literature [144,145].

In the literature, there are many examples of electrochemical sensors for the determination of QR, evidencing the importance of this flavonoid both as an antioxidant and as assisting in anticancer therapies. Herein, we reported some significant examples involving gold nanomaterials.

An electrochemical sensor for QR was developed using a AuNPs/MWCNTs nanocomposite. AuNPs were deposited onto *p*-aminothiophenol-functionalized MWCNT layers via self-assembling [146]. The MWCNT’s electrical conductivity, chemical stability, and mechanical strength integrated the AuNP’s conductivity and electrocatalytic activity in a proper manner. The *p*-aminothiophenol functionalization served to increase the nanocomposite stability via the nanoparticles self-assembling. A nanocomposite suspension was dropped on GCE to modify it. After its electrochemical characterization, the analytical parameters were optimized by SWV to detect QR and its derivative rutin (RT).

The corresponding linearity range was 1.0 × 10^−9^–5.0 × 10^−8^ M with an LOD of 3.3 × 10^−10^ M for both QR and RT. Stability (a 1.5% decrease in response after one month) and reproducibility (RSD1.6 %) were evaluated, and the results can be considered promising. The proposed method allowed for the simultaneous determination of RT and QR, but the selectivity with respect to other possible interfering compounds was not analyzed. AuNPs/MWCNTs/GCE was applied to determine QR and RT in fruit juices, and the results were comparable to those from LC-MS analysis, with recoveries ranging from 96.3 to 99.4%.

Lin’s group proposed an electrochemical sensor for QR determination using a nanocomposite with AuNPs and gelatin (GEL) as “ingredients”, synthesized by the one-pot photochemical method [147]. GEL is a denatured animal protein collagen with excellent film-forming capabilities. In this example, GEL acted for AuNPs as chitosan in other mentioned examples [80,81] under UV-light radiation. The resulting AuNPs were well distributed in the nanohybrid film structure.

The QR sensor was assembled by dropping a nanocomposite suspension on an SPCE. The electrochemical behavior of QR was investigated, and the analytical parameters were optimized by DPV. The linear response ranged from 0.02 to 34.5 μM with a detection limit of 0.0019 μM. The selectivity of the QR sensor was verified by DPV in the presence of possible interferences such as ascorbic acid, dopamine, acetaminophen, rutin, morin, glycine, glucose, and some inorganic salts. No significant changes in the DPV response in the presence of all the interfering substances were evidenced. The storage stability (2.1% decrease in response after 20 days at 4 °C in the refrigerator) and reproducibility (RSD 2.9%) were investigated, obtaining interesting results. GEL/AuNPs/SPCE was applied to detect QR in spiked onion extracts and apple samples, with recoveries ranging from 99.2 to 99.9%, but no comparison with an external reference method was provided.

Kawde and coworkers proposed an interesting application of a nanocomposite, including Au and CeO_2_ nanoparticles and functionalized glassy carbon microspheres (FGCMs), for the determination of QR and to detect QR interactions with DNA [148]. FGCMs were obtained via acidic treatment of commercial glassy carbon microspheres (GCMs), exactly as the functionalization of CNTs is performed [148]. GCMs are well known because of their conductivity, mechanical stability, high surface-to-volume ratio, porosity, and easy functionalization [149]. The integration of CeO_2_NPs and AuNPs with FGCMs produced a nanocomposite with peculiar and merged mechanical and electrochemical properties with respect to the single starting material. The nanocomposite morphology indicated that the nanoparticles were uniformly and well-distributed on the microspheres.

An Au/CeO_2_@FGCM–paraffin oil paste electrode (PE) (Au/CeO_2_@FGCM-PE) was then assembled by mixing the appropriate amount of the nanocomposite with paraffin oil, and electrochemically characterized. The analytical parameters were optimized by SWV, and a linear response from 4.8 × 10^−8^ to 1.09 × 10^−6^ M and a detection limit of 3.7 × 10^−10^ M were achieved. Satisfactory results were obtained from the investigation of stability, reproducibility, and repeatability. Many compounds were considered for selectivity, in particular uric acid, ascorbic acid, aspartic acid, glutamic acid, glucose, histidine, methionine, tryptophan, inorganic salts, and flavonoids such as RT. No clear detectable interference was evidenced, except for rutin and ascorbic acid. With regard to RT, concentrations (≤10-fold with respect to the QR concentration) did not allow interference to be detected, while for ascorbic acid, it should be noted that the problem was not fully solved. Au/CeO_2_@FGCM-PE was applied to determine QR in spiked real samples of apple juice, grape juice, green tea, honeysuckle, and onion. Results were comparable to those from external reference methods (HPLC and spectrophotometry), with recoveries ranging from 97.8 to 102%.

An interesting application of this sensor was the possibility to determine ds-DNA using the interaction of DNA with QR. In the presence of ds-DNA, the electrochemical signal of QR decreased linearly as the amount of DNA in the solution increased, due to the formation of a slowly diffusing QR-ds-DNA complex. It was hypothesized that the interaction between DNA and QR can involve an intercalation, according to the literature findings [150].

Mostafavi and coworkers reported an electrochemical sensor for QR detection based on a nanocomposite, including Fe_3_O_4_@SiO_2_ and Au nanoparticles and PANI (Fe_3_O_4_@SiO_2_/AuNPs/PANI) [151]. Finally, the nanocomposite was casted on GCE to assemble the sensor. Fe_3_O_4_ NPs were coated with inert silica (SiO_2_) nanoparticles to prevent their aggregation and increase their chemical stability (Fe_3_O_4_@SiO_2_). The integration of Fe_3_O_4_@SiO_2_ with PANI produced a conducting polymer with a more efficient pathway of electron transfer with respect to the polymer alone, according to the literature [152]. Moreover, as usual, the successive combination with AuNPs imparted the final nanocomposite with biocompatibility and electrocatalytic properties. The Fe_3_O_4_@SiO_2_/AuNPs/PANI synthesis involved several successive steps, from the synthesis of Fe_3_O_4_ to that of AuNPs, as the final step.

Fe_3_O_4_@SiO_2_/AuNPs/PANI/GCE was electrochemically characterized, and its analytical parameters were optimized by DPV. A linear response was evidenced from 1.0 × 10^−8^ to 1.5 × 10^−5^ M with an LOD of 3.8 × 10^−9^ M. Reproducibility was analyzed, with satisfactory results in terms of RSD% (5.1%). Concerning selectivity, glucose, uric acid, caffeine, and ascorbic acid were tested as potential interfering compounds. In this regard, since the tolerance limit was defined as the maximum concentration of interference that caused a corresponding electrochemical change in terms of a relative error of ± 5%, the results showed that no detectable interference was observed. Spiked real samples of human serum and urine, tea, radish leaves, and apple juice were analyzed with Fe_3_O_4_@SiO_2_/AuNPs/PANI/GCE with recoveries from 95.6 to 101.16%.

A structurally complex nanohybrid based on a Au-Co nanoparticle-embedded N-doped carbon nanotube hollow polyhedron was employed to develop an electrochemical sensor for QR by modifying GCE [153].

First, we have to consider the genesis of the nanocomposite. A core-shell ZIF-8@ZIF-67 was pyrolyzed to prepare a Co nanoparticle-embedded N-doped carbon nanotube hollow polyhedron (Co@NCNHP). Then, AuNPs were synthesized via chemical reduction on the surface of Co@NCNHP to obtain an Au-Co bimetal-decorated NCNHP (Au-Co@NCNHP). ZIF-8 and ZIF-67 are metal organic frameworks (MOFs), classified as zeolitic imidazolate frameworks (ZIFs). MOFs are defined as porous coordination polymers, constituted by organic linkers and metal ions or clusters [106,154]. Among them, ZIFs are characterized by coordination bonds between transition metal ions and organic ligands, large surface area, high nitrogen content, and porosity.

ZIF-8 ([Zn (2-mIm)_2_]_n_ and ZIF-67 ([Co (2-mIm)_2_]_n_) included the same zeolitic structure and the same organic ligand, i.e., 2-methylimidazole (2-mIm). Exploiting these features, a core-shell nanostructure was obtained from ZIF-8 and ZIF-67 (Zif-8@ZIF-67) and it was used to produce a Co nanoparticle-embedded N-doped carbon nanotube hollow polyhedron (Co@NCNHP). During the pyrolysis, the produced gas assisted in Zn^2+^ removal and promoted the formation of the hollow carbon polyhedron. After the subsequent chemical reduction, AuNPs were deposited on the hollow carbon structure and Au-Co@NCNHP was obtained. The hollow carbon and porous structure can promote analyte diffusion to the electrode surface and the AuNPs can enhance the electrocatalytic activity.

The GCE modification was performed by drop-casting the nanocomposite suspension on the electrodic surface. The modified electrode was electrochemically characterized and used for determining QR by DPV. A linear response of 0.050 to 35.00 μM and a detection limit of 0.023 μM were achieved. To test the sensor’s selectivity, the following potential interfering compounds were evaluated: Glucose, glycine, citric acid, ascorbic acid, uric acid, dopamine, rutin, luteolin, baicalein, and inorganic salts. The electrochemical response of QR showed no relevant interference from the above-mentioned compounds. Repeatability and storage stability were analyzed, with acceptable results. Finally, the sensor was applied to spiked real samples of onion and drugs, with recoveries ranging from 97.96 to 102.5%. Results were comparable with those obtained by HPLC, assumed as the external reference method.

The next two sensors involved porous gold films.

Zhao’s group developed a QR electrochemical sensor based on a three-dimensional, highly porous gold film (hp-Au), used to modify a gold electrode [155]. The porous film was prepared through a two-step, self-templating method: The first step consisted of electrodeposition of the gold template, followed by the electrodeposition of the final porous layer. The Hp-Au film showed a three-dimensional dendritic nanostructure. The hp-Au conductivity enhanced the electrodic active surface area and promoted the electron transfer from and to the Au electrode surface.

The modified Au electrode was electrochemically investigated and used to detect QR by amperometry. A linear response in the concentration range of 20 nM to 100 μM and an LOD of 3.9 nM were observed. Common amino acids, biological molecules, inorganic salts, and some phenols were considered as possible electroactive interferences, but they did not significantly affect the QR electrochemical response. Repeatability (RSD 0.25%), reproducibility (RSD 0.20%), and stability (a 4.70% decrease in the response after 7 days at room temperature) could be considered satisfactory.

The sensor’s applicability to real samples was examined. Spiked real samples of green tea, honeysuckle, commercial apple juice, onion, and pharmaceutical products were analyzed, obtaining recoveries ranging from 97.43% to 101.59%, but no external reference method was considered.

In the second example, a nanoporous gold film was electrochemically prepared, including the anodizing of a gold layer in the presence of an appropriate anionic surfactant and the reduction of the gold oxide formed in the first step, using ascorbic acid as a reducing agent [156]. The anionic surfactant assisted the formation of a gold oxide layer during the anodizing step because, being negatively charged, it could interact electrostatically with the positively charged gold electrode surface. The resulting nanoporous electrode, due to its particular nanostructure, showed enhanced electrocatalytic activity. It was electrochemically investigated and used to determine QR by means of DPV. Two linear concentration ranges were observed: The first was between 0.01 and 12.0 μM and the second was between 12.0 and 100.0 μM, with a detection limit of 1.1 nM. Caffeine, ascorbic acid, citric acid, glycine, L-cysteine, glucose, fructose, and some inorganic salts were assumed to be possible interferences, and no clear interferences were noted. Reproducibility tests (RSD 1.8%) reported satisfactory results, but no repeatability, stability, or real sample applicability data were provided.

The following examples reported nanocomposites including gold nanomaterials, generally AuNPs and G or its derivatives such GO, GQDs, and so on.

The first example is an electrochemical sensor based on a composite including graphene functionalized with *p*-aminothiophenol and AuNPs to determine QR in the presence of ascorbic acid [157]. The *p*-aminothiophenol functionalization served to increase the nanocomposite’s stability via the nanoparticle’s self-assembly, as already mentioned for a previous electrochemical sensor for QR [146] where MWCNTs were functionalized with the same thiophenol.

As previously reported for MWCNTs, the electrical, chemical, and mechanical properties of G were integrated with the conductivity and electrocatalytic activity of AuNPs in the proper manner. A nanocomposite suspension (AuNPs/G) was dropped on GCE to modify it. After electrochemical characterization, the sensor was employed to detect QR. A linear response in the range of 1.0 × 10^−12^–10 × 10^−12^ M and a detection limit of 3.0 × 10^−12^ M were achieved. Stability (a 3.63% decrease in electrochemical signal after one month) and reproducibility (RSD 1.12%) were considered acceptable.

Selectivity was tested towards ascorbic acid, as the principal interference in the pharmaceutical product analysis. It showed no clear interference in the determination of QR due to their different electrochemical oxidation potentials. The electrochemical sensor was applied to determine QR in real pharmaceutical preparations and the results were comparable with those indicated by the manufacturer.

Harrington and co-workers [158] developed an electrochemical sensor based on GCE modified through a nanohybrid including AuNPs and GQDs.

AuNPs and GQDs were synthesized in two successive steps, then AuNPs were casted on the GCE surface and, subsequently, GQDs were added. In the resulting nanocomposite, the GQDs’ high electrical conductivity and mechanical stability and the AuNPs’ electrocatalytic activity were merged. A scheme of the sensor’s assembly and electrochemical detection of QR is reported in Figure 21.

AuNPs/GQDs/GCE was used to determine QR by DPV. The linear concentration range resulted in being 0.01–6.0 μM with a detection limit of 2.0 nM. Ascorbic acid, L-cysteine, L-phenylalanine, L-tryptophan, inorganic salts, and flavonoids such as apigenin, puerarin, naringenin, luteolin, and rutin were considered as potential interferences. Considering a tolerance limit with an error of ±5%, all these compounds did not represent clear interference for QR determination. After achieving acceptable results for stability (a 5.3% decrease in response after one week at 4 °C) and reproducibility (RSD 5.8%), the sensor was applied to spiked real samples of human serum. The recoveries ranged between 95 and 104.6%, but no external reference method was indicated.

The next two sensor examples involved mercapto-β-cyclodextrin (SH-β-CD) and AuNPs [159,160].

Fei’s group developed these two sensors for the detection of QR. In the first case, they used, in addition to cyclodextrin and AuNPs, rGO functionalized with 1-pyrenebutyrate (PB-rGO) and in the second case, aminated graphene quantum dots (NH_2_-GQDs) were used as the carbon nanomaterial component. As reported in Section 3.3.1, a similar nanocomposite was employed for the detection of LUT [137], including SH-β-CD, AuNPs, and G, with promising analytical performances.

Starting from [159], the functionalization of rGO with 1-pyrenebutyrate promoted rGO dispersion in water, preventing the irreversible agglomeration of the carbon nanomaterial and improving both the film-forming and conductivity properties of the final nanocomposite. SH-β-CD acted as a dispersing agent for the graphene component, as illustrated in [137], due to the interaction between non-covalent bonds and π–π stacking with the graphene layers. As usual, the integration of rGO and AuNPs provided the nanohybrid with enhanced electrical conductivity and s higher active surface area. A glassy carbon electrode was modified by dropping the PB-rGO/TCD/AuNPs nanohybrid on the electrodic surface. The electrode was characterized with different electrochemical techniques and used to determine QR by DPV. Under optimized experimental conditions, the linear response range for QR was 0.005 to 0.4 μM with a detection limit of 1.83 nM.

Stability (a 7.2% decrease in response after two weeks at room temperature) and reproducibility (RSD 3.1%) were monitored, achieving interesting results. Inorganic salts, glucose, dopamine, and some antioxidants such as baicalein, galangin, resveratrol, citric acid, ascorbic acid, and morin were selected as possible interferences. They did not affect the QR determination, except morin. This issue was further investigated in [160]. The sensor was applied to spiked real samples of apple juice, red wine, and honeysuckle, and the recoveries range was 95.0–104.3%, but no external reference method was used for comparison.

In the second case [160], rGO was substituted by NH_2_-GQDs. As reported in Section 2.4.2, this represented a step forward with respect to graphene itself. In addition, functionalization with a NH_2_ group supported the dispersion of GQDs, avoiding their agglomeration. GCE was modified by dropping a suspension of NH_2_-GQDs/AuNPs-β-CD on its surface and was then characterized by means of different electrochemical techniques.

The determination of QR was performed by DPV, and a linearity range of 0.001–0.21 μM and LOD of 285 pM were achieved. It has to be underlined that the substitution of rGO with GQDs resulted in a 10-fold lower LOD (0.285 nM vs. 1.87 nM). Stability (a 4.8% decrease in signal after two weeks at 4 °C) and reproducibility (RSD 3.22%) were investigated, achieving promising results. Several interferences were considered, such as inorganic salts, organic compounds, and biological molecules, and they did not affect the QR response under a tolerance limit of 5%. Morin represented interference for the detection of QR trace amounts, and the use of the standard addition method in the presence of QR traces and high concentrations of morin was strongly recommended. NH_2_-GQDs/Au-β-CD/GCE was employed to test QR in real spiked samples of honey, tea, honeysuckle, and human serum, and the obtained recoveries were in the range of 106.2–97.5%, but in this case, no external reference method data were provided for comparison.

The last example is a sensor based on a nanocomposite including GQDs and AuNPs [161]. GQDs were electrodeposited on SPCE to achieve better stability and reproducibility, while AuNPs were synthesized separately by chemical reduction and then drop-casted on GQDs/SPCE. It is evidenced that the sensor was connected to a portable and miniaturized electrochemical instrument to evaluate the possibility of performing analytical determination on-site using ad hoc miniaturized instrumentation.

On the other hand, material characterization was not reported, so no considerations of the nanocomposite morphology and structure were discussed. Accurate electrochemical characterization was reported and the QR determination was carried out by SWV. A linear response ranging from 1.0 × 10^−10^ to 1.0 × 10^−3^ M and LOD of 3.3 × 10^−11^ M were obtained. Considering that SPCEs are disposable, acceptable results were achieved regarding repeatability (inter-day repeatability RSD 5.10% and intra-day repeatability RSD 5.30%) and stability (a 5.4% decrease in signal after two days).

The sensor’s selectivity was addressed considering several organic compounds, such as glucose, sucrose, ascorbic acid, riboflavin, phenylalanine, L-tryptophan, L-tyrosine, bisphenol A, lysine, uric acid, and inorganic salts. Considering a tolerance limit involving a relative error of ±5%, no clear interference was evidenced at the interference concentrations used. The sensor was applied to determine QR in human serum droplets with recoveries ranging from 92.6 to 101.7%, but no external reference method data were provided or compared.

#### 3.3.4. Rutin

Rutin (3,3′,4′,5,7-pentahydroxyflavone-3-rhamnoglucoside, RT, chemical structure in Figure 22) is a very important and active flavonoid, commonly present in plants, vegetables, fruits, and food products. It is also known as vitamin P and is a quercetin derivative as quercetin-3-O-rutinoside.

Due to its antioxidant, anti-inflammatory, antiviral, and antihypertensive properties, it has attracted increasing attention and interest. Consequently, smart and reliable methods have become very important for its determination [162].

Herein, we report some recent examples of electrochemical sensors based on gold nanomaterials applied for RT detection.

Cruz Vieira and co-workers developed an RT electrochemical biosensor based on a CPE modified with a nanocomposite including AuNPs synthesized and dispersed in β-cyclodextrin (AuNPs-CD), where the enzyme laccase (LAC) was immobilized [163]. LAC acted as the key recognition element and was effectively immobilized through electrostatic interactions with the positively charged AuNPs-CD nanohybrid, given that LAC is negatively charged under immobilization conditions. The presence of AuNPs and the efficient enzymatic immobilization improved the electronic transfer between LAC electroactive sites and RT.

After the nanohybrid’s morphological and structural characterization, the resulting AuNPs-CD/LAC/CPE was electrochemically studied via different techniques. The RT determination was carried out by SWV, obtaining a linearity range of 0.30 to 2.97 μM and an LOD of 0.14 μM. The biosensor stability (a 5.0% decrease in signal after 180 days at room temperature) and repeatability (RSD 6.0%) were analyzed, with satisfactory results. The Lanette cream base and starch with aerosol were investigated as potential interfering excipients, because of their presence in different pharmaceuticals formulation. Considering a tolerance limit involving a relative error of ± 5%, no clear interference was evidenced. The biosensor was applied to determine RT in different pharmaceutical formulations such as capsules and a dermatological cream, with recoveries ranging from 96.8 to 103.8%. The results were compared with those obtained with the spectrophotometric standard method and resulted in good agreement.

An interesting electrochemical sensor for RT was developed using an AuNPs-decorated helical carbon nanotubes (AuNPs-HCNTs) nanocomposite [164]. Helical carbon nanotubes (HCNTs) showed a peculiar 3D-helical structure, so they were widely used in different application fields such as micromagnetic sensors, among others. They also demonstrated peroxidase-like catalytic activity and electrocatalytic activity [165]. The AuNPs-HCNTs nanocomposite integrated the biocompatibility and electrocatalytic activity of AuNPs with the conductivity and mechanical stability of the particular 3D-helical structure of HCNTs. The carbon nanotubes were functionalized with poly(diallyldimethylammonium chloride) (PDDA) to interact with the negatively charged AuNPs through electrostatic interactions and immobilizing AuNPs on PDDA-HCNTs. Finally, the AuNPs-HCNTs nanohybrid was casted on glassy carbon electrode to obtain AuNPs-HCNTs/GCE.

After material characterization, the modified electrode was electrochemically investigated by CV. The RT detection was amperometrically performed, and a linear concentration range of 0.1–30 μM with a detection limit of 0.081 μM was obtained. Glucose, sucrose, uric acid, and ethanol, together with several inorganic salts, were included as possible interfering compounds and excipients. Considering a tolerance limit involving a relative error of ±5%, no clear interference was evidenced. Stability (a 7.0% decrease in signal after one week at room temperature) and repeatability (RSD 2.1%) investigations showed acceptable results.

Finally, the sensor was applied to determine RT in pharmaceutical products such as tablets, with recoveries ranging from 99.6 to 102.1%, but no comparison with a standard method was provided.

An electrochemical sensor for RT analysis was developed [166] using a gold nanocages (AuNCs)-modified carbon ionic liquid electrode (CILE), adopting the same procedure reported by Chen’s group to assemble an LUT sensor [134]. CILE was assembled using 1-hexylpyridinium hexafluorophosphate (IL) as the binder, and then AuNCs were deposited on its surface, resulting in AuNCs-modified CILE (AuNCs/CILE). Regarding the role of AuNCs and the electrochemical properties and behavior of CILE, we refer to what has been already reported about AuCNs/CILE in the luteolin subsection.

Concerning RT determination, AuCNs/CILE demonstrated a linearity range of 0.004–700.0 μM and an LOD of 1.33 nM: these results for RT were comparable with those obtained for LUT (1–1000 nm; LOD 0.4 nm), a similar flavonoid but not a glycoside as RT is. Threonine, alanine, valine, and some inorganic salts were assumed to be possible interfering molecules and excipients and, considering a tolerance limit involving a relative error of ± 10%, no evident interference was observed. The sensor’s operational stability (a 10% decrease in signal after 80 consecutive scans) and reproducibility (RSD 6.51%) were analyzed, with interesting results.

Finally, the sensor was applied to determine RT in pharmaceutical formulations, with satisfactory recoveries in the range of 94.00–108.40%, but no data from a standard analytical method were given.

Huang and coworkers prepared an electrochemical RT biosensor, using the polyphenol oxidase (PPO) enzyme, AuNPs, and FDU-15, a mesoporous carbon material [167].

FDU-15 was synthesized according to the evaporation-induced self-assembly (EISA) method [167] and showed a layered and mesoporous structure from the morphological analysis. Its conductivity, porosity, and surface-to-volume ratio were integrated with the conductivity and biocompatibility of AuNPs. In addition, PPO acted as the key recognition element for the RT detection. The biosensor (PPO/AuNPs/FDU-15/GCE) was prepared by modifying GCE via layer-by-layer assembly, and was electrochemically investigated by CV. The RT detection was carried out by amperometry, and a linear concentration range of 1.5 × 10^−6^–2.8 × 10^−5^ M with a detection limit of 5.1 × 10^−7^ M was evidenced. Glucose, *p*-aminophenol, ascorbic acid, and inorganic salts were selected as possible interferences, but no clear interference was evidenced. The biosensor’s stability and reproducibility were not investigated. The biosensor was applied to determine RT in black tea samples, with recoveries ranging from 98.5% to 105.5%. Moreover, in this case, no data from a standard analytical method were given.

A nanohybrid involving metal organic frameworks (MOFs), electro-reduced graphene oxide (ErGO), and AuNPs was reported for the assembly of an electrochemical sensor for RT [168].

MOFs can be assumed to be a porous coordination polymer, prepared by chemical or electrochemical methods [106,153]. MOFs showed high specific surface area, porosity, and a tailored structure, but, due to poor electron transport properties, they have to be coupled with appropriate nanosized materials for application in the electrochemical (bio)sensing area. The electrocatalytic activity, biocompatibility, and high specific surface area of AuNPs and the mechanical stability, functionalities, and surface-to-volume ratio of ErGO seem to be appealing to improve and integrate the MOF’s properties. The sensor was assembled via one-pot electrodeposition of GO, AuNPs, and a MOFs precursor on GCE, as illustrated in Figure 23.

ErGO-AuNPs-MOFs/GCE was electrochemically investigated by CV and EIS and used to determine RT by DPV. Two linearity ranges of 7 × 10^−9^–1.4 × 10^−7^ M and 1.4 × 10^−7^–4 × 10^−7^ M and an LOD of 3.44 nM were achieved. Stability (a 9.8% decrease in signal after eight days in the refrigerator), repeatability (RSD 4.7%), and reproducibility (RSD 5.3%) were analyzed, with acceptable results. Selectivity was examined, choosing glucose, citric acid, ascorbic acid, uric acid, and some inorganic salts as possible interfering compounds, but data and conclusions were not convincing. The sensor was applied to determine RT in spiked real samples of pharmaceutical products, with a recovery range of 86.9 to 104.8%, but no comparison with a standard method was provided.

Herein, we present some particular examples of electrochemical sensors for RT detection where graphene or its derivatives were present in the sensing interface.

A nanomaterial was synthesized that included boron-doped graphene (B-doped graphene) and Au-Pt bimetallic core-shell nanoparticles (Au@AuPt NPs) and was used to modify GCE to detect RT by DPV [169]. B-doped graphene was prepared by the hydrothermal process with B_2_O_3_ as the reductant and boron source. B-doping can act as an electron acceptor, modifying the mechanical properties, electrocatalytic activity, and electron transfer properties of graphene, while also improving the surface and adsorption properties and, in this case, inducing a positive charge to assist in the adsorption of negatively charged nanoparticles.

It is well known that despite these properties, G presents a serious problem consisting of agglomeration and aggregation of graphenic nanosheets. To prevent this drawback, metal, bimetallic, and/or metal oxide nanoparticles can be incorporated in the graphene nanostructure. The metal nanoparticles also improve the electrocatalytic efficiency and electron transfer rate. Au@AuPt NPs were synthesized using AuNPs as nucleating agents; then, a solution containing the two metallic precursors supported the deposition of the AuPt alloy on the surface of Au seeds, forming the Au@AuPt NPs.

Au@AuPt NPs/B-doped graphene was prepared via sonication of a solution containing Au@AuPt NPs and B-doped graphene suspensions. The nanohybrid morphological and structural analysis evidenced the bimetallic nanoparticles uniformly dispersed on B-graphene nanosheets. The nanocomposite was dropped on the GCE surface, and the modified electrode was electrochemically characterized by CV and EIS. The electrochemical behavior of RT on the modified electrode was investigated and compared with that obtained with bare GCE and GCE modified with only Au@AuPt NPs or B-doped graphene.

The determination of RT was carried out by DPV, obtaining a linearity range of 2.00 × 10^−9^ to 4.00 × 10^−6^ M and a detection limit of 2.84 × 10^−10^ M. Reproducibility and operational stability were analyzed, with satisfactory results. The selectivity was tested, considering metals salts, fructose, glucose, urea, glycine, leucine, asparagine, phenol, hydroquinone, and ascorbic acid as possible interferences, and no clear interference was demonstrated. The sensor was applied to determine RT in pharmaceutical products, evidencing recoveries in the range of 97.23–101.65%, but no comparison with data from a standard method was given.

A traditional stainless-steel acupuncture needle (AN) was used as the electrode to assemble an electrochemical sensor for the determination of RT in pharmaceutical and human urine samples [170].

AN was modified with G and AuNPs through two successive electroreduction steps. Firstly, AuNPs were electrodeposited on AN by electroreducing their precursor, and then GO was electroreduced to G at the AuNPs/AN electrode. From the nanocomposite’s structural and morphological analysis, G layers filled the voids among AuNPs. This nanohybrid can be assumed to be another example of integration of the properties of G and AuNPs.

Under optimized conditions, the determination of RT was carried out by DPV, evidencing two linear concentrations ranges, 8.0 × 10^−8^–1.0 × 10^−5^ M and 2.0 × 10^−5^–2.0 × 10^−4^ M, as well as an LOD of 2.5 × 10^−8^ M. Concerning the reproducibility (RSD 4.2%) and stability (a 4.8% decrease in signal after three days at room temperature), interesting results were achieved. Considering inorganic salts, lactose, glucose, citric acid, dopamine, uric acid, and ascorbic acid as possible interfering compounds, no evident interference was found, with a tolerance limit involving a relative error of ± 4%. The sensor was applied to spiked real samples of pharmaceutical products and human urine, with recoveries ranging from 97.40% to 98.05% and 97.20% to 103.60%, respectively. In addition, the results were in agreement with those obtained from the standard method, i.e., UV–vis spectrophotometry.

An Au-Ag nanothorns (Au-Ag NTs) composite has been synthesized through a seed-mediated chemical reduction and then assembled on N-doped graphene (NG) [171].

In detail, Au-Ag NTs were prepared using L-dopa as a reducing agent of HAuCl_4_ in the presence of Ag nanoparticles as seeds, and then they were deposited on NG.

The combination of Au-Ag NTs and NG produced a Au-Ag NTs/NG nanocomposite with improved electrocatalytic activity, electron transfer rate, mechanical stability, and conductivity with respect to the starting nanomaterials. The Au-Ag NTs/NG nanohybrid was drop-casted on GCE and the resulting Au-Ag NTs/NG/GCE was used for RT electroanalysis by DPV. A linearity range of 0.1–420 μM and a detection limit of 0.015 μM were achieved. The stability was investigated, and no clear decrease in response compared to its initial value was evidenced after 20 days, but no data on reproducibility and repeatability were given. The selectivity was tested against inorganic salts, dopamine, uric acid, glucose, and ascorbic acid, and no interference was observed. Au-Ag NTs/NG/GCE was applied to detect RT in pharmaceutical formulations, and the results, with a recovery range of 97–106%, were in agreement with the RT amount declared by the manufacturers.

Apetrei developed an RT electrochemical sensor, modifying SPCE with a suspension containing AuNPs and G [172]. Graphene was firstly dispersed in a chitosan solution to avoid the agglomeration and aggregation of the G nanosheets. AuNPs were prepared via chemical reduction and then mixed with a G/chitosan suspension. Finally, the AuNPs/G suspension was casted on the SPCE surface. The AuNPs/G nanocomposite evidenced a typical 3D morphology, involving a few layers of G nanoflakes, decorated with spherical AuNPs.

AuNPs/G/SPCE was electrochemically characterized by CV. The RT electroanalysis was performed by SWV, achieving a linear concentration range of 1 × 10^−7^–1.5 × 10^−5^ M and an LOD of 1.1 × 10^−8^ M. Repeatability (intra-day repeatability RSD 2.1% and inter-day repeatability RSD 3.0%) provided satisfactory results in terms of RSD%. Citrate, acetate, lactose, sodium lauryl sulfate, lactic acid, starch, sucrose, ascorbic acid, and inorganic salts were assumed to be possible interfering molecules and excipients, and no evident interference was observed considering a tolerance limit involving a relative error of ± 5%. The sensor was applied to detect RT in different pharmaceutical formulations, with recoveries ranging from 96.52 to 102.97%. Results were also comparable with those obtained via the standard spectrophotometric method.

A composite involving G, AuNPs, and MIP was prepared and used to modify SPCE, to develop an RT electrochemical sensor [173]. The G/AuNPs composite was synthesized by the simultaneous electrochemical reduction of GO and the AuNPs precursor at SPCE. As already reported, MIP was then synthesized onto the modified SPCE via chemical polymerization, with RT acting as the template molecule and AM as the functional monomer. RT was removed just after polymerization. According to the morphological and structural analysis, AuNPs were well-distributed on G nanosheets, and the MIP film covered the AuNPs uniformly, providing specific binding cavities complementarily to RT molecules, thus acting as the recognition key element.

The modified electrode was electrochemically characterized by CV and EIS. The electroanalysis of RT was carried out by DPV, with a linearity range of 0.04 to 60.0 μM and detection limit of 0.014 μM. The storage stability (a9.6% decrease in response after six week in dry conditions at room temperature) and reproducibility (RSD 3.64%) were addressed, with acceptable results. Selectivity was investigated considering dopamine, ascorbic acid, uric acid, glucose, hydroquinone, phenol, morin, serine, valine, histidine, asparagine, caffeine, and some inorganic salts as possible interfering compounds. The results indicated the sensor’s good selectivity. G/AuNPs/MIP/SPCE was applied to determine RT in drug formulations, with satisfactory recovery results in the range of 98.0–104.7%, but no comparison with a standard method was given.

An MIP based on chitosan (CHI) has been used to develop a nanohybrid including a rose-like Au nanoparticles-MoS_2_-graphene composite (AuNPs-MoS_2_-G) [174], used to modify GCE for the detection of RT. In particular, G was prepared by hydrothermal reduction [175], using GO as a precursor and water as a reducing agent, and the rose-like AuNPs-MoS_2_-G composite was produced by the solvothermal method, according to the procedure in the literature [176]. G provided an appropriate platform to grow MoS_2_ nanoflowers (NFs), where AuNPs were deposited. The AuNPs-MoS_2_-G composite had a high specific surface area, good conductivity, and electrocatalytic activity. The corresponding AuNPs-MoS_2_-G/GCE, obtained by casting the nanocomposite on the electrode surface, showed an enhanced electrochemical response vs. RT. Finally, the RT-imprinted chitosan film was electrodeposited on the AuNPs-MoS_2_-G/GCE. MIP, as usual, acted as the key recognition element to improve the sensor selectivity.

MIP/AuNPs-MoS_2_-G/GCE was characterized by microscopic and electrochemical techniques and used to determine RT by DPV. A linear concentration range of 0.01–45 μM and a detection limit of 0.004 μM were achieved. Naringenin, morin, quercetin, glucose, and dopamine were considered as possible interferences: The MIP presence guaranteed good selectivity in comparison with AuNPs-MoS_2_-G/GCE. In addition, the storage stability (a 5.4% decrease in signal after 14 days at 4 °C) and reproducibility (RSD 4.5%) were analyzed, showing acceptable results The sensor was applied to real samples of *Ginkgo biloba*, buckwheat tea, and RT tablets, obtaining recoveries ranging from 95.2% to 104.7%, but no comparison with a standard method was provided.

An electrochemical sensor for RT analysis in pharmaceutical products was developed based on GCE modified with a nanohybrid involving Cu_2_O, AuNPs, and N-doped graphene (Cu_2_O-AuNPs/NG) [177].

Firstly, Cu_2_O nanoparticles were synthesized by chemical reduction using ascorbic acid as a reducing agent, and the resulting nanoparticles had a cubic shape. The AuNPs were then prepared by chemical reduction in the presence of Cu_2_O and G, with Cu_2_O acting as the reductant. The final nanocomposite morphology consisted of Cu_2_O cubes well dispersed on G and AuNPs grown on the surface of Cu_2_O nanocubes.

Cu_2_O-AuNPs/NG/GCE was characterized by CV and DPV, and RT was determined by DPV. Under optimized experimental conditions, a linearity range of 0.06–512.90 μM and an LOD of 30 nM were obtained. Glucose, ascorbic acid, dopamine, uric acid, and inorganic salts were tested as possible interferences and excipients, and the sensor showed good selectivity. Cu_2_O-AuNPs/NG/GCE was applied to determine RT in pharmaceutical products, and recoveries ranging from 98.95 to 101.81% were obtained. In addition, the data in agreement with those obtained by HPLC as an external reference method.

Nitrogen and sulphur co-doped with reduced GO decorated with AuNPs (NS-rGO/AuNPs) were synthesized to assemble an RT electrochemical sensor [178]. NS-rGO was prepared by the one-step thermal annealing approach, and AuNPs resulted in being anchored onto NS-rGO because of the interactions of Au with N and S atoms. The morphological and structural analysis showed that AuNPs were well distributed on the rGO sheets. NS-rGO layers provided an appropriate nanostructured surface for promoting effective RT absorption, and AuNPs contributed to speeding up the oxidation of absorbed RT.

The NS-rGO/AuNPs-modified GCE exhibitedpeculiar electrocatalytic activity towards RT compared with NS-rGO-, N-rGO-, and S-rGO-modified electrodes. The particular performance was achieved by the synergy and interactions of AuNPs with NS-rGO via N and S, the two heteroatoms. After the sensor electrochemical investigation, RT was detected by DPV, and the NS-rGO/AuNPs/GCE showed a linear range of 0.2–1400 nM with a detection limit of 0.067 nM. Citric acid, sucrose, tryptophan, and glycine were tested as possible interferences, and the sensor showed good selectivity. Moreover, the storage stability (a 1.05% decrease in signal after one week at room temperature), operational stability (a 6.37% decrease in signal after 100 successive measurements), and reproducibility (RSD 4.97%) were studied, evidencing satisfactory results in terms of RSD%. The sensor was applied to real samples of RT tablets, obtaining recoveries ranging from 97.08% to 100.90%, but no comparison with a standard method was provided.

#### 3.3.5. Catechin

Catechin [(2*R*,3*S*)-2-(3,4-dihydroxyphenyl)-3,4-dihydro-2*H*-chromene-3,5,7-triol, CAT, chemical structure in Figure 24] is present in chocolate, grapes, green tea, and wine [179,180].

CAT and its isomer epicatechin (EC), together with its derivatives such as esterified catechins (gallate-type), including epigallocatechin (EGC) and epigallocatechin gallate (EGCG), are the most abundant polyphenols in green tea. They are well-known for their antioxidant, antibacterial, and anti-aging properties. It is to be noted that CAT and its family are not chemically stable compounds [179]; for this reason, CAT can be encapsulated in natural biopolymers to prevent its degradation in food and/or in pharmaceutical formulations [180].

Herein, we reported some examples of electrochemical (bio)sensors for CAT detection involving gold nanomaterials.

Singla et al. developed an electrochemical biosensor for CAT analysis in apple juice [181]. A nanocomposite including AuNPs and polypyrrole (PPY) (AuNPs/PPY) was electrochemically synthesized on a Pt electrode, and TYR was entrapped in the nanohybrid matrix during the electrosynthesis procedure. The presence of AuNPs in the polymer matrix enhanced conductivity, the electrocatalytic response, and the electron transfer rate. The modified electrode (Tyr/AuNPs-PPY/Pt) was electrochemically characterized and employed for the detection of CAT by CV. A linear concentration range of 1 × 10^−9^ to 1 × 10^−8^ M and a detection limit of 1.2 × 10^−9^ M were achieved. The biosensor selectivity was tested for gallic acid and catechol, and the CV response showed no signal ascribable to the two molecules. In addition, the reproducibility of the sensor was tested using the same electrode for CAT analysis approximately 15 times, and the study showed that for approximately 12 to 13 assays, the current intensity was almost constant. The long-term storage stability of the sensor was monitored for 80 days. The signal response decreased to 85% up to 70 days, but then it gradually decreased, likely because of the loss of catalytic activity or change in the stability of the AuNPs/PPY composite. The biosensor was applied to determine CAT in apple juices, and results were comparable to those obtained by LC-MS/MS analysis.

Sarkar and co-workers developed a CAT biosensor to immobilize AuNPs together with TYR on a GCE [182].

An eco-friendly, one-pot, reagent free, extracellular synthetic approach of AuNPs was employed, involving two bacteria, *Pseudomonas alcaligenes* RJB-B and *Pseudomonas resinovorans* RJB-3, as reducing agents, isolated from arsenic-contaminated soil. Extracellular synthesis of AuNPs is connected to the presence of reductase enzymes released by the two bacteria, acting as reducing agents. Moreover, the AuNPs were stabilized by the proteins present in the synthesis environment. TYR was immobilized on AuNPs/GCE via cross-linking with glutaraldehyde. The biosensor’s analytical performances were investigated by DPV, and an LOD of 7.7 μM was obtained. Selectivity, reproducibility, and stability were not investigated. The biosensor was applied to detect the total polyphenols content, expressed in terms of CAT amount, in commercial tea samples. The results were comparable to those obtained by HPLC.

An electrochemical sensor was assembled using a nanocomposite involving N-doped graphene (NG) and Au@Pt core-shell nanoparticles [183].

Au@Pt core shell-nanoparticles were synthesized by means of the seed-mediated method [56,57], and the morphological analysis showed a core-shell structure with Au as the core and Pt as the shell. As already reported, NG and Au@Pt NPs acted together to integrate and improve their properties in terms of mechanical stability, conductivity, surface area, and electron transfer capability. The N-G/Au@Pt NPs composite was dropped onto a gold electrode, and the resulting sensor was used to detect CAT by DPV. After the optimization of analytical parameters, a linear concentration range of 1.0 × 10^−7^–4.5 × 10^−5^ M and detection limit of 2.85 × 10^−9^ M were achieved. Reproducibility and stability were examined with acceptable results in terms of RSD%, but, unfortunately, selectivity was not considered. The sensor was applied to determine CAT in tea samples, with recoveries in the range of 99.94–101.50 %, but no comparison with a standard method was given.

#### 3.3.6. Some Considerations on Flavonoids (Bio)sensors Based on Au Nanomaterials

As a conclusive comment, regarding all the sensor examples for the detection of flavonoids, we can observe that LODs are generally nanomolar, as well as picomolar in several examples, independently of the analyte. The corresponding linearity ranges are not particularly wide, and double linearity ranges are provided for several sensors, but seldom is an adequate explanation suggested.

Questionable points are represented by the data relating to the sensor’s selectivity and the subsequent validation of the proposed method; in fact, these issues are not always adequately addressed. Concerning selectivity, in many cases, the choice of interfering compounds does not appear related to the sensor application field, which would have been more appropriate.

Finally, a comparison between data obtained with the sensor and data obtained with a standard reference method would be very useful, for better evaluation of the performance of the sensor and its reliability and accuracy, highlighting the potential of the electrochemical approach.

We have to stress the fact that, in many cases, a green and eco-friendly approach has been achieved for the synthesis of the nanomaterials involved, and this is very promising for possible future developments.

The analytical performances of the electrochemical sensors reported for the determination of flavonoids as well as the corresponding sensor format are summarized in Table 3. The linearity range and LOD data are indicated in the same unit of measurement (μM) to facilitate a comparison among them.

### 3.4. Synthetic Phenolic Antioxidants

As already mentioned in the Introduction, the most common synthetic antioxidants used in food to slow down the unwanted oxidative reactions of oils and fats and to preserve the life and quality of food products [13,14] are synthetic phenolic antioxidants (SPAs).

In the present review, we have reported some significant examples of electrochemical sensors including gold nanomaterials for the detection of the most common SPAs, such as butylated hydroxyanisole (BHA), *tert*-butylhydroquinone (TBHQ), propyl gallate (PG), and octyl gallate (OG), whose chemical structures are shown in Figure 25.

#### 3.4.1. Butylated Hydroxyanisole

Butylated hydroxyanisole (BHA) is a synthetic antioxidant, widely used in foods, pharmaceutical and cosmetic formulations, as well as in animal feed. Generally, it is added to foods to avoid the fat rancidification process that produces an unpleasant odor.

A sensor based on GCE modified with AuNPs was used to detect BHA and two other antioxidants, such as *tert*-butylhydroquinone (TBHQ) and butylated hydroxytoluene (BHT), in samples of oil for use in food [184].

As usual, the electrodeposited AuNPs provided a high surface-to-volume ratio and enhanced the electron transfer rate to and from the electrode surface. The electrochemical behavior of the three SPAs with the modified GCE was investigated and compared to that at the bare electrode, evidencing the positive action of the AuNPs, considering the electrochemical response amplification and the possibility to determine the three SPAs simultaneously. BHA, BHT, and TBHQ were detected by means of linear sweep voltammetry (LSV), showing linear concentration ranges of 0.10–1.50 μg mL^−1^, 0.20–2.20 μg mL^−1^, and 0.20–2.80μg mL^−1^, respectively. The corresponding LODs were 0.039, 0.080, and 0.079 μg mL^−1^ for BHA, BHT, and TBHQ, respectively.

Different common interfering compounds present in oil samples such as ascorbic acid, vitamin E (DL-α-tocopherol), phthalate, citric acid, and metal salts were tested, considering a tolerance limit involving a relative error of ±10%. The results indicated that the analysis of all SPAs in oil samples was not significantly affected by the above-mentioned interference. The reproducibility of the AuNPs/GCE was investigated, and results were satisfactory (RSD 2.30, 3.82, and 3.16% for BHA, BHT, and TBHQ). The sensor was applied to commercial samples of oils for food use, with recoveries ranging from 93.7 to 105% for BHA, from 92.3 to 97% for BHT, and from 91.6 to 103% for TBHQ. Results were comparable to those obtained by HPLC as the reference external method.

Kan employed exfoliated graphite paper (EGP) as the electrodic material for the electrochemical determination of BHA by DPV [185]. After the electrodeposition of AuNPs and NiO nanoparticles, the final assembled sensor evidenced a multi-layer structure with all the nanoparticles uniformly dispersed.

Moreover, it should be noticed that the sensor’s performances are strictly related to the high surface area of EGP, the conductivity of both EGP and AuNPs, and the electrocatalytic activity of NiO nanoparticles. Under optimal conditions, the sensor showed a linear range of 3.0 × 10^−8^–5.0–10^−5^ M with a limit of detection of 2.0 × 10^−8^ M. Stability and reproducibility (RSD 1.29%) were analyzed, with promising results. The sensor was used for detection at 10-day intervals over a period of 20 days. The current response decreased to 95.20% and 90.62% of the initial current for the first and the second 10-day periods, respectively. Concerning repeatability, after the measurement of 5.0 × 10^−5^ M BHA, the sensor was dipped in ethanol to remove the adsorbed BHA. Then, it was employed for the second measurement. After five cycles of measurement/removal, the current response decreased by 3.3%. Ascorbic acid, BHT, TBHQ, and PG did not significantly affect the electrochemical response of BHA. The sensor was applied to the analysis of BHA in oils for use in food, with recoveries ranging from 97.00–104.00%.

An electrochemical sensor for the analysis of BHA in foodstuffs was developed using CHI capped with AuNPs [186].

The nanocomposite was self-assembled on SPCE, and CHI was used as an MIP by employing BHA as a template. It is well known that CHI presents a large number of amino (–NH_2_) and hydroxyl (–OH) groups in the polymer backbone, several reaction sites, and a highl crosslinking degree as well as good stability. On the other hand, AuNPs possess a large surface area and high conductivity. In this example, the choice of CHI as a polymer was based on its -NH_2_ groups, possibly interacting with the -OH groups of the BHA through a hydrogen bond. This allows for the formation of MIP with appropriate recognition sites.

The BHA-imprinted MIP sensor exhibited good sensitivity and selectivity compared to interfering species such as ascorbic acid and citric acid. Under the optimal experimental conditions, it showed a linear concentration range of 0.01 to 20 μg mL^−1^, with a low detection limit of 0.001 μg mL^−1^. Reproducibility, stability, and repeatability were tested, and selectivity was evaluated using citric and ascorbic acid as possible interfering compounds. The results evidenced no clear interference as a result of the MIP presence. The MIP sensor was applied to detect BHA in real spiked food samples such as chewing gum, mayonnaise, and potato chips, with an RSD value of ≤8%. Spectrophotometry was utilized as a validation method, with acceptable results.

The next two examples involved nanocomposites where a peculiar and very important role is played by graphene.

Wang and coworkers developed an electrochemical sensor for the simultaneous detection of BHA and TBHQ in edible oils, using GCE modified with a binary nanocomposite including AuNPs and ErGO [187]. The nanocomposite was electrochemically prepared by the co-reduction of GO and the AuNPs precursor onto the GCE surface. The AuNPs’ role consisted of improving the electron transfer between the analyte and the electrode surface and, on the other hand, ErGO increased the electroactive area of the electrode surface, thus resulting in the enhancement of the electrochemical response.

BHA and TBHQ were detected by LSV, showing linear concentration ranges of 0.1–10 μg mL^−1^ and 0.1–7 μg mL^−1^ with limits of detection of 0.0419 μg mL^−1^ and 0.0503 μg mL^−1^, respectively. BHT and metal salts were tested as possible interfering compounds. Results indicated that the analysis of BHA and TBHQ in oil samples was not significantly affected by the above-mentioned interferences. The sensor’s operational and storage stability were also evaluated, with interesting results. The sensor was then applied to detect BHA and TBHQ in edible oil samples, producing results comparable with those obtained by HPLC, considered as the validation method.

A nanocomposite including AuNPs, G, and polyvinylpyrrolidone (PVP) was synthesized to modify GCE to determine BHA in soybean oil and flour samples using LSV [188]. As already mentioned, the AuNPs’ role consisted of improving the electron transfer rate to and from the electrode surface, and on the other hand, graphene increased the electroactive area of the electrode surface, thus amplifying the electrochemical response in a synergistic way. Finally, PVP acted as an effective dispersing agent to prevent the agglomeration of both AuNPs and graphene sheets. A linear concentration range of 0.2–100.0 μM and an LOD of 0.04 μM were obtained. Repeatability (RSD 2.0%) and stability (a 2.0% decrease in the response after one week in air) were investigated, with acceptable results. Vitamin E (DL-α-tocopherol), BHT, and citric acid were selected as possible interfering compounds, together with different inorganic salts. The results indicated that they did not affect the BHA electrochemical response. The sensor was applied to real samples of soybean oil and flour, with recoveries ranging from 93 to 105%, but no comparison with a standard method was provided.

A biosensor was assembled to determine BHA and OG in olive and peanut oils, potato chips, and cookies, employing spiny Au-Pt nanotubes (SAP NTs) and an AuNPs/graphene nanohybrid as an immobilizing/entrapping platform for Horse-radish peroxidase (HRP) [189].

A GCE was firstly modified with the AuNPs/G nanohybrid, and then HRP was immobilized on it through electrostatic interactions. Finally, SAP NTs were deposited on the HRP/AuNPs/G/GCE, so HRP resulted in being entrapped in a collaborative network including SAP NTs and the AuNPs/G nanohybrid. The simultaneous determination of BHA and OG was carried out by LSV, and under optimized conditions, a linear concentration range of 0.3–50 mg L^−1^ for BHA and 0.1–100 mg L^−1^ for OG, as well as a detection limit of 0.046 mg L^−1^ and 0.024 mg L^−1^, respectively, was obtained. Reproducibility, repeatability, and stability were analyzed, with satisfactory results, but, unfortunately, the selectivity issue was not addressed. The repeatability and reproducibility were tested with an intra-assay and inter-assay by measuring a fixed concentration via LSV across six measurements. The RSD% values of the intra- and inter-assays for BHA and PG were 4.6%, 3.8% and 5.2%, 4.5%. Moreover, the biosensor stability was tested in a solution mixture of BHA and PG over a period of 10 weeks, and the electrochemical response showed random fluctuations within the limits of statistical control.

SAP NTs/HRP/AuNPs/G/GCE was applied to spiked real samples of olive and peanut oils, potato chips, and cookies, with recoveries ranging from 87.3 to 126.2%. In addition, data resulted in being comparable to those achieved using HPLC.

#### 3.4.2. Tert-Butylhydroquinone

*Tert*-butylhydroquinone (TBHQ) is a synthetic and low-cost antioxidant widely used as an additive in edible oils and food products tp prevent fat degradation.

A nanocomposite including Au and SnO_2_ nanoparticles, GNs, and MWCNTs was designed and developed to assemble an electrochemical sensor of TBHQ in cooking oils and beverages [190].

The Au–SnO_2_/GNs-SWCNTs nanocomposite synthesis involved several steps. The first included the synthesis of the GNs-SWCNTs hybrid via the hydrothermal reduction of GO in the presence of SWCNTs. The next step included SnO_2_ nanoparticles synthesis in the presence of GNs-SWCNTs. Finally, AuNPs were deposited via photo-reduction onto SnO_2_/GNs-SWCNTs. As usual, the combination of different nanomaterials, such as Au and SnO_2_ nanoparticles and the GNs-SWCNTs nanohybrid, resulted in an enhancement of the electron-transfer reaction and an improvement of the electroactive area.

The TBHQ’s electrochemical behavior was investigated on GCE modified with the nanocomposite and determined by DPV under optimized conditions. A linear concentration range of 5.0 × 10^−8^–2.3 × 10^−4^ M with a detection limit of 5.8 × 10^−8^ M was achieved. Reproducibility, stability, and selectivity were investigated, with satisfactory results. Methanol, ethanol, resorcinol, catechol, and some inorganic salts were tested as possible interferences. Finally, the sensor was applied to determine TBHQ in cooking oils and beverages, with recoveries ranging from 97.9 to 100.8% and results comparable with those acquired with HPLC as the reference external method.

Kan reported an electrochemical sensor for TBHQ detection in edible oils using a nanocomposite based on exfoliated graphite paper (EGP) as the electrodic material, modified with MIP and AuNPs [191].

As a reminder, the same group developed an electrochemical sensor for BHA detection using EGP modified with AuNPs and NiO, as already mentioned in Section 3.4.1 [185]. EGP presented a three-dimensional (3D) structure where AuNPs and MIP were electrodeposited, and the nanoparticles resulted in being well-distributed. MIP was obtained by means of electropolymerization of pyrrole at AuNPs/EGP in the presence of TBHQ as the template molecule. TBHQ was removed just after the end of polymerization. As already discussed, MIP acted as the key recognition element, while AuNPs and EGP participated in increasing the electron transfer rate and the electroactive surface area. MIP/AuNPs/EGP was used to detect TBHQ by DPV and, after optimizing the experimental parameters, a linearity range of 8.0 × 10^−8^–1.0 × 10^−4^ M with an LOD of 7.0 × 10^−8^ M was acquired.

Butylated hydroxyanisole (BHA), hydroquinone (HQ), methyl hydroquinone (MHQ), 2,4-di-*tert*-butylphenol (TBP), and 2,5-di-*tert-*butylhydroquinone (BHT) were tested as possible interfering compounds, and the selectivity resulted in being satisfactory thanks to the presence of MIP. Reproducibility (RSD 1.17%) and stability (a 19.94 % decrease in the response after 20 days at 4 °C) were considered, with acceptable results. The sensor was also applied to spiked real samples of oils for food use, with recoveries from 95.80% to 102.3%, but no comparison with data from an external reference method was provided.

Au-Pt nanocrystals were prepared to modify GCE to assemble an electrochemical sensor to detect TBHQ in canola oil and soybean oil [192]. The bimetallic nanocrystals were synthesized starting with Au nanorods (AuNRs), obtained via Au seed reduction, in the presence of cetyltrimethylammonium bromide (CTAB) as a dispersing agent to avoid possible nanorod agglomeration. Subsequently, Pt NPs were deposited on AuNRs via chemical reduction to obtain the bimetallic nanocomposite. At this point, cetyltrimethylammonium chloride (CTAC) acted as a stabilizing agent to prevent nanocrystal aggregation and preserve their morphology. A suspension of the nanocrystals and chitosan was then dropped on GCE. The TBHQ’s electrochemical behavior on Au-Pt/GCE was investigated and compared to those with bare and AuNRs-modified electrodes, evidencing an amplification of the electrochemical response due to the presence of the bimetallic nanomaterial. The TBHQ electroanalysis was performed using DPV, and a linear range of 0.35 μM to 625 μM with a detection limit of 0.075 μM was achieved. The reproducibility and stability of Au-Pt/GCE were analyzed, with acceptable results in terms of RSD%. Ethanol, ascorbic acid, hydroquinone, BHA, PG, and inorganic salts were selected as possible interfering compounds. The results indicated that the TBHQ response was not affected by the presence of such interferences. The sensor was then applied to real samples of canola oil and soybean oil, with recoveries of 99.2–102.1% and 98.4–102.5%, respectively.

A nanocomposite was prepared by integrating an MIP with carbon and metallic nanomaterials to assemble an electrochemical sensor to determine TBHQ in edible oils [193]. In particular, bimetallic nanoparticles (Au-PdNPs) were electrodeposited onto ErGO. Finally, MIP was electropolymerized onto a Au-PdNPs/ErGO composite, using *o*-phenylenediamine (*o*-PD) as the functional monomer and TBHQ as the template molecule. TBHQ was removed at the end of electropolymerization. The introduction of Au-PdNPs and G increased the active surface area of the electrode, while the molecularly imprinted polymer acted as the key recognition element, as usual, therefore improving the sensor’s performance and selectivity.

The MIP/Au-PD NPs/ERGO nanohybrid was employed to modify GCE, and the assembled sensor was electrochemically characterized. After optimizing the analytical parameters, TBHQ was determined by DPV, and a linearity range of 0.5–60 μg mL^−1^ with a limit detection of 0.046 μg mL^−1^ was acquired. The reproducibility and repeatability of MIP/PdAuNPs/ERGO/GCE were investigated, and a decrease of the initial current response to 88.57% for TBHQ was obtained, after the electrode was continuously used for 15 days at 4 °C, indicating good reproducibility and repeatability. BHA, BHT, and PG were tested as possible interfering molecules, and the corresponding results evidenced their negligible effects thanking to the key role of MIP as the recognition element. The sensor was then applied to real samples of edible oils with recoveries of 99.44–108.50%, and the results were comparable to those acquired by HPLC.

A composite including AuNPs deposited on tungsten carbide (AuNPs-WC) was synthesized, as shown in Figure 26, and used to fabricate an electrochemical sensor for TBHQ in soybean oil, blended oil, and red wine [194].

WC showed peculiar properties, such as its d-band electronic structure, large surface area, and physicochemical and catalytic characteristics [195], but it resulted in being not particularly stable with low conductivity, considering its semiconducting nature. For these reasons, the coupling of conducting AuNPs with the large WC surface area produced a nanocomposite to be used as an active sensing platform.

A GCE was modified with a AuNPs-WC nanohybrid and was used to determine TBHQ by DPV, obtaining a linear concentration range of 5–75 nM, with an LOD of 0.20 nM. Good reproducibility (RSD 3.26%), operational stability (a 5.28% decrease in response after 100 consecutive assays), long-term stability (a 2.62% decrease in the signal after three weeks in the refrigerator at 4 °C), and repeatability (RSD 2.54%) data were acquired. Ascorbic acid, glucose, dopamine, vanillic acid, uric acid, rutin, caffeic acid, catechol, 4-nitrophenol, quercetin, gallic acid, ferulic acid, and inorganic salts had no significant effect on the TBHQ electrochemical response (i.e., a signal change of less than 5%). The sensor was also applied to real samples of soybean oil, blended oil, and red wine, with satisfactory recovery rates from 88 to 97% for the blended oil, from 92 to 99% for the soybean oil, and from 94 to 97% for the red wine. Finally, results were comparable to those acquired by HPLC.

#### 3.4.3. Propyl Gallate and Octyl Gallate

PG and OG are both esters of gallic acid (3,4,5-trihydroxy benzoic acid) with two different alcohols, propanol, and n-octanol, used in foods to retard oil and fat rancidification and/or decolorization.

An electrochemical sensor was developed by modifying a GCE with a nanocomposite based on G, CNTs, bimetallic nanoparticles (PtAu NPs), and MIP to detect PG in vegetable oils [196]. This sensor was assembled via electropolymerization using *o*-phenylenediamine as functional monomer in the presence of PG as template molecule on the surface of GCE modified by the PtAu-rG-CNTs nanohybrid. The nanohybrid was prepared via co-reduction of PtAuNPs precursors and GO, in the presence of CNTs, according to the procedure reported previously in the literature [187]. Using thee PtAu-rG-CNTs composite as the electrode material can improve the electrochemical sensor performances, increasing the active surface area and the electron transfer rate, while MIP acted as a key recognition element, as already reported.

The analytical performances of MIP/PtAu-rG-CNTs/GCE were investigated, and PG was determined by chronoamperometry, obtaining a linear concentration range of 7 × 10^−8^ to 1 × 10^−5^ M, with a limit of detection of 2.51 × 10^−8^ M. BHA, BHT, and TBHQ were evaluated as possible interfering compounds, since they are structural analogues of PG. The PG electrochemical response was not affected by the presence of such interferences because of the MIP presence. Stability (a 7.0% decrease in the response after two weeks at 4 °C) and reproducibility (RSD 4.1%) were analyzed, with satisfactory results. Finally, the sensor was tested for detecting PG in vegetable oil, and it was found that the recovery was 98.3 to 103.0%.

The next example described the use of (AuNPs)/poly(*p*-aminobenzenesulfonic acid) [poly(*p*-ABSA)] composite-modified GCE for the electrochemical determination of PG [197]. Poly(p-ABSA) and AuNPs were electrodeposited in two successive steps onto the electrode surface. The AuNP/poly(*p*-ABSA) composite facilitated the electron transfer between the sensing interface and the analyte by lowering the PG oxidation potential and amplifying the corresponding electrochemical response. After the optimization of experimental parameters, the determination of PG was carried out using DPV, and a linearity range of 9.0 × 10^−6^ to 1.0 × 10^−4^ M with a limit of detection of 1.9 × 10^−7^ M was obtained. Moreover, the sensor showed satisfactory results in terms of reproducibility (RSD 1.1%), and concerning stability, no clear change in the signal response was evidenced after one-week storage.

Acetic and citric acid and EDTA were considered as interferences, and the PG electrochemical response was not affected by their presence. The developed sensor was then applied to determine PG in coconut and sunflower oils, and recoveries were between 98.9 and 101.3%. Finally, the obtained data were comparable with those from the standard spectrophotometric assay.

A GCE modified with AuNPs followed by a self-assembled monolayer of dodecane thiol (DDT) was developed and employed for the determination of OG in margarine, butter, and sunflower oil [198]. In comparison to the bare GCE and AuNPs/GCE, the OG electrochemical behavior in DDT/AuNPs/GCE indicated easier electron transfer between the electrode surface and the target analyte by lowering the corresponding oxidation potential and increasing the electrochemical response. The electrochemical detection of OG was performed by means of SWV, and a linear concentration range of 0.20–1.20 μM and LOD of 8.3 nM were acquired. In addition, acceptable results were found in terms of repeatability (RSD 4.56%) and reproducibility (RSD 2.24%). Antioxidants such as TBHQ, PG, citric acid, dodecyl gallate (DG), and BHA, together with inorganic salts, were investigated as possible inferences, but the OG’s electrochemical response was not affected by their presence. DDT/AuNPs/GCE was employed to determine OG in margarine, butter, and sunflower oil, and the recoveries ranged from 99.1 to 100.6%. Finally, the obtained results agreed with those of the standard spectrophotometric method.

#### 3.4.4. Some Considerations on Synthetic Phenolic Antioxidants (Bio)sensors Based on Au Nanomaterials

As a conclusive comment, regarding all the sensor examples for the detection of synthetic phenolic antioxidants, we can observe that the LODs, independently of the analyte, are generally micromolar, and the corresponding linearity ranges are not particularly wide.

Concerning the selectivity, the criterion of choice of the different interferences does not appear clear because, above all, the sensors were usually applied to the same type of real samples, i.e., edible oils.

A comparison with a standard reference method was adopted in the majority of reported examples, underlying the value of the electrochemical approach.

Finally, we have to stress that several sensors can be used for the simultaneous detection of different SPAs, and they should be viewed as attractive for commercial exploitation.

The analytical performances of the electrochemical sensors reported for the determination of SPAs, as well as the corresponding sensor format, are summarized in Table 4. The linearity range and LOD data are indicated in the same unit of measurement (μM) to facilitate comparisons among them.

## 4. Conclusions

In this section, we would like to outline and summarize some considerations regarding, in particular, the electrode materials and nanomaterials involved.

We would also like to draw some conclusions concerning detectability, detection limits, selectivity, the validation of sensors with a standard method of analysis, and the possibility of determining several analytes at the same time.

Finally, possible future perspectives for employing gold nanomaterials in the realization of reliable and performant sensors for phenolic antioxidants will be indicated.

Glassy carbon was the most-used electrodic material, probably because of its conductivity, thermal stability, mechanical strength, low background current, regenerability, large potential window, and ease of modifying and/or functionalizing its surface without compromising the material properties.

Several examples introduced screen-printed carbon electrodes, modified with different gold nanomaterials [75,84,85,88,89,93,109,114,115,147,148,161,172,173,186].

To summarize, screen-printed technology provides important features, such as miniaturization of the devices, implementing the analysis on-site, cost-effective sensors, easy procedures, and small sample volumes.

Finally, exfoliated graphitic paper [185,191] and carbon ionic liquid electrodes [134,138,166] have to be evidenced as interesting electrodic materials to be modified with appropriate gold nanomaterials, representing promising sensing platforms.

The integration of nanomaterials in the development of electrochemical (bio)sensors was crucial for improving their analytical performances.

We have to stress that, in most cases, nanocomposites and/or nanohybrids have been adopted, including not only gold nanomaterials, but also other nanomaterials of different nature, or polymers, both natural and synthetic, at times developing very complex nanostructures with tailored architecture. The combination of different materials, integrating the electron transfer capability, the conductivity, and the electrocatalytic properties likely result in improved analytical performances of the sensor in terms of the linearity range, detection limit, stability, and so on.

In the majority of cases, gold nanoparticles were used as the gold nanomaterial, because the corresponding synthesis processes are well established and easier to perform. Several examples involved the combination of gold nanomaterials with carbon nanomaterials, generally graphene and its derivatives, while carbon nanotubes were used much less frequently, at least in the most recent examples. This is likely due to the absence of metal impurities and the higher surface area of graphene-based nanomaterials.

Considering polymers, traditional conducting polymers such as polypyrrole, polyaniline, and PEDOT [82,84,94,95,124,135,181,197] were preferentially employed for their conductivity and electrocatalytic properties, while natural polymers such as chitosan or gelatin and eggshell membranes [74,80,81,147,186] acted as green and eco-friendly reducing, stabilizing, and dispersing agents for gold nanomaterials.

At this point, we have to consider the role of the key recognition element. The trend reported in this review indicated that chemosensors represent the main sensor typology. Likely, the introduction of innovative and ad hoc designed nanomaterials in the electrochemical sensing area evidenced several enhancements regarding some critical issues, such as selectivity, fouling, and electrodic surface inactivation.

Considering the use of an enzyme as the key recognition element, examples of enzyme-based biosensors are few [74,84,93,114,163,167,181,182,189] compared to the total number of examples reported. It is well-known that electrochemical enzyme-based biosensors are easy to assemble and generally reusable, but their major drawback is the enzyme’s stability over time.

The number of examples using a molecularly imprinted polymer as the key recognition element was comparable to that of the enzyme-based sensors [115,122,123,124,173,174,186,191,193,196], indicating that it could represent a promising synthetic receptor to be used instead of a “classical” enzyme.

Examining the analytical performances of all reported sensors, the linearity ranges and detection limits were at least at the micromolar level, with several examples involving nanomolar concentrations with LODs at the picomolar level. In our opinion, a clear trend indicating the most performant sensing system cannot be evidenced. On the other hand, it is to be underlined that having the most performant system in terms of a wide linearity range and the lowest detection limit is not particularly important because it depends on the target analyte and the kind of detection field, i.e., food, clinical, pharmaceutical.

Sensor selectivity requires separate comments. In most examples, it is unclear how potentially interfering compounds were chosen, whether considering those with similar molecular structures or those present in the matrix to be analyzed. Very often, they seem to be randomly chosen. Consequently, it is difficult to compare the selectivity of different sensors with the same target analyte.

Moreover, in terms of the reproducibility, repeatability, and stability of the sensors, it is difficult to compare the data from different sensors, even with the same target analyte, because comparable experimental procedures are not used. For example, when considering long-term stability, durability and storage conditions such as temperature and wet or dry storage are different from case to case and do not allow a proper comparison of data.

Generally, the sensors reported in this review were applied to spiked real samples, which appears to be fundamental in the perspective of using them in the real life, but validation with a reference external method, such as HPLC, spectrophotometry, or LC-MS, was not always performed. In our opinion, the validation step is mandatory to compare the sensing system’s performance with those of the more traditional analytical methods, which are expensive, time demanding, and require skilled personal.

Another important issue is the capability of simultaneously detecting different analytes in a sample, as this is an attractive factor for commercial exploitation. As a general strategy, electrochemical approaches for multiple and simultaneous detection of phenolic antioxidants involved (bio)sensing platforms and devices using nanomaterials and/or hybrid nanocomposites ad hoc synthesized.

Different examples of multianalyte detection have been introduced in this review, involving phenolic acids, flavonoids, and synthetic phenolic antioxidants [74,146,184,187,189].

It should be noted that the examples are few, and several aspects have to be improved, such as implementing micro- and nano-fabrication techniques allowing multianalyte detection with the same device, and finally, integrating these sensing platforms into portable systems.

In general, smart portability can be obtained by integrating electrochemical sensors with ICT devices such as smartphones and tablets, but no example concerning the electrochemical detection of phenolic antioxidants has been reported.

As a future perspective, the combination of gold nanomaterials with artificial receptors such as aptamers or peptides for the determination of phenolic antioxidants should be evaluated, analogously with what is reported in the literature for antibiotics and pesticides, where many examples are available.

As a general comment, accurate and detailed investigations concerning the toxicity and degradation of nanomaterials are required, even if gold is generally considered a stable biocompatible material.

In particular, it should be mentioned that the wide use and combinations of nanomaterials must address different issues, ranging from the sustainability of nanostructures in sensor applications to their sustainable design and synthesis. For this reason, further efforts must be made to synthesize more environmentally friendly nanocomposites; some examples are already present [74,80,81,136,147,182,186], but the task is still challenging and complex.

Despite their advantages, at present, electrochemical (bio)sensors are not commercially available, except for very specific and well-known examples such as the glucometer. We have to consider several drawbacks, including:

(1) The lack of stability of the biorecognition element. This is a general problem, but it can be partially solved using a synthetic recognition element and/or an ad hoc modified sensing layer, as indicated in this review, where very few examples involved biomolecules.

(2) A complex synthetic procedure and the use of expensive materials. Many sensors, in order to enhance their analytical performances such as the limit of detection, linearity range, and so on, are based on a highly complicated and difficult-to-reproduce synthesis design and/or involve expensive materials (nanoparticles, rare elements, etc.). Consequently, real competition with conventional analysis techniques is difficult.

(3) The barrier to marketing. The industrial interest in the marketing of electrochemical (bio)sensors depends on the market demand. In addition, it is important to consider that many costs are associated with new technologies. Before commercialization, regulatory agencies must approve the product, and all apparatus, instrumentation, and analytical procedures must be appropriate and suitable for producing low-cost and reproducible devices on an industrial scale.

(4) Finally, the appropriate testing of real samples. Some devices, despite presenting interesting analytical performance in the laboratory, are not appropriate for application in complex real matrices because of the presence of interfering species not previously examined, biofouling of the electrodic surface, and the analyte behavior in a real matrix, e.g., an unexpected complex formation or chemical reactions with molecules present in the real samples. These are still great challenges to be addressed in order to make general electrochemical biosensors commercially available.

On the other hand, we have to consider that different technological advancements can represent effective tools for dealing with the marketing challenges. These include, for example, the use of 3D-printed electrodes (for lower costs and wider accessibility), the design of innovative nanocomposites, the use of innovative synthetic and/or bioreceptors, and the development of flexible devices, therefore increasing the number of sensing system applications.

In particular, regarding 3D-printing technology, the electrochemistry field can certainly take advantage of the available low-cost electrodes to design a new generation of electrochemical sensor and biosensor devices, therefore replacing conventional electrodes such as glassy carbon, screen-printed carbon, and carbon composite electrodes and improving electrodic surface functionalization and biomolecule immobilization.

## Figures and Tables

**Figure 1 nanomaterials-12-00959-f001:**
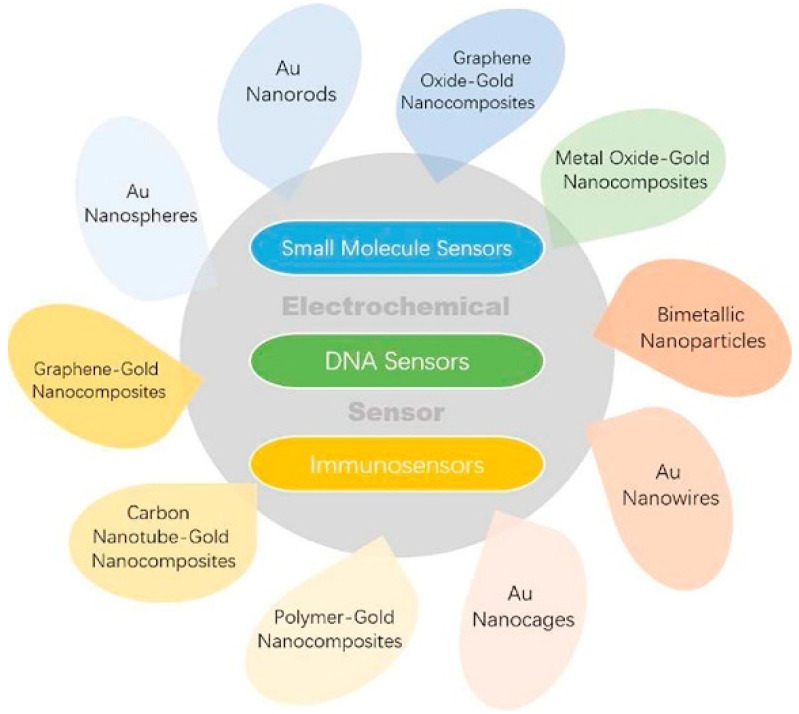
Electrochemical sensor applications of Au and Au-based nanomaterials. Reprinted with permission from [39] Copyright 2020, Elsevier.

**Figure 2 nanomaterials-12-00959-f002:**
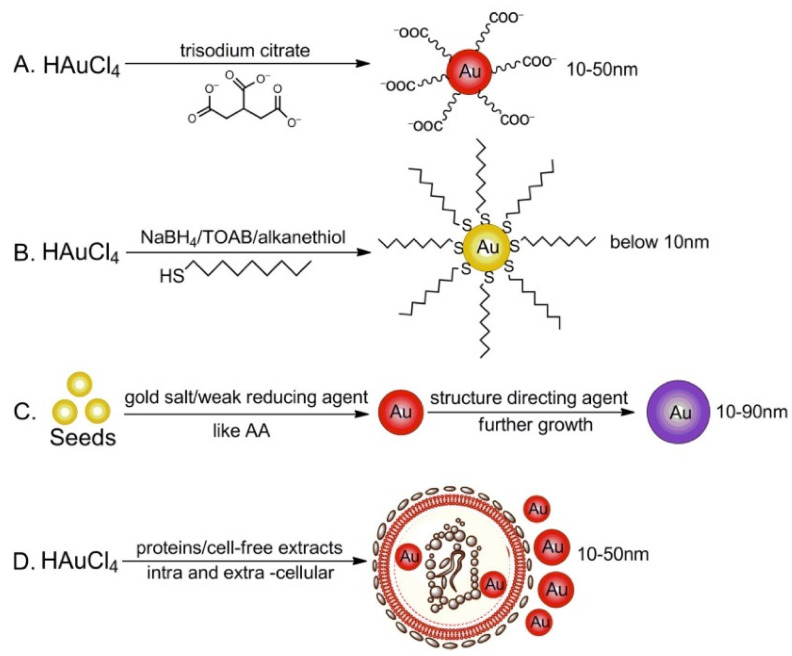
(**A**) Turkevich–Frens method for synthesis of AuNPs via reduction of gold salts in the presence of trisodium citrate; (**B**) Brust–Schiffrin strategy for two-phase fabrication of small-size AuNPs via reduction of gold salts in the presence of thiol ligands; (**C**) seed-mediated growth method for AuNPs; (**D**) green synthesis of non-toxic AuNPs through intra- and extra-cellular biosynthesis in the presence of proteins or cell-free extracts. Reprinted with permission from [40] Copyright 2018 Elsevier.

**Figure 3 nanomaterials-12-00959-f003:**
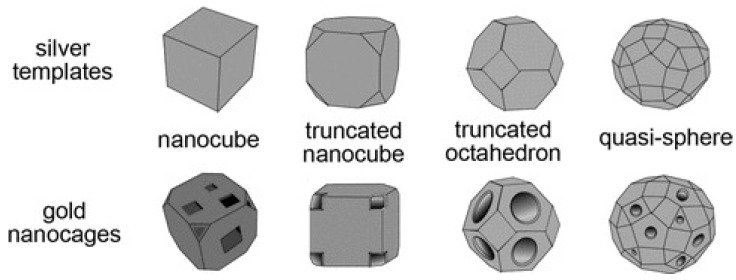
Schematic illustrations summarizing four major types of Au nanocages derived from the corresponding templates of Ag: A single-crystal cube with sharp corners; a single-crystal cube with truncated corners; a single-crystal octahedron with truncated corners; a polycrystalline, quasi-spherical particle. Reprinted with permission from [51]. Copyright 2010, Wiley.

**Figure 4 nanomaterials-12-00959-f004:**
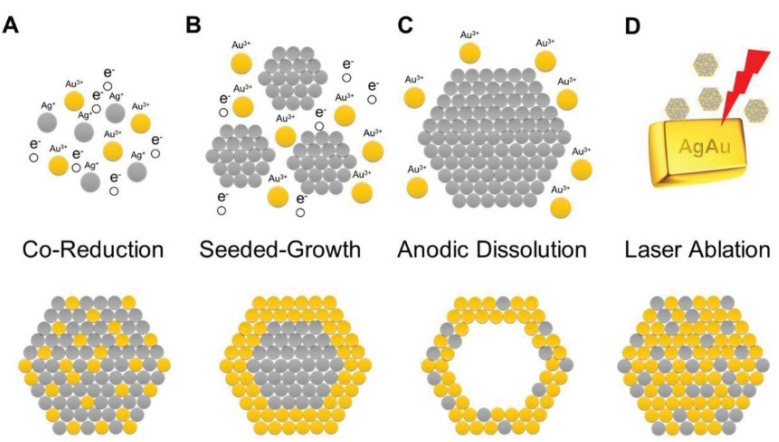
Schematic representations of the synthesis of alloyed bimetallic nanoparticles considering the silver-gold system. (**A**) Bottom-up synthesis, leading to alloyed silver-gold nanoparticles after co-reduction. (**B**) Bottom-up synthesis, leading to silver-gold core-shell nanoparticles (seeded-growth approach). (**C**) Bottom-up synthesis, leading to hollow gold nanoshells (anodic dissolution of the silver core). (**D**) Top-down synthesis to prepare alloyed silver-gold nanoparticles starting from a bimetallic alloy by laser ablation. Reprinted with permission from [57]. Copyright 2020, Wiley.

**Figure 5 nanomaterials-12-00959-f005:**
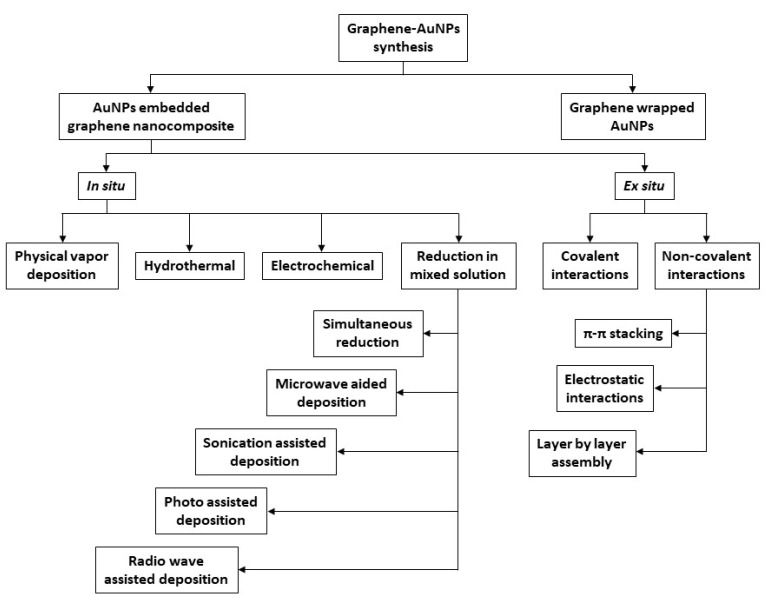
Schematic representation of the synthetic pathways of graphene-AuNPs nanocomposites.

**Figure 6 nanomaterials-12-00959-f006:**
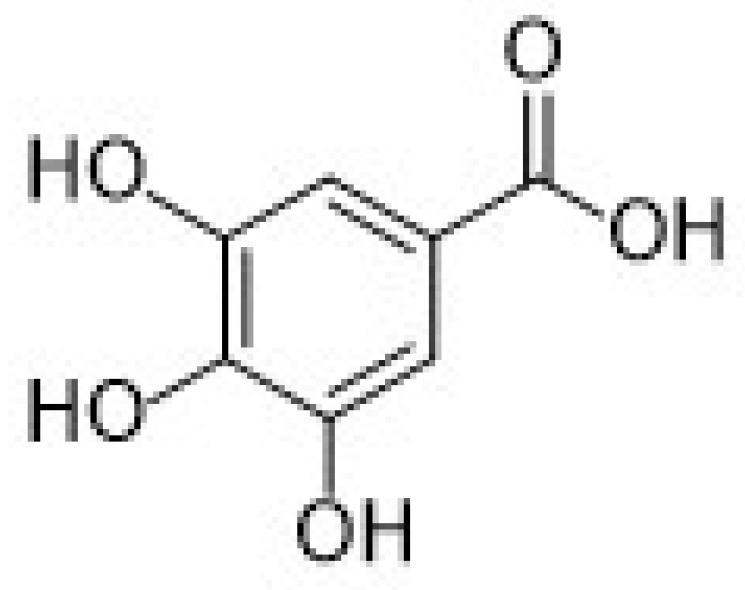
Chemical structure of gallic acid (GA).

**Figure 7 nanomaterials-12-00959-f007:**
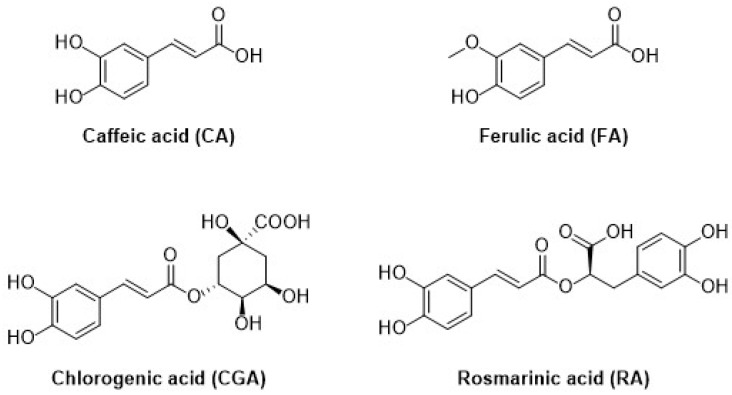
Schematic representation of the hydroxycinnamic acids and derivatives considered in this review.

**Figure 8 nanomaterials-12-00959-f008:**
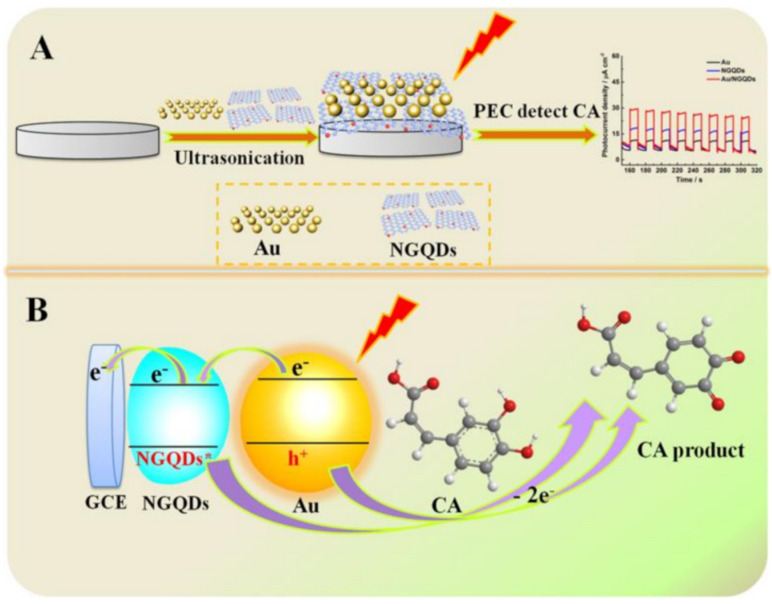
(**A**) Assembly of Au/NGQDs sensing platform for the photo-electrochemical (PEC) detection of caffeic acid (CA). (**B**) Schematic illustration of the sensing mechanism. Reprinted from [98].

**Figure 9 nanomaterials-12-00959-f009:**
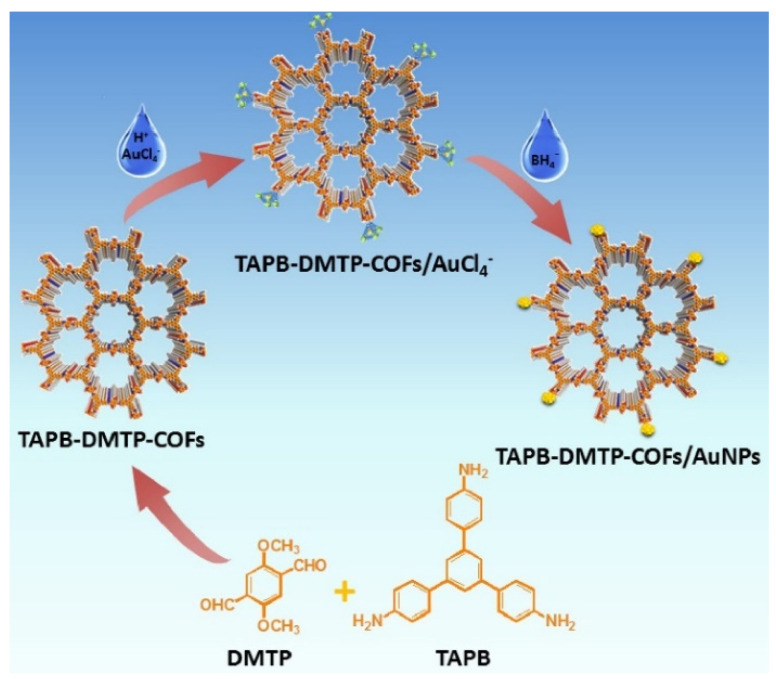
Illustration of synthesis route for TAPB-DMTP-COFs/AuNPs. Reprinted with permission from [103]. Copyright 2018, Elsevier.

**Figure 10 nanomaterials-12-00959-f010:**
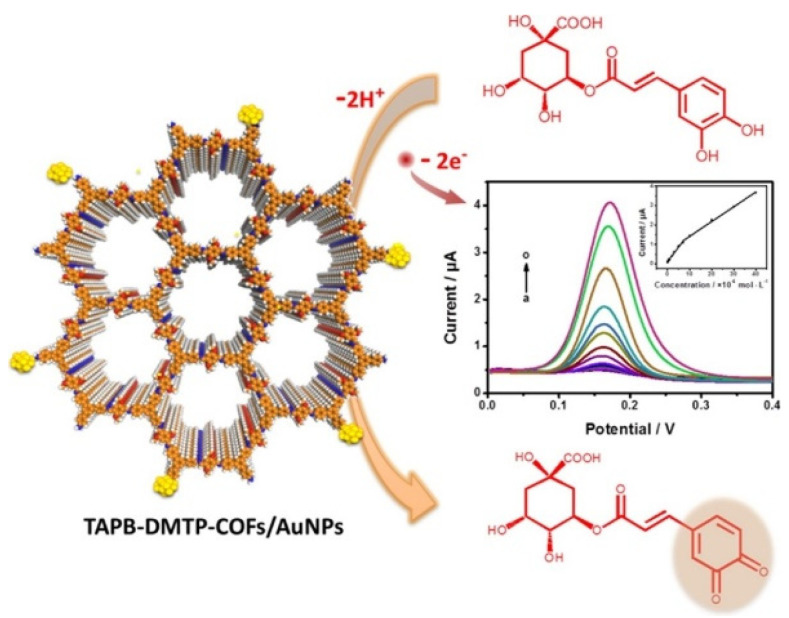
Schematic representation of GCA detection at TAPB-DMTP-COFs/AuNPs/GCE. Reprinted with permission from [103]. Copyright 2018, Elsevier.

**Figure 11 nanomaterials-12-00959-f011:**
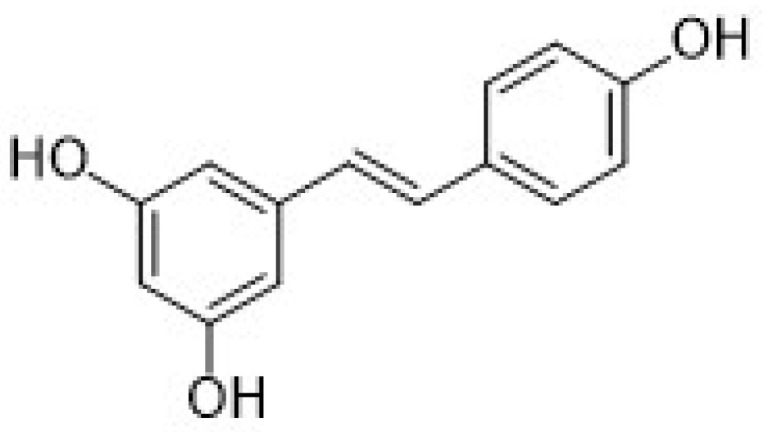
Chemical structure of resveratrol (RES).

**Figure 12 nanomaterials-12-00959-f012:**
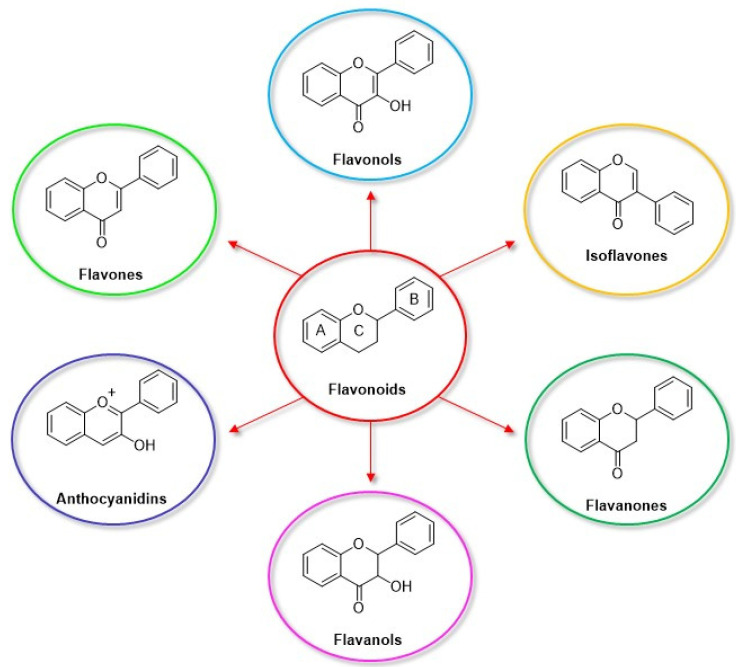
Structures of flavonoids and their subgroup classification.

**Figure 13 nanomaterials-12-00959-f013:**
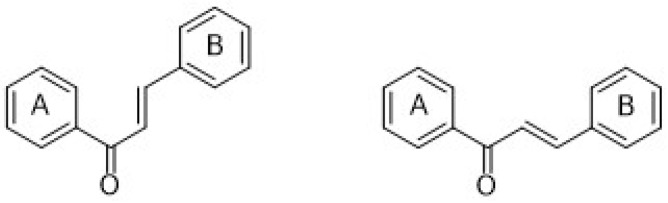
General structure of chalcones.

**Figure 14 nanomaterials-12-00959-f014:**
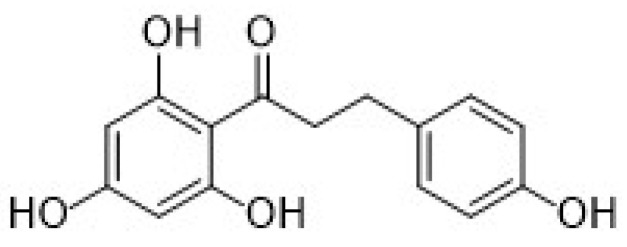
Chemical structure of phloretin (PH).

**Figure 15 nanomaterials-12-00959-f015:**
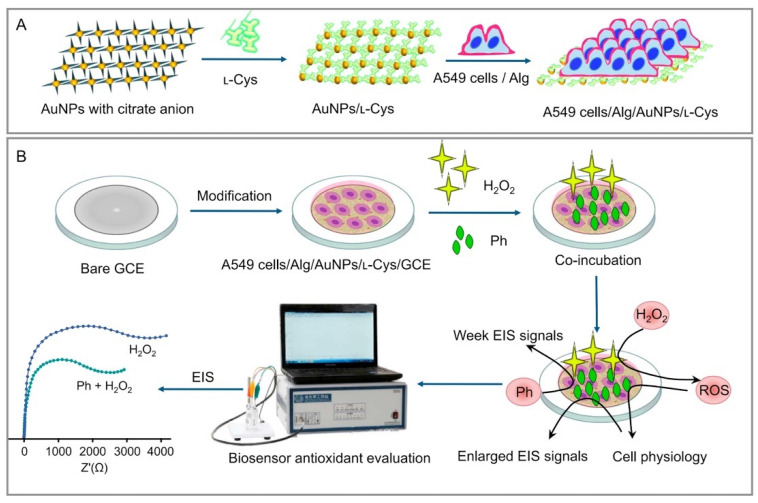
Schematic illustration of the assembly of A549 cell-based sensor (**A**) and of the evaluation process of the PH antioxidant capacity (**B**). Reprinted with permission from [129]. Copyright 2018, Elsevier.

**Figure 16 nanomaterials-12-00959-f016:**
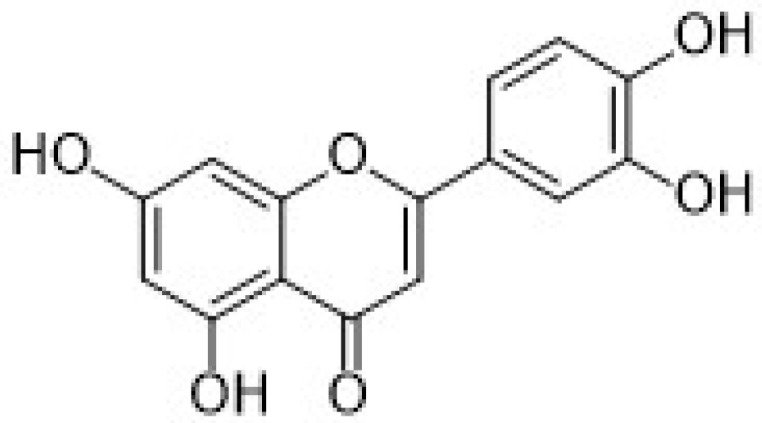
Chemical structure of luteolin (LUT).

**Figure 17 nanomaterials-12-00959-f017:**
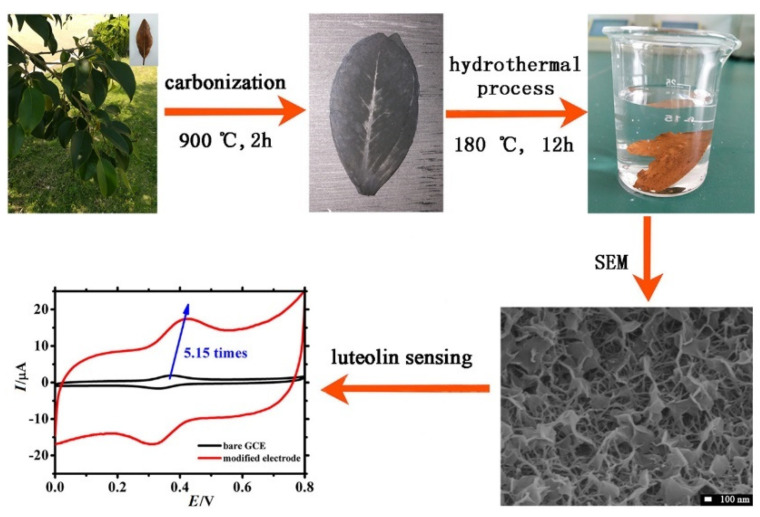
Schematic illustration of the synthesis process of AuNFs-BPC and the LUT electrochemical determination. Reprinted with permission from [136]. Copyright 2021, Elsevier.

**Figure 18 nanomaterials-12-00959-f018:**
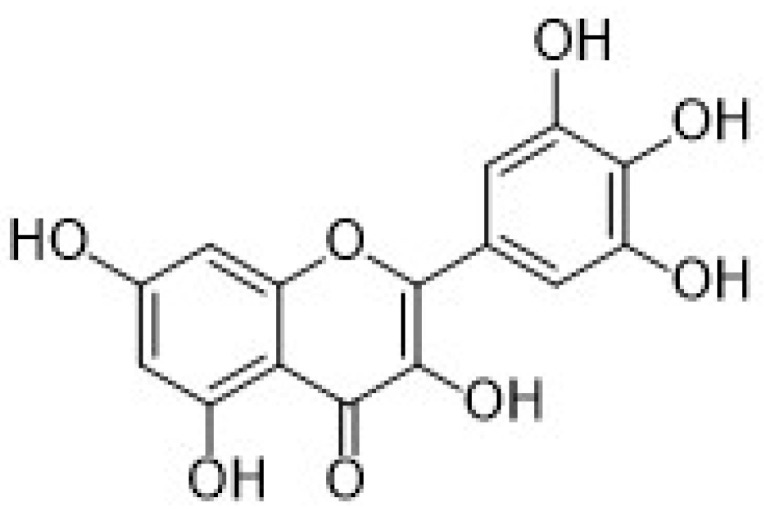
Chemical structure of myricetin (MYR).

**Figure 19 nanomaterials-12-00959-f019:**
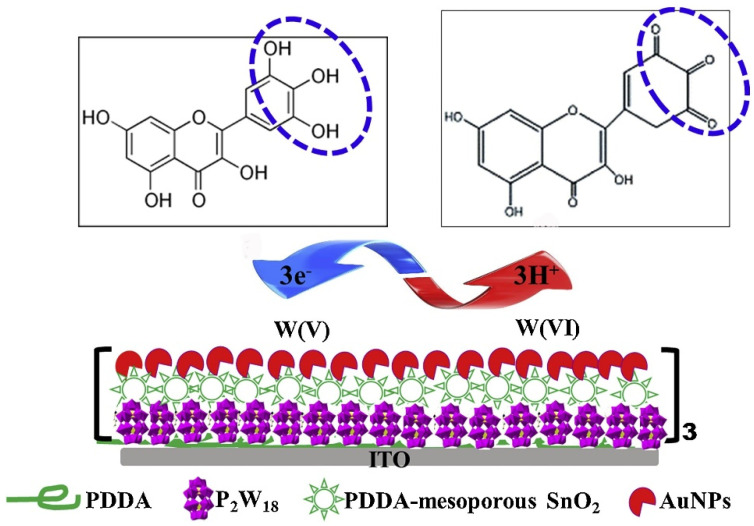
A schematic representation of the electrochemical oxidation of myricetin at the (P_2_W_18_-SnO_2_-AuNPs)_3_/ITO electrode. Reprinted with permission from [143]. Copyright 2019, Elsevier.

**Figure 20 nanomaterials-12-00959-f020:**
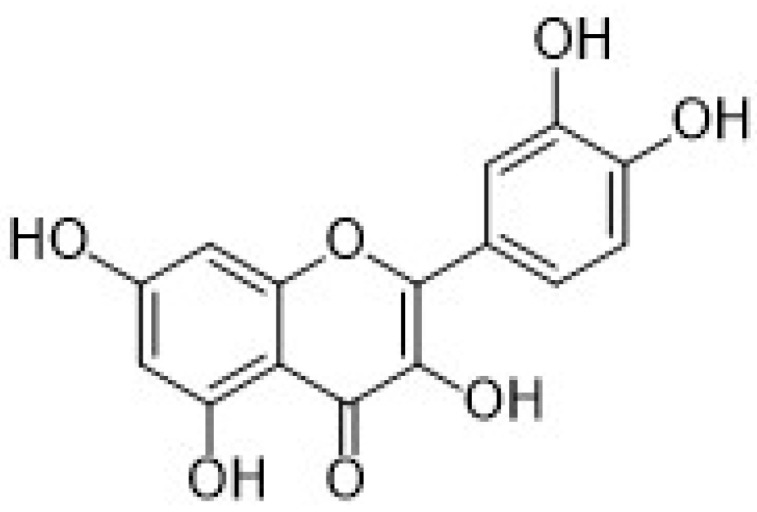
Chemical structure of quercetin (QR).

**Figure 21 nanomaterials-12-00959-f021:**
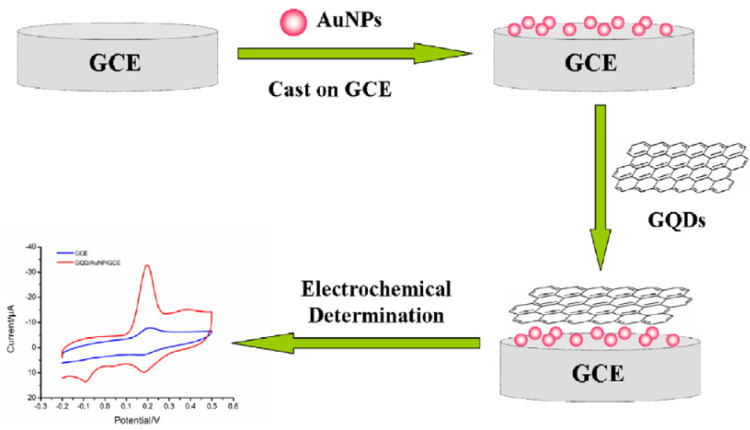
Schematic illustration of assembling of GODs/AuNPs/GCE and the electrochemical determination of quercetin (QR). Reprinted with permission from [158]. Copyright 2016, Wiley.

**Figure 22 nanomaterials-12-00959-f022:**
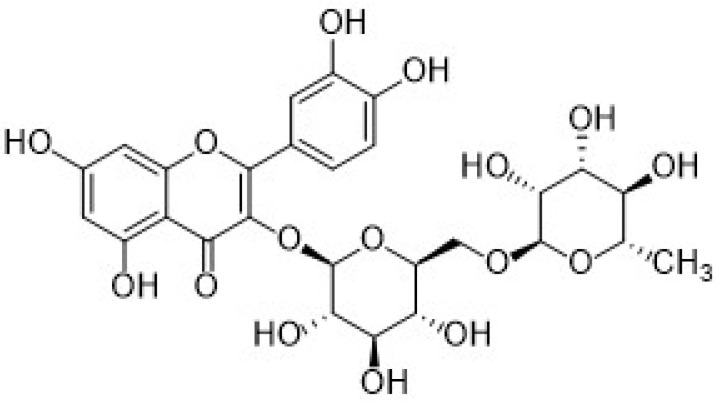
Chemical structure of rutin (RT).

**Figure 23 nanomaterials-12-00959-f023:**
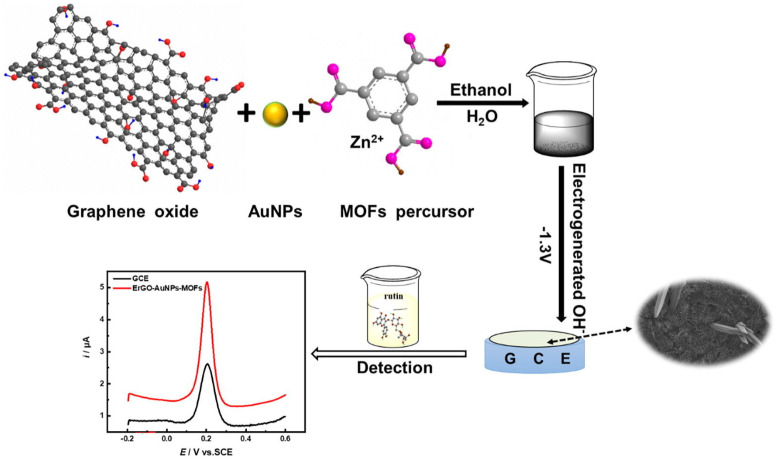
Assembliy of ErGO-AuNPs-MOFs/GCE as rutin electrochemical sensor. Reprinted with permission from [168]. Copyright 2021, Elsevier.

**Figure 24 nanomaterials-12-00959-f024:**
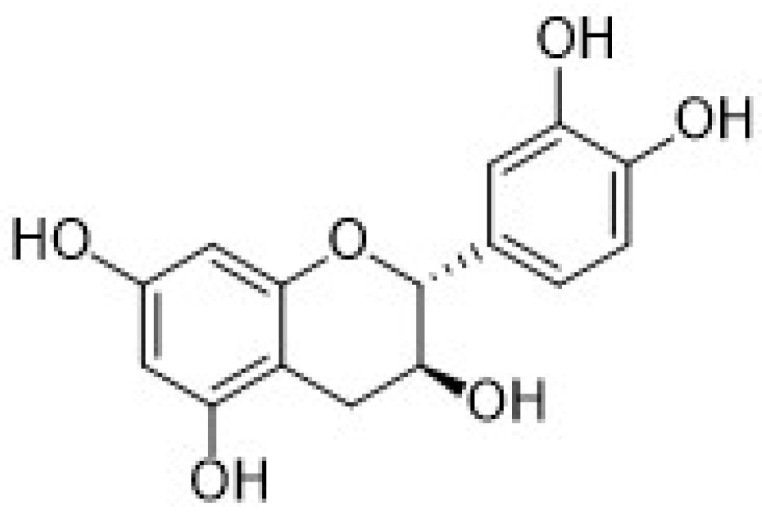
Chemical structure of catechin (CAT).

**Figure 25 nanomaterials-12-00959-f025:**
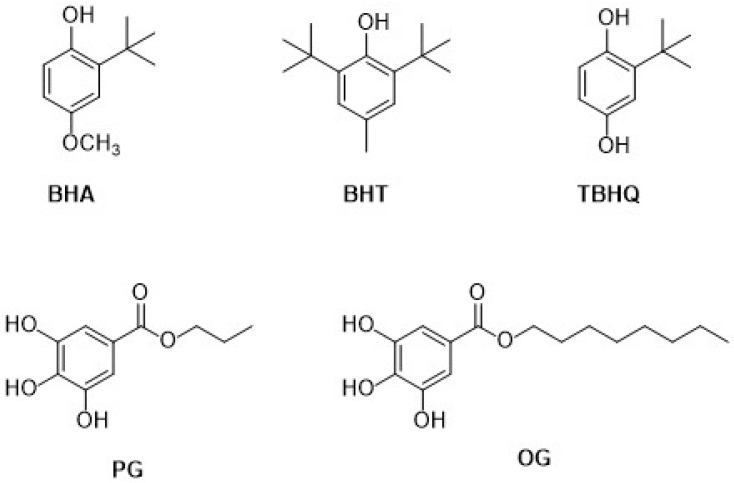
Chemical structure of the most common SPAs.

**Figure 26 nanomaterials-12-00959-f026:**
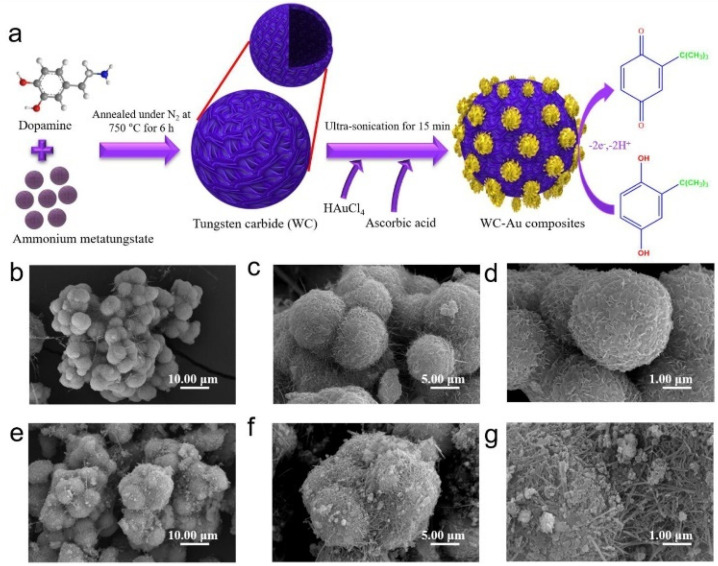
(**a**) Schematic representation of the AuNP-WC composite synthesis and of TBHQ oxidation mechanism. SEM images of (**b**–**d**) WC and (**e**–**g**) AuNP-WC at different magnifications. Reprinted with permission from [194]. Copyright 2021, Elsevier.

**Table 1 nanomaterials-12-00959-t001:** Performances of electrochemical sensors for phenolic acid detection.

Electrode	(Bio)sensor Format	Electrochemical Technique	Analyte/Sample	Linearity Range (μM)	LOD (μM)	Recovery (%)	Reference Method	Ref.
GCE	Electrochemical sensor based on AuMCs/G nanocomposite	DPV	GA/black teas, urine	0.05–8.00	0.0007	96–102.4	HPLC	[72]
CPE	Electrochemical sensor based on ZrO_2_NPs/ChCl/ AuNPs nanocomposite	DPV	GA/green teas, urine, fruit juices	0.22–55.00	0.025	97.9–103.8	no	[73]
GCE	Electrochemical biosensor based on TYR immobilized by crosslinking on AuNPs coated with ESM	DPV	GA, CA, CH/black tea, red wine	GA 6–70 CA 5–65 CH 5–115	GA 1.707 CA 0.752 CH 0.714	no	HPLC	[74]
SPCE	Electrochemical sensor based on AuNPs prepared using *Achantophora algae* extracts as reducing agent	DPV	GA/*Acanthophora algae* extracts	no	no	no	no	[75]
AuE	Electrochemical sensor based on AuNPs/CHI nanocomposites	DPV	CA/red and white wines	0.050–2000	0.025	no	no	[80,81]
GCE	Electrochemical sensor based on PEDOT/AuNPs nanocomposite	CV	CA/fruit juices	10–1000	4.24	no	HPLC	[82]
GCE	Electrochemical sensor based on PEDOT-MeSH/YRFC/AuNPs	DPV	CA/human serum, urine	0.02–320	0.014	97.4–105.0	no	[83]
SPCE	Biosensor obtained by AuNPs electrodeposition, LAC immobilization, pyrrole electropolymerization	Amperometry	CA/propolis extracts	1–250	0.83	no	FC	[84]
AuNPs/SPCE	Biosensor obtained by TYR immobilization on AuNPs/SPCE	Amperometry	CA/beers	2.5–12.50	2.3	no	no	[85]
GCE	Electrochemical sensor obtained modifying a GCE with trimetallic PtAuRu NPs	DPV	CA	7.9–16600	0.39	no	no	[87]
SPCE	Electrochemical sensor based on g-C_3_N_4_/AuNPs nanocomposite	DPV/amperometry	CA/fruits, leaves, wine	DPV 0.0005–0.155 Amperometry 0.0025–1.025	DPV 0.0001 Amperometry 0.0005	98.0–100.3	no	[88]
SPCE	Electrochemical sensor based on WS_2_/catechin capped AuNPs/CB nanocomposite	DPV	CA/rapeseed oil, *Kalanchoe crenata*, apple puree, apple homogenized, apple juices	0.3–200	0.1	86.0–108.0	no	[89]
GCE	Electrochemical sensor based on GNs/AuNPs nanocomposite	DPV	CA/drug samples	0.50–50	0.05	97.6–101.8	no	[92]
MWCNTs/SPCE	Electrochemical biosensor obtained by immobilizing TvL and depositing AuNPs simultaneously	Chronoamperometry	CA	1–100	0.5	no	no	[93]
GCE	Electrochemical sensor based on PdAu/PEDOT/rG nanocomposite	DPV	CA/red wine	0.001–55	0.00037	98–104	no	[94]
GCE	Electrochemical sensor based on Au/PEDOT/rG nanocomposite	DPV	CA/red wine	0.01–46	0.004	no	no	[95]
GCE	Electrochemical sensor based on AuPdNPs/GFs nanocomposite	DPV	CA/red wine	0.03–938.97	0.006	no	no	[96]
GCE	Electrochemical sensor based on Au-αFe_2_O_3_ NPs/rGO nanocomposite	DPV	CA/coffee	19–1869	0.098	no	no	[97]
GCE	Electrochemical sensor based on AuNPs/NGQDs nanocomposite	Amperometry	CA	0.11–30.25 30.25–280.25	0.03	no	no	[98]
GCE	Electrochemical sensor based on AuNPs-POM/MPC nanocomposite	DPV	CGA/pharmaceutical samples	0.00228–3.24	0.00215	no	Spectrophotometric analysis	[100]
GCE	Electrochemical sensor based on AuNPs doped COF composite	DPV	CGA/coffee, apple juice, honeysuckle	0.010–40	0.0095	99.2–102.5	HPLC	[103]
GCE	Electrochemical sensor based on AuNPs modified electrode	SWV	CGA/black and green teas	0.0011–0.010	0.040	98–99	no	[108]
AuNPs/G/SPCE	Electrochemical sensor using commercial SPCEs modified with AuNPs and G	CV	CGA/nutraceutical products	0.1–1.20	0.062	no	FT-IR	[109]
CNFs/AuNPs/SPCE	Electrochemical biosensor obtained immobilizing TYR on commercial SPCEs modified with AuNPs and CNFs	CV	FA/cosmetic products	0.1–1.60	0.00289	no	FT-IR	[114]
SPCE	Electrochemical sensor based on MIP/AuNPs/GO nanocomposite	DPV	FA/orange peels	0.01–1 2–10	0.0031	99–103	no	[115]
CPE	Electrochemical biosensor obtained immobilizing PER in AuNPs-BMIPF_6_- CTN composite	SWV	RA/pharmaceutical samples	0.50–23.70	0.070	98–106.2	no	[118]

Abbreviations: AuE: Gold electrode; AuMCs: Gold microclusters; AuNPs: Gold nanoparticles; BMIPF_6_: 1-butyl-3-methylimidazolium hexafluorophosphate; CA: Caffeic acid; CB: Carbon black; CGA: Chlorogenic acid; CGE: Glassy carbon electrode; ChCl: Choline chloride; CH: Catechin hydrate; CHI: Chitosan; g-C_3_N_4_: Graphitic carbon nitride; CNFs: Carbon nanofibers; CPE: Carbon paste electrode, CTN: Chitin; CV: Cyclic voltammetry; DPV: Differential pulse voltammetry; FA: Ferulic acid; G: Graphene; GA: Gallic acid; GNs: Graphene nanosheets; rGO: Reduced graphene oxide; GO: Graphene oxide; YRCF: Resorcinol formaldehyde yolk-shell-structured carbon spheres; LAC: Laccase; MeSH: Methane thiol; LOD: Limit of detection; MIP: Molecular imprinted polymer; MPC: Macroporous carbon; MWCNTs: Multi-walled carbon nanotubes; NGQDs: N-doped graphene quantum dots; PEDOT: Poly(3,4-ethylenedioxythiophene); PER: Peroxidase; POM: Polyoxomethalates; PPY: Polypyrrole; RA: Rosmarinic acid; SPCE: Screen-printed carbon electrode; SWV: Square-wave voltammetry; SWASV: Square-wave anodic stripping voltammetry; TYR: Tyrosinase; TvL: Laccase from *Trametes versicolor*.

**Table 2 nanomaterials-12-00959-t002:** Performances of electrochemical sensors for resveratrol detection.

Electrode	(Bio)Sensor Format	Electrochemical Technique	Analyte/Sample	Linearity Range (μM)	LOD (μM)	Recovery (%)	Reference Method	Ref.
ITOE	Electrochemical sensor based on MIP/Au@Ag nanoshells nanocomposite	CV	RES/grape seed extracts	2.0 × 10^−5^–9.0 × 10^−^^3^	7.1 × 10^−^^6^	96.7–106.3	HPLC	[122]
GCE	Electrochemical sensor based on MIP/AuNPs/ GNs nanocomposite	CV	RES/red wine, grape skin	0.01–10	0.0044	94.5–102.5	HPLC	[123]
GCE	Electrochemical sensor based on MIP/AuNPs/ PANI	DPV	RES/red wine	1.0–200	0.087	97.7–108	HPLC	[124]

Abbreviations: AuNPs: Gold nanoparticles; CV: Cyclic voltammetry; DPV: Differential pulse voltammetry; GCE: Glassy carbon electrode; GNs: Graphene nanosheets; ITOE: Indium-tin oxide electrode; LOD: Limit of detection; MIP: Molecularly imprinted polymer; PANI: Polyaniline; RES: Resveratrol; SPCE: Screen-printed carbon electrode.

**Table 3 nanomaterials-12-00959-t003:** Performances of electrochemical sensors for flavonoid detection.

Electrode	(Bio)sensor Format	Electrochemical Technique	Analyte/Sample	Linearity Range (μM)	LOD (μM)	Recovery (%)	Reference Method	Ref.
GCE	Electrochemical sensor based on BNNs/AuNPs nanocomposite	SWV	LUT/peanuts hull, *Perilla*	5 × 10^−6^–0.0012 0.02–10	1.7 × 10^−6^	98.3–108.38	HPLC	[132]
CILE	Electrochemical sensor based on AuNCs-modified CILE	DPV	LUT/drugs	0.001–1	0.0004	95.0–96.7	no	[134]
GCE	Electrochemical sensor based on AuNPs/MWCNTS/PEDOT nanohybrid	SWV	LUT/human serum	1–100 100–15000	0.22	99.63–103.1	no	[135]
GCE	Electrochemical sensor based on AuNPs/BPC/GCE	DPV	LUT/drugs and human urine	0.15–1.8 1.8–10	0.07	97.00–105.10 (drugs) 98.5–106.17 (urine)	no	[136]
GCE	Electrochemical sensor based on AuNPs/SH-β-CD-GNs/GCE	DPV	LUT/human serum	1.0 × 10^−5^–10	3.3x10^−6^	97.1–103.2	no	[137]
CILE	Electrochemical sensor based on AuNPs/3DG-modified CILE	DPV	LUT/pharmaceutical products	0.05–50	0.00759	95.28–103.77	no	[138]
GCE	Electrochemical sensor based on AuNPs/GQDs/GCE	DPV	LUT/peanut hull	0.01–10	0.001	98.8–101.4	no	[139]
GCE	Electrochemical sensor based on AuNPs/enMWCNTs/GCE	CVAdSV	MYR/black and green teas, fruit juices	0.050–40	0.0120	101–104	no	[141]
CE	Electrochemical sensor based on AuNPs/CE	SWV	MYR/no real samples	6.3–31.4	1.3	no	no	[142]
ITOE	Electrochemical sensor based on AuNPs-mSnO_2_NS/ POM/ITO	Amperometry	MYR/fruit juices	1–110	0.067	97.65–103.07	no	[143]
GCE	Electrochemical sensor based on AuNPs/MWCNTs/GCE	SWV	QR,RT/fruit juices	0.001–0.050	0.00033	96.3–99.4	no	[146]
SPCE	Electrochemical sensor based on AuNPs/GEL/SPCE	DPV	QR/onion, apple	0.02–34.5	0.0019	99.2–99.9	no	[147]
PE	Electrochemical sensor based on AuNPs/CeO_2_ NPs@FGCM/PE	SWV	QR/apple and grape juices green tea, honeysuckle	0.048–1.09	0.00037	97.8–102	HPLC and spectrophotometry	[148]
GCE	Electrochemical sensor based on Fe_3_O_4_@SiO_2_/AuNPs/PANI/ GCE	DPV	QR/apple juice, tea, radish leaves, human serum and urine	0.01–15	0.0038	95.6–101.1	no	[151]
GCE	Electrochemical sensor based on Au-Co@NCNHP/GCE	DPV	QR/onion and drugs	0.050–35	0.023	97.96–102.5	HPLC	[153]
AuE	Electrochemical sensor based on hp-Au/AuE	Amperometry	QR/apple juice, green tea, onion, honeysuckle, pharmaceutical products	0.02–100	0.0039	97.43–101.59	no	[155]
NPG-AuE	Electrochemical sensor based on NPG-Au/AuE	DPV	QR/no real samples	0.01–12 12–100	0.0011	no	no	[156]
GCE	Electrochemical sensor based on AuNPs/GO/GCE	DPV	QR/pharmaceutical products	1.0 × 10^−6^–10 × 10^−5^	3.0 × 10^−6^	no	no	[157]
GCE	Electrochemical sensor based on AuNPs/GQDs/GCE	DPV	QR/human serum	0.01–6	0.002	95–104.6	no	[158]
GCE	Electrochemical sensor based on PB-rGO/TCD/AuNPs/GCE	DPV	QR/apple juice, red wine, honeysuckle	0.005–0.4	0.00183	95–104.3	no	[159]
GCE	Electrochemical sensor based on NH_2_-GQDs/AuNPs-β-CD/GCE	DPV	QR/honey, tea, honeysuckle, human serum	0.001–0.21	0.285 × 10^−3^	97.5–106.2	no	[160]
SPCE	Electrochemical sensor based on GQDs/AuNPs/SPCE	SWV	QR/human serum	0.0001–1000	3.3 × 10^−5^	92.6–101.7	no	[161]
CPE	Electrochemical biosensor based AuNPs-CD/LAC/CPE	SWV	RT/pharmaceutical products	0.30–2.97	0.14	96.8–103.8	Spectrophotometry	[163]
GCE	Electrochemical sensor based on AuNPs/HCNTs/GCE	Amperometry	RT/pharmaceutical products	0.1–30	0.081	99.6–102.1	no	[164]
CILE	Electrochemical sensor based on AuCNs/CILE	DPV	RT/pharmaceutical products	0.004–700.0	0.00133	94.00–108.40	no	[166]
GCE	Electrochemical biosensor based PPO/AuNPs/EDU-15/GCE	Amperometry	RT/black tea	1.5–28	0.51	98.5–105.5	no	[167]
GCE	Electrochemical sensor based on ErGO/AuNPs-MOFs/GCE	DPV	RT/pharmaceutical products	0.007–0.14 0.14–0.4	0.00344	86.9–104.8	no	[168]
GCE	Electrochemical sensor based on Au@ AuPtNPs/B-doped grahene/GCE	DPV	RT/pharmaceutical products	0.002–4.0	2.84x10^−4^	97.23–101.65	no	[169]
ANE	Electrochemical sensor based on AuNPs/G/ANE	DPV	RT/pharmaceutical products, human urine	0.08–10 20–200	0.025	97.40–98.05 (pharmaceutical products) 97.20–103.60 (urine)	UV-vis	[170]
GCE	Electrochemical sensor based on Au-Ag NTs/NG/GCE	DPV	RT/pharmaceutical products	0.1–420	0.015	97–106	no	[171]
SPCE	Electrochemical sensor based on AuNPs/G/SPCE	SWV	RT/pharmaceutical products	0.1–15	0.011	96.52–102.97	UV-vis	[172]
SPCE	Electrochemical sensor based on MIP/AuNPs/G/SPCE	DPV	RT/pharmaceutical products	0.04–60	0.014	98.0–104.7	no	[173]
GCE	Electrochemical sensor based on MIP/AuNPs-MoS_2_-G/GCE	DPV	RT/gingko biloba, buckwheat tea, pharmaceutical products	0.01–45	0.004	95.2–104.7	no	[174]
GCE	Electrochemical sensor based on Cu_2_O-AuNPs/NG/GCE	DPV	RT/pharmaceutical products	0.06–512.90	0.03	98.95–101.81	HPLC	[177]
GCE	Electrochemical sensor based on NS-rGO/AuNPs/GCE	DPV	RT/pharmaceutical products	0.0002–1.4	6.7 × 10^−5^	97.08–100.90	no	[178]
Pt	Electrochemical biosensor based on TYR/AuNPs-PPY/Pt	CV	CAT/apple juices	0.001–0.01	0.0012	no	LC/MS/MS	[181]
GCE	Electrochemical biosensor based on TYR/AuNPs/GCE	DPV	CAT/teas	no	7.7	no	HPLC	[182]
AuE	Electrochemical sensor based on NG/Au@Pt NPs/AuE	DPV	CAT/teas	0.1–45	0.00285	99.94–101.50	no	[183]

Abbreviations: ANE: Acupuncture needle electrode; AuE: Gold electrode; AuNPs: Gold nanoparticles; AuCNs: Gold nanocages; BNNs: Boron nitride nanosheets; BPC: Biomass-derived porous carbon; CAT: Catechin; CD: Cyclodextrin; CE: Graphite electrode; CILE: Carbon ionic liquid electrode; CV: Cyclic voltammetry; CVAdSV: Cyclic voltammetry adsorptive stripping voltammetry; 3DG: Three-dimensional graphene; DPV: Differential pulse voltammetry; FGCMs: Functionalized glassy carbon microspheres; GCE: Glassy carbon electrode; G: Graphene; GEL: Gelatin; GQDs: Graphene quantum dots; GNs: Graphene nanosheets; rGO: Reduced graphene oxide; GO: Graphene oxide; HCNTs: Helical carbon nanotubes; hp-Au: High porous gold; ITOE: Indium-tin oxide electrode; LAC: Laccase; LOD: Limit of detection; LUT: Luteolin; MIP: Molecular imprinted polymer; MOFs: Metal organic frameworks; MWCNTs: Multi-walled carbon nanotubes; MYR: Myricetin; NCNHP: N-doped carbon nanotube hollow polyhedron; NG: N-doped graphene; NGQDs: N-doped graphene quantum dots; NPG: Nanoporous gold; NS: Nanospheres; NTs: Nanotubes; PB-rGO: Graphene oxide functionalized with 1-pyrenebutyrate; PE: Paste electrode; PEDOT: Poly(3,4-ethylenedioxythiophene); PER: Peroxidase; POM: Polyoxometalate; PPY: Polypyrrole; QR: Quercetin; RT: Rutin; SH-β-CD: Mercapto-β-cyclodextrin; SPCE: Screen-printed carbon electrode; SWV: Square-wave voltammetry; SWASV: Square-wave anodic stripping voltammetry; TCD: Thiocyclodextrin; TYR: Tyrosinase.

**Table 4 nanomaterials-12-00959-t004:** Performances of electrochemical sensors for synthetic phenolic antioxidant detection.

Electrode	(Bio)sensor Format	Electrochemical Technique	Analyte/Sample	Linearity Range (μM)	LOD (μM)	Recovery (%)	Reference Method	Ref.
GCE	Electrochemical sensor based on GCE modified with AuNPs	LSV	BHA, BHT, TBHQ/edible oils	BHA 0.55–8.32 BHT 0.9–10 TBHQ 1.2–16.8	BHA 0.22 BHT 0.36 TBHQ 0.48	BHA 93.7–105 BHT 92.3–97 TBHQ 91.6–103	HPLC	[184]
EGP	Electrochemical sensor based on EGP modified with AuNPs and NiO NPs.	DPV	BHA/edible oils	0.03–50	0.02	97.00–104.00	no	[185]
SPCE	Electrochemical sensor based on SPCE modified with AuNPs and CHI, acting as MIP	DPV	BHA/chewing-gum, mayonnaise, potato chips	0.055–111	0.0055	no	Spectrophotometry	[186]
GCE	Electrochemical sensor based on a binary nanocomposite including AuNPs and ErGO	LSV	BHA, TBHQ/ edible oils	BHA 0.55–55.5 TBHQ 0.6–42	BHA 0.23 TBHQ 0.30	no	HPLC	[187]
GCE	Electrochemical sensor based on a nanocomposite including AuNPs, G and PVP as dispersing agent	LSV	BHA/ flour, soybean oil	0.2–100	0.04	93–105	no	[188]
GCE	Electrochemical biosensor using AuNPs, HRP and SAP NTs	LSV	BHA, PG/ olive and peanut oils, potato chips, cookies	BHA 1.7–277 PG 0.47–470	BHA 0.26 PG 0.11	87.3–126.2	HPLC	[189]
GCE	Electrochemical sensor using a nanocomposite including Au and SnO_2_ NPs, GNs, MWCNTs	DPV	TBHQ/edible oils, teas	0.05–230	0.058	97.9–100.8	HPLC	[190]
EGP	Electrochemical sensor based on EGP modified with AuNPs and MIP	DPV	TBHQ/edible oils	0.08–100	0.07	95.80–102.3	no	[191]
GCE	Electrochemical sensor based on GCE modified with AuPt bimetallic nanocrystals	DPV	TBHQ/canola oil, soybean oil	0.35–625	0.075	Soybean oil 99.2–102.1 Canola oil 98.4–102.5	no	[192]
GCE	Electrochemical-based sensor based on GCE modified with Au-PdNPs, ErGO and MIP	DPV	TBHQ/edible oils	3.0–361	0.28	99.4–108.5	HPLC	[193]
GCE	Electrochemical-based sensor based on GCE modified with WC-AuNPs composite	SWV	TBHQ/soybean oil, blended oil, red wine	0.005–0.075	0.0002	Blended oil 88–97.3 Soybean oil 92–99 Red wine 94–97	HPLC	[194]
GCE	Electrochemical-based sensor based on GCE modified with MIP/PtAuNPs-G/CNTs composite	Chronoamperometry	PG/vegetable oils	0.070–10	0.0251	98.3–103.0	no	[196]
GCE	Electrochemical-based sensor based on GCE modified with AuNPs/poly(p-ABSA) composite	DPV	PG/vegetable oils	9–100	0.19	98.9–101.3	no	[197]
GCE	Electrochemical sensor based on GCE modified with AuNPs and DDT as SAM	SWV	OG/ margarine, butter, and sunflower oil.	0.20–1.20	0.0083	99.1–100.7	Spectrophotometry	[198]

Abbreviations: AuNPs: Gold nanoparticles; BHA: Butylated hydroxyanisole; BHT: Butylated hydroxytoluene; CHI: Chitosan; DPV: Differential pulse voltammetry; DDT: Dodecane thiol; G: Graphene; GNs: Graphene nanosheets; EGP: Exfoliated graphite paper; ErGO: Electroreduced graphene oxide; GCE: Glassy carbon electrode; GO: Graphene oxide; HRP: Horseradish peroxidase; LSV: Linear sweep voltammetry; LOD: Limit of detection; MIP: Molecular imprinted polymer; MWCNTs: Multi-walled carbon nanotubes; NPs: Nanoparticles; NTs: Nanotubes; OG: Octyl gallate; PG: Propyl gallate; poly(***p****-*ABSA): Poly(*p-*aminobenzenesulfonic acid; SAM: Self-assembling monolayer; SAP NTs: Spiny Au-Pt nanotubes; SPCE: Screen-printed carbon electrode; SWV: Square-wave voltammetry; TBHQ: Tert-butylhydroquinone; WC: Tungsten carbide.

## Data Availability

Not applicable.

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
