# Peer review of "Gold Nanomaterials-Based Electrochemical Sensors and Biosensors for Phenolic Antioxidants Detection: Recent Advances"

_nanomaterials, 2022, doi:10.3390/nano12060959_

Round 1

Reviewer 1 Report

This work is interesting. Particularly, the writing style is non-conventional in dealing with literature. However, authors should consider the following necessary changes.

  1. The introduction would have been better with proper language. Compared to the body of the review, it seems weak.
  2. I see that authors have written numerous small sentences as paragraphs throughout the draft. These sentences must be combined to form suitable paragraphs. e.g lines 970-990, the description is for one article but distributed in many small paragraphs. 
  3. Many spelling errors
  4. What is the need for table 1? All the information related to the references is provided in the text then why this is again repeated in table 1. for instance, see reference 117 its LOD, linear range, recovery, reference method, analysis technique, (bio) sensor format are discussed in text (lines 999-1014) and the same information is in table 1 as well.

Author Response

The reply to reviewer 1 is in the attached file

Reviewer 2 Report

The review article entitled Application of gold nanomaterials to electrochemical (bio)sensors for phenolic antioxidants detection. submitted to Nanomaterials Journal. This manuscript is generally well written and clearly presented however still needs to address some comments, and thus require substantial major revision to improve the quality of the manuscript.

  1. Title should modify which can describe the whole research theme.
  2. Abstract looks very general Add more discussion related to the research,  and future research directions as well.
  3. Provide a nice graphical abstract representing the overview of the MS with key highlights. For review article it should be compulsory.
  4. In the introduction section, write the novelty of the work and the problem statement clearly. Authors fails to explain the novelty and importance of the proposed research work thus substantial discussion is essential. Some investigators recently used gold nanosensors for the detection of antibiotics Chemosensors 9 (12), 358, 2021 authors should refer and cite somewhere. 
  5. All text should be rearrange so many paragraphs. Add one figure describing the applications and advantages and disadvantages of the research topic.
  6. Techno Economic challenges and limitations of the developed system should be included ?

  7. The conclusion of the study needs to be added with the specific output obtained from the study, it could be modified with precise outcomes with a take home message. The present form looks like repetition of abstract

  8.  

    Some English and grammar mistakes are present that need to be correct to improve the quality of the manuscript.

Author Response

The reply to reviewer 2 is present in the attached file

Reviewer 3 Report

The manuscript analyzes and discuss most recently reported examples on the development of electrochemical sensors and biosensors based on a broad variety of gold nanomaterials for the quantification of phenolic antioxidants. The discussion of the reported papers is focused on the characteristics of the nanomaterials involved and on the analytical performance of the devices.

Since there are no equivalent reviews and the contribution is well organized and useful, I recommend its publication after attending to the following comments and suggestions:

  1. Owing to the length of the manuscript, it is convenient to include a table of contents.
  2. Lines 290-291. The authors must clarify the reason why only the labelled sensors are included in the review.
  3. Figure 5. The scheme of figure 5 must be restructured in relation to covalent interactions since it considers non-covalent interactions such as electrostatic ones.
  4. Lines: 470-472, 489-490, 668-669, 687-688, 731-732, 748-749, 770, 790-791, 866-867, 922-923, 948-949, 982-983, 1009, 1089-1090, 1106, 1194, 1211-1212, 1228-1229, 1254-1255, 1288, 1296, 1325, 1357-1358, 1384-1385, 1402-1403, 1455, 1509-1510, 1527, 1542-1543, 1566, 1588, 1603, 1621-1622, 1653-1654, 1680, 1698, 1738, 1791-1792, 1808-1809, 1824-1825, 1846-1847, 1872-1873, 1909-1910, 1938-1939, 2042, 2054-2055, 2104, 2121, 2167, 2203, 2223, 2251, 2262-2263, 2276. The repeatability, reproducibility, reusability and stability is reported in the manuscript in an ambiguous way as "interesting", "promising", "acceptable" "good", "satisfactory". The authors must report the values of these analytical parameters.
  5. Tables 1, 2, 3 and 4. To make easier the contrast of results between the devices, please report the linearity range as well as the LOD in the same units.
  6. Lines 1135-1136. The authors must clarify the reason why only the labelled flavonoids are included in the review.
  7. Figure 19. Symbol of P2W18 is missing bellow of the scheme.
  8. Line 1612. Please replace “GDQs” with “GQDs”.
  9. Line 2045. The discussion of the contrast between methods must be included.

Author Response

The reply to the reviewer 3 is present in the attached file

Round 2

Reviewer 1 Report

No comments

Reviewer 2 Report

Authors have substantially improved the manuscript according to the comments. Thus the present form of the manuscript can be accepted for publication.